# Cenozoic history of the tropical marine biodiversity hotspot

Skye Yunshu Tian[1,2,3,4,5,19 ✉], Moriaki Yasuhara[1,2,3,4,6,19 ✉], Fabien L. Condamine[7,19], Huai-Hsuan M. Huang[8], Allan Gil S. Fernando[9], Yolanda M. Aguilar[10], Hita Pandita[11], Toshiaki Irizuki[12], Hokuto Iwatani[13], Caren P. Shin[14,15], Willem Renema[16,17] & Tomoki Kase[18]

The region with the highest marine biodiversity on our planet is known as the Coral Triangle or Indo-Australian Archipelago (IAA)[1,2]. Its enormous biodiversity has long attracted the interest of biologists; however, the detailed evolutionary history of the IAA biodiversity hotspot remains poorly understood[3]. Here we present a high-resolution reconstruction of the Cenozoic diversity history of the IAA by inferring speciation–extinction dynamics using a comprehensive fossil dataset. We found that the IAA has exhibited a unidirectional diversification trend since about 25 million years ago, following a roughly logistic increase until a diversity plateau beginning about 2.6 million years ago. The growth of diversity was primarily controlled by diversity dependency and habitat size, and also facilitated by the alleviation of thermal stress after 13.9 million years ago. Distinct net diversification peaks were recorded at about 25, 20, 16, 12 and 5 million years ago, which were probably related to major tectonic events in addition to climate transitions. Key biogeographic processes had far-reaching effects on the IAA diversity as shown by the long-term waning of the Tethyan descendants versus the waxing of cosmopolitan and IAA taxa. Finally, it seems that the absence of major extinctions and the Cenozoic cooling have been essential in making the IAA the richest marine biodiversity hotspot on Earth.

It is unclear how the global centre of marine biodiversity, the IAA, has developed, and why its biodiversity is disproportionally high compared to that of other tropical regions. These are key uncertainties in organismal biology. We have gradually gained a better understanding of global-scale diversity dynamics throughout the Cenozoic[4,5], revealing complex waxing and waning of diversity related to climatic and other environmental changes. However, regionally resolved Cenozoic diversity trends remain poorly understood owing to the scarcity of historical data and their compilation. This is particularly true for the tropics in deeper time, making the origins of high biodiversity an enigma[6–8]. The current knowledge of the fossil record suggests that the locations of peak diversity (that is, biodiversity hotspots) shifted throughout the Cenozoic from the western Tethys during the Eocene to the Arabian Peninsula during the late Eocene–Oligocene, before being established at the current location of the IAA in Southeast Asia in the early Miocene: the process known as the hopping hotspots model[3,9]. Plate tectonics is postulated to be the ultimate driver of this process by regulating the broad-scale availability and configuration of shallow-marine habitats with successive continent collisions[9]. Each hop of the biodiversity hotspots from the ancient location to the new one was probably underpinned by considerable speciation and extinction events, but also could be associated with the palaeobiogeographic shifts of some component taxa tracking suitable habitats[3]. However, the detailed Cenozoic history of the IAA hotspot remains elusive as explicated below. Better understanding the deep-time origin, evolution and maintenance of this most diverse place in the marine realm is crucial for macroevolutionary and macroecological studies and provides a solid theoretical framework for conservation efforts.

As one of the most conspicuous biogeographic and biodiversity patterns today, the IAA hotspot is characterized by an exceptional concentration of coastal benthic species, whereas pelagic groups show widespread distributions without a distinguished centre of diversity[2,10]. Historical evidence from benthic taxonomic groups has advanced our understanding of tropical diversification yet suffers from various limitations. Recent molecular studies on corals and reef fishes revealed their biogeographic and evolutionary history to build the IAA hotspot,

[1]School of Biological Sciences, Area of Ecology and Biodiversity, The University of Hong Kong, Hong Kong, Hong Kong SAR. [2]Swire Institute of Marine Science, The University of Hong Kong, Hong Kong, Hong Kong SAR. [3]Institute for Climate and Carbon Neutrality, The University of Hong Kong, Hong Kong, Hong Kong SAR. [4]Musketeers Foundation Institute of Data Science, The University of Hong Kong, Hong Kong, Hong Kong SAR. [5]Bonner Institut für Organismische Biologie, Paläontologie, Universität Bonn, Bonn, Germany. [6]State Key Laboratory of Marine Pollution, City University of Hong Kong, Hong Kong, Hong Kong SAR. [7]CNRS, Institut des Sciences de l'Evolution de Montpellier, Université de Montpellier, Montpellier, France. [8]Department of Geosciences, Princeton University, Princeton, NJ, USA. [9]National Institute of Geological Sciences, University of the Philippines, Diliman, Quezon City, The Philippines. [10]Marine Geological Survey, Mines and Geosciences Bureau, Quezon City, The Philippines. [11]Department of Geological Engineering, Faculty of Mineral Technology, Institute Teknologi Nasional Yogyakarta, Yogyakarta, Indonesia. [12]Department of Geoscience, Interdisciplinary Graduate School of Science and Engineering, Shimane University, Matsue, Japan. [13]Division of Earth Science, Graduate School of Sciences and Technology for Innovation, Yamaguchi University, Yamaguchi, Japan. [14]Paleontological Research Institution, Ithaca, NY, USA. [15]Department of Earth and Atmospheric Sciences, Cornell University, New York, NY, USA. [16]Naturalis Biodiversity Center, Leiden, The Netherlands. [17]IBED, University of Amsterdam, Amsterdam, The Netherlands. [18]National Museum of Nature and Science, Department of Geology and Paleontology, Tsukuba, Japan. [19]These authors contributed equally: Skye Yunshu Tian, Moriaki Yasuhara, Fabien L. Condamine. ✉e-mail: skyeystian@gmail.com; moriakiyasuhara@gmail.com

suggesting that it was a centre of species accumulation and origin at different intervals of the Cenozoic[11,12]. However, these phylogeny-based models are subject to uncertainties in the case of incomplete sampling and undocumented extinctions. They also are not geographically defined to quantify a clear scenario of regional diversity changes in the IAA. In addition to molecular phylogeny evidence, previous palaeontological studies collectively indicate a compatible trend of increased IAA species richness at a coarse spatiotemporal resolution, which was dominated by immigration into the IAA in the Eocene–Oligocene and proliferation inside the IAA in the Oligocene–recent[6,9,13–15]. However, difficulties in reconstructing a more detailed IAA history for benthic fossil groups include: a lack of sufficient fossil data or data synthesis for molluscs and bryozoans[8,15]; a small species pool despite a good fossil record for larger benthic foraminifera[6]; and uncertainties in species-level identification for fossil corals[7]. Another critical, yet often overlooked, limitation is that our current knowledge of the marine hotspot is severely biased towards (sub)tropical reef-associated groups (mostly corals and reef fishes), even though the IAA peak in species richness is a common feature shared by many shallow-marine lineages regardless of reef affiliation[10]. By relying on reef taxa as the sole descriptor of the IAA hotspot, the strong correlation of their diversity patterns with reef habitat may mask the effects of other putative primary drivers on macroevolutionary and macroecological dynamics. Overall, Ostracoda (Arthropoda: Crustacea; known as seed shrimps) is one of the few benthic microfossil organisms that has left a rich fossil record within and beyond reef ecosystems for quantitative analysis[16]. Their high species diversity and robust taxonomy are two other major advantages[16]. Benthic ostracods show a normal latitudinal diversity gradient[17] and depth diversity gradient[18]. They also exhibit a similar biogeographic distribution with other invertebrates[19]. Thus, instead of being a contrarian, ostracods are regarded as a normal benthic taxon that tends to follow standard ecological patterns[17]. In addition, small (<0.5 mm) benthic metazoan invertebrates account for most (more than two-thirds of) marine biodiversity[17,20], and the ostracod is probably the best fossil representative for this group in terms of general biotic response[13,16]. These features make ostracods a very useful proxy for broad marine benthic biodiversity to investigate the ancient history of hotspots before the timescale of modern observations[13,16].

Here we used ostracods as a model proxy to assemble the first comprehensive Cenozoic fossil dataset for the IAA hotspot. Applying a birth–death model that incorporates the preservation process and mitigates bias in the fossil record, we first inferred a detailed regional diversity trajectory by explicitly estimating speciation and extinction rates. Given the regional scope of this study, speciation here corresponds to the first appearance of every species in the IAA to construct the emerging hotspot. The same rationale applies to extinction, which is defined as the final extirpation of any species from the IAA instead of global extinction. We then correlated the IAA's macroevolutionary dynamics with a set of biotic and abiotic parameters to assess the potential biodiversity drivers. Finally, we placed the evolution of the IAA hotspot in a biogeographic context to explore how key biogeographic processes (that is, migration and origination of the Tethyan, cosmopolitan and endemic IAA fauna) altered long-term diversification.

## Cenozoic diversity history of the IAA

We first assembled a Cenozoic fossil ostracod dataset from 216 samples across the IAA region (Extended Data Fig. 1), totalling 47,727 specimens for 874 morphospecies (including 94 spp. entries in which at least a part of a genus was impossible to identify at species level and was treated as 1 species entry apiece in the analyses). We estimated the best-fit model that accounts for the preservation rates of the dataset, which was a non-homogeneous Poisson process (significance at $P < 0.01$), in which preservation rates change over the lifetime of each lineage according to a bell-shaped distribution (median estimate of 45.04 occurrences per million years per taxon). Through a Bayesian process-based birth–death analysis of our integrated dataset, we quantitatively documented a detailed Cenozoic diversity history of the IAA hotspot, revealing the generation and maintenance of its exceptional high diversity (Fig. 1). Species richness was very low during most of the Palaeogene, consistent with the hypothesis that the IAA was not the place of a biodiversity hotspot during this period. With an initial increase in diversity in the late Oligocene, in line with the hopping hotspots model[9], rapid diversification began about 25 million years ago (Ma), making the IAA a rising hotspot. These early diversifications are probably broadly related to the collision of the southeast Eurasian margin with the Australian and Pacific plates and the resulting development of the vast area of complex habitat in the IAA[3,9,21]. Thereafter, strong diversification continued throughout the Miocene–Pliocene and culminated in the Pleistocene about 2.6 Ma, when species richness increased more than sixfold relative to the Eocene average, reaching an exceptionally high value of more than 650 species, and finally exhibiting a diversity plateau until the present. In addition to the long-term trends, speciation rates peaked at about 25, 20, 16, 12 and 5 Ma, corresponding to phases of rapid increase in the diversity curve. Extinction rates remained comparatively low throughout the Cenozoic, except for four peaks consistent with those of speciation rates at about 25, 20, 16 and 5 Ma. Nevertheless, it should be noted that there is a major sampling gap across the early Oligocene, so the possibility of an earlier diversification before 25 Ma cannot be completely ruled out[7,14]. Our raw analyses of the ostracod dataset, based on rarefaction and species range plots, indicated a consistent pattern of striking diversity increase at each geological interval from the early Miocene until the Pleistocene (Extended Data Figs. 2 and 3 and Supplementary Table 1). Within the IAA, the Philippines emerged as the bull's-eye of ostracod diversity from the late Miocene to Pleistocene, which is congruent with modern distributions of overall marine species richness[10,22] (Extended Data Fig. 4).

Most noticeably, extinction occurred at very low background rates in the IAA throughout its Cenozoic history, except for the small and discrete peaks as mentioned in the last paragraph (Fig. 1b). Our results are in agreement with a previous study that showed that huge spatial differences in extinction rates between the IAA and Caribbean may be crucial for the modern longitudinal diversity patterns across the tropical belt, with the largest hotspot situated in the IAA in contrast to much lower diversity in the Caribbean[8]. As two major hotspots of the early Neogene in similar tectonic and oceanographic settings, the IAA and the Caribbean may exhibit parallel processes of diversification during the Miocene to encompass comparable levels of species richness[3,8]. The modern disparity in diversity between the two regions may have developed during the Plio-Pleistocene with the final closure of the Central American Seaway between 4 and 2 Ma triggering the Caribbean mass extinction[3,23]. Such a catastrophic event terminated the Caribbean hotspot, whereas our analysis suggests that a longer trend of diversification continued smoothly in the IAA after the Miocene to promote modern-scale species richness. Thus, the absence of mass extinction is a prerequisite for the development and maturation of the IAA hotspot as a global centre of diversity under long-term stable and favourable palaeoenvironmental conditions across the Neogene and Quaternary. The present-day global tropical diversity pattern may be primarily shaped by deep-time extinctions, suggesting that the study of modern diversity and environments alone is not sufficient to fully understand the biosphere of our planet.

## Drivers of the Cenozoic diversification

We deciphered potential drivers of IAA's long-term diversity dynamics by simultaneously assessing the effects of key biotic (diversity dependency) and abiotic (habitat size and complexity, temperature and sea level) factors, using a multivariate birth–death (MBD) model spanning the entire time frames (that is, Eocene–present). The abiotic variables chosen here reflect the changes in tectonics, climate and oceanography

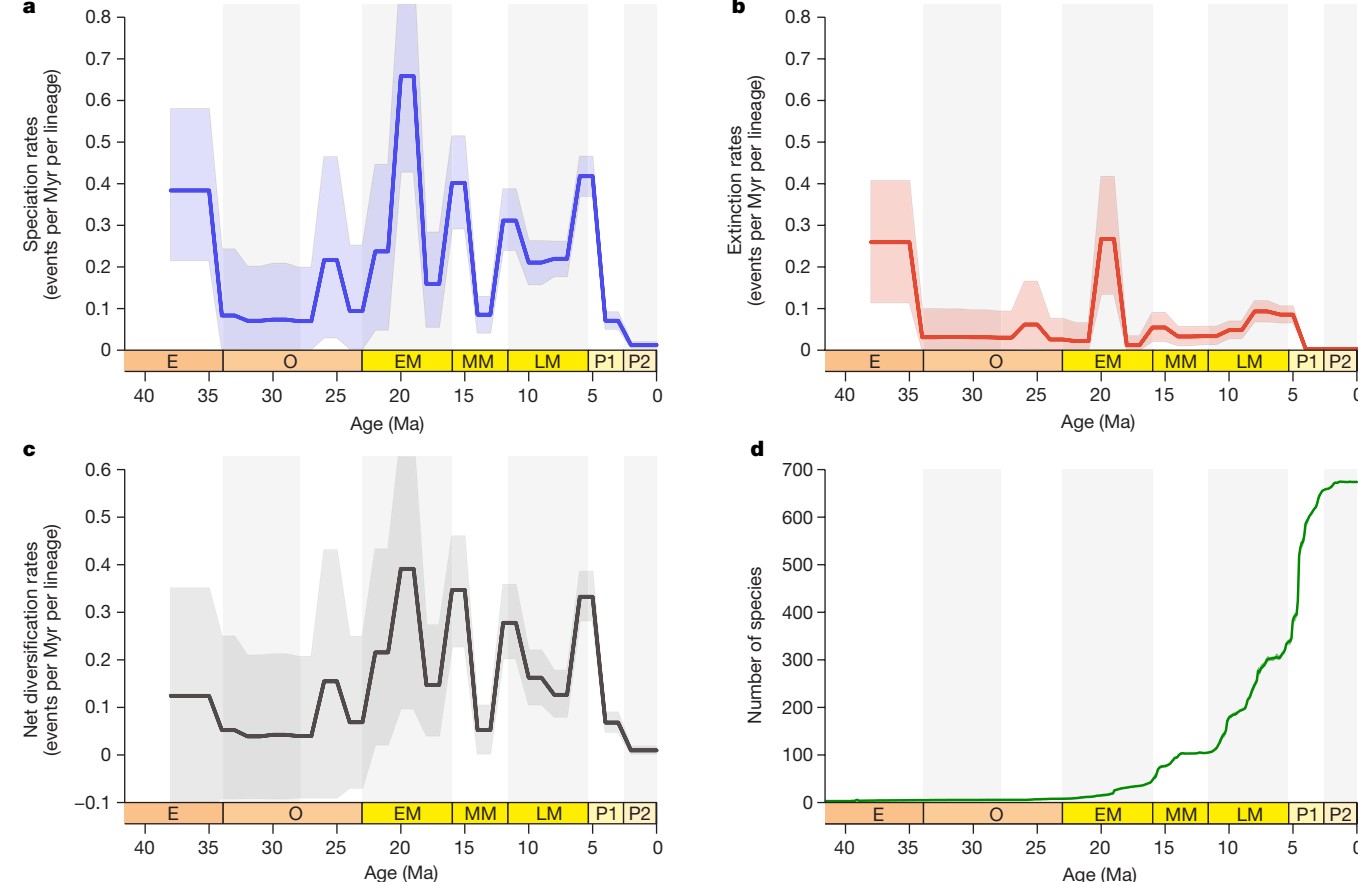

**Fig. 1 | Estimation of regional diversity dynamics in the IAA across the Cenozoic. a–d**, Bayesian inferences of speciation rates per lineage (**a**), extinction rates per lineage (**b**), net diversification rates as the difference between speciation and extinction rates (rates below 0 indicate declining diversity; **c**) and species richness of all ostracods (**d**). Solid lines indicate mean posterior rates, and the shaded areas show the 95% confidence interval. E, Eocene; O, Oligocene; EM, early Miocene; MM, middle Miocene, LM, late Miocene; P1, Pliocene; P2, Pleistocene. Early Oligocene, early Miocene, late Miocene and Pleistocene are shaded.

that may have shaped the most fundamental aspects of Earth's physical environment and played an overarching role in determining broad biological outcomes in the tropics[1,3]. First, we revealed a substantial degree of diversity dependence within ostracod assemblages, with a strong negative correlation between diversity and speciation rate (correlation parameter $G = -2.735$, strength of correlation parameter $\omega = 0.863$; Fig. 2a and Supplementary Table 2). Accordingly, we observe a long-term decrease in speciation rate and an approximately logistic growth of diversity for the Neogene–Quaternary development of the IAA hotspot (Fig. 1a,d), which is consistent with the bounded diversity hypothesis[24–26]. Indeed, lineages and communities may commonly experience a slowdown in their diversification rates coupled with a damped increase in diversity. Higher numbers of species may indicate increased biotic (competitive) interactions and expanded niche occupancy in a region, which conversely reduces ecological opportunities for speciation[24,26]. Finite resources, ultimately determined by extrinsic factors (for example, climate and regional area), may result in an upper limit to the niche space shared by all species, and thus hypothetically limit the maximum diversity that can be achieved as 'carrying capacity'[24]. Bounded diversification seems to be a recurrent and pervasive pattern in macroevolutionary studies across different times, places and taxa, although empirical evidence for diversity truly exhausting its potential to increase with niche saturation remains scarce[25]. In our case of the IAA hotspot, it seems that long-term diversification since about 25 Ma has led to a steady, asymptotic phase of equilibrium diversity following the Plio-Pleistocene transition about 2.6 Ma. This may represent the maturation of the IAA hotspot after a

long Neogene history of expansion, but the possibility of future diversity growth remains if the carrying capacity increases in response to changing environmental conditions, or if a key innovation evolves to conquer new niche space[25,26], which would make the current phase only a diversity plateau in the hotspot's life cycle.

In addition to diversity dependence that imposed a strong biotic control on the IAA, we showed that habitat size (shelf area) as the most important abiotic determinant has a positive, albeit weak, effect on speciation, indicating an evolutionary species–area relationship ($G = 2.272 \times 10^{-7}$, $\omega = 0.647$; Fig. 2a and Supplementary Table 2). Larger shelf areas could promote speciation either through direct effects of area or through the effects of factors that are highly correlated with area, such as population size, species range and environmental heterogeneity[27,28]. The Neogene expansion of shallow-marine habitats in the IAA, which was driven by the prolonged collisions between Southeast Asia and Australia[21,29], was therefore pivotal in the rise of the IAA hotspot. Unexpectedly, habitat complexity (coastline length) as another putative diversification driver has no effect here (insignificant correlation with both speciation and extinction rates; $\omega = 0.272$; Fig. 2a and Supplementary Table 2). Our results seem to contradict previous theories suggesting that complex island archipelagos of the IAA with an extensive array of shallow seas accelerate allopatric speciation through vicariance, although isolated small populations may also face higher extinction risks[1,3]. The effects of habitat complexity on regional diversification are worthy of further investigation. Indeed, in the marine realm, dispersal (both positive and negative) as the dominant process may overwhelm the effects of isolation at a regional scale, for which the physical barriers

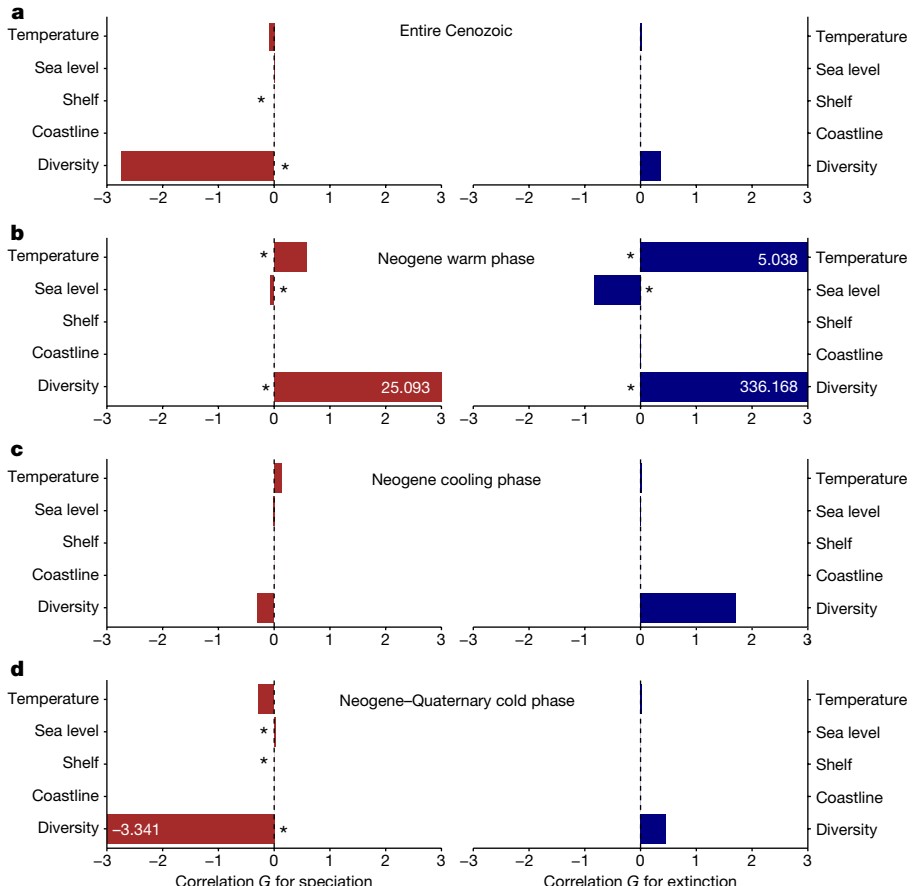

**Fig. 2 | Environmental controls on diversity dynamics in the IAA across the Cenozoic. a–d**, Bayesian inferences of correlation parameters on speciation (red) and extinction (blue) with abiotic factors including global temperature, global sea level, IAA habitat size (shelf area) and IAA habitat complexity (coastline length), and diversity dependence factor (that is, the diversity over time of the entire ostracod assemblage), for the time frame of the entire Cenozoic (**a**), Neogene warm phase (23.04–13.9 Ma; **b**), Neogene cooling phase (13.9–5.33 Ma; **c**) and Neogene–Quaternary cold phase (5.33–0 Ma; **d**). The asterisk indicates a significant correlation parameter for a given variable (shrinkage weights $\omega > 0.5$).

are often permeable and relatively sparse[30–32]. Connectivity among marine basins may be substantially restricted because of a dynamic mosaic of ever-changing geographic and oceanographic conditions within the IAA, but not strictly or permanently blocked, and thus undermines the effectiveness of conventional allopatry. Instead, our findings may imply the importance of sympatric speciation, given the positive effect of habitat size but no effect of habitat complexity[31,33]. In large, continuous shelf habitats across wide environmental gradients, such as the IAA, diversification may occur along ecological partitions (that is, intense competition invokes finer subdivision and niche specialization) in addition to geographic partitions[25,31–33]. Evidence of sympatric speciation is prevalent among marine clades of all different dispersal abilities and life histories, ranging from fishes, to corals, to gastropods and to ostracods[31,33,34], which potentially highlights the importance of ecological factors in marine diversification. Collectively, our MBD analysis partly supports the hopping hotspots model that proposes tectonic activity as the principal forcer of biodiversity hotspot by creating larger and more complex shallow-marine habitats[1,9]. We instead suggest that suitable habitat in terms of size but not complexity is important for the generation of enormous IAA diversity throughout the Cenozoic. However, note that the habitat estimations available now are based on global palaeogeographic reconstructions, so some uncertainty remains about the accuracy of such an inference. Other than the habitat factors, global temperature and sea level do not significantly correlate with diversity dynamics in the entire time frame analysis (*G* parameters overlapping with zero; Fig. 2a and Supplementary Table 2). This seemingly indicates that not all tectonic, eustatic, climatic, oceanographic

and geomorphological processes[1] are as critical as previously thought in fostering a biodiversity hotspot, or their short-term effects may be masked across a long-time frame (see the next section).

After establishing the general long-term drivers of the IAA's diversification dynamics, we then scrutinized how their effects may vary over time across different climate regimes[35]. A time-stratified MBD analysis revealed further details for the warm phase (23.04–13.9 Ma), cooling phase (13.9–5.33 Ma) and cold phase (5.33–0 Ma) of the Neogene–Quaternary interval (Fig. 2b–d and Supplementary Tables 3–5). The results for the cold phase are highly concordant with those for the entire time frame, with diversity ($G = -3.341$, $\omega = 0.912$) and habitat size ($G = 2.445 \times 10^{-7}$, $\omega = 0.702$) negatively and positively affecting speciation, respectively (Fig. 2d and Supplementary Table 5). Higher sea level also expedited speciation during this interval ($G = 0.022$, $\omega = 0.819$), probably through the species–area relationship. Large and frequent fluctuations in sea level associated with the glacial cycles may strongly affect the expansion and contraction of epicontinental seas and consequently the size of shallow-marine habitats[36,37]. There is no correlation between diversification dynamics and any biotic or abiotic factors for the cooling phase (Fig. 2c and Supplementary Table 4), which may indicate it as a transitional stage between two opposite climate regimes. Indeed, during the warm phase, biotic and abiotic controls on the diversification of IAA faunas occurred in very different ways from those of the cold phase (Fig. 2b and Supplementary Table 3). The strong positive effect of diversity on extinction conforms to our understanding of diversity dependency ($G = 336.168$, $\omega = 1$), but its positive effect on speciation ($G = 25.093$, $\omega = 0.999$) may be explained

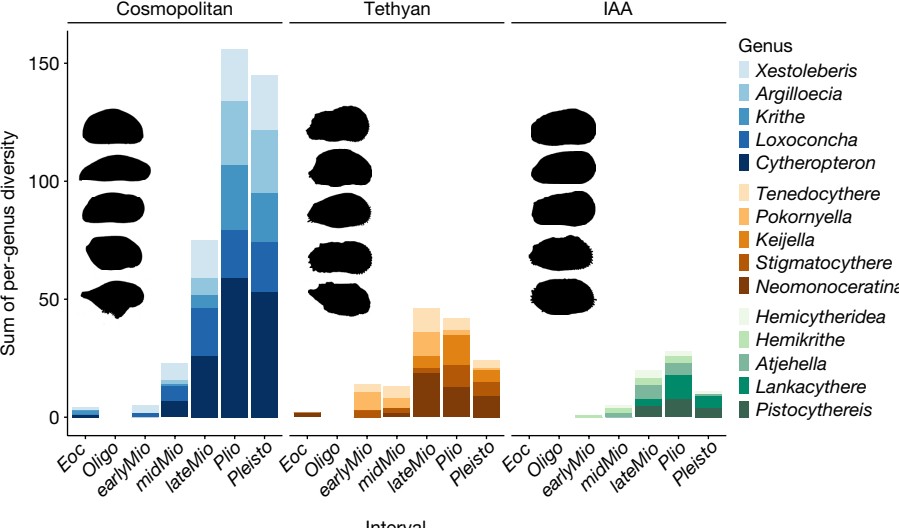

**Fig. 3 | Diversification of key biogeographic groups.** Stack bars show the sum of per-genus diversity of the five most diverse cosmopolitan, Tethyan and IAA genera during each interval of the Cenozoic. Eoc, Eocene; Oligo, Oligocene; earlyMio, early Miocene; midMio, middle Miocene; lateMio, late Miocene; Plio, Pliocene; Pleisto, Pleistocene.

by higher availability of empty niches when there were far fewer species, as the IAA hotspot just originated in the early Miocene. Higher sea level reduced extinction as expected ($G = -0.843$, $\omega = 0.999$), but it also reduced speciation, albeit more weakly ($G = -0.069$, $\omega = 0.995$). A possible explanation is that Southeast Asia was still separated from the Australian and Pacific plates by deep oceans[21,38], so high sea level may not contribute to form large areas of shelf habitats in the IAA. Notably, high temperature triggered both speciation ($G = 0.584$, $\omega = 0.993$) and extinction ($G = 5.038$, $\omega = 0.999$) during the warm phase, which may be due to the coexistence of warm-adapted and cold-adapted species. As Earth's climate showed a long-term cooling trend across the Cenozoic[39], some warm-adapted species that were resilient to or even preferred very high temperatures of the Eocene might not yet be extinct in the relatively warm early–middle Miocene[40,41]. They may have responded to temperature positively with more active proliferation. Cold-adapted species, by contrast, gradually rose throughout the Neogene–Quaternary as the climate further cooled[4,40,42], but their vulnerability to high temperature may put them at a much higher risk of extinction during the relative warm early–middle Miocene. Indeed, recent studies suggest that tropical temperatures >25 °C may be too elevated for marine organisms owing to metabolic trade-offs driven by temperature effects on hypoxia sensitivity[43–45]. Consistent evidence for thermal stress on tropical biodiversity comes from various marine taxa (for example, coral, reef fish, reptiles and calcareous algae) that declined in diversity in the tropics and exhibited poleward range shifts during historical warm intervals[45–47]. As temperature has a much stronger impact on extinction ($G = 5.04$) than on speciation ($G = 0.58$), our analysis suggests that too high a tropical temperature in warm climates slows down diversification and hinders the growth of the hotspot. The effects of temperature are not significant for other time windows, suggesting that thermal stress may be relieved in cooler climates. We need to be aware of future extinction risks and tropical biodiversity loss exacerbated by anthropogenic warming under high greenhouse gas emission scenarios, as tropical biotas are already living close to their ecophysiological thresholds[44,45].

## Imprints of key biogeographic events

We have shown that the long-term diversification of the IAA was influenced by a time-sensitive series of biotic and abiotic factors. We suggest that critical shifts in tectonics, climate and oceanography may be responsible for short-term events of speciation and extinction, but

also drove the biogeographic evolution of the IAA hotspot (Extended Data Fig. 5). Diversification initiated in the IAA with concurrent speciation and extinction peaks at 25 Ma, when Australia first collided with Southeast Asia[21,29]. The second and the largest peak in both speciation and extinction occurred at 20 Ma. It correlated in time with two major tectonic events outside and within the IAA, namely the closure of the Tethys Seaway separating the Indo-Pacific from the Mediterranean Sea and Atlantic Ocean[9,48], and the closure of the Indonesian deep-water passage between Southeast Asia and Australia[21]. The first event enabled the delineation of the Indo-Pacific biogeographic province through the breakup of the formerly global tropical sea belt (that is, the Tethys Sea), which could naturally spur both speciation and extinction through large-scale vicariance. Also, the loss of western connectivity placed the IAA in the geographic centre of vast Indo-Pacific oceans, where the overlapping of species distributional ranges could translate to a centre of diversity (that is, mid domain effect)[49]. The second event established a shallow-marine connection between Southeast Asia and Australia for the first time and thus assembled a large expanse of shelf seas as suitable habitat for diversification. It could have also facilitated the convergence of peripheral faunas in the central IAA, leading to an increase in regional diversity and changes in biogeographic patterns. Then, the 16-Ma peak coincided with the Mid-Miocene Climatic Optimum[40], which is in line with our MBD results indicating that both speciation and extinction rates accelerated with high temperature in a warm climate state (Fig. 2b). Similar diversification peaks at about 18 and 15 Ma were also found in Caribbean bryozoans[8], which may correspond to our 20-Ma and 16-Ma events. The 12-Ma peak was apparent only in the speciation rate when the East Asian monsoon intensified[50]. A stronger regional environmental gradient, such as salinity through the enhanced monsoon and resulting precipitation, could provide a possible explanation. Finally, the 5-Ma speciation and extinction peaks may be related to the restriction of the Indonesian Throughflow at the Miocene–Pliocene boundary[38,51], which allowed reversed faunal dispersal from the Indian Ocean to the Pacific Ocean[3]. In summary, abrupt transitions in Earth's physical environments probably introduced short-term instability into the biotic system. For each of the biotic events, extinctions were overcompensated with much stronger speciation to support overall diversification, indicating the role of the IAA as a reservoir for species accumulation and a source of species origination.

Among all palaeoenvironmental events discussed above, the restriction and final closure of the Tethys Seaway during about 25–20 Ma is

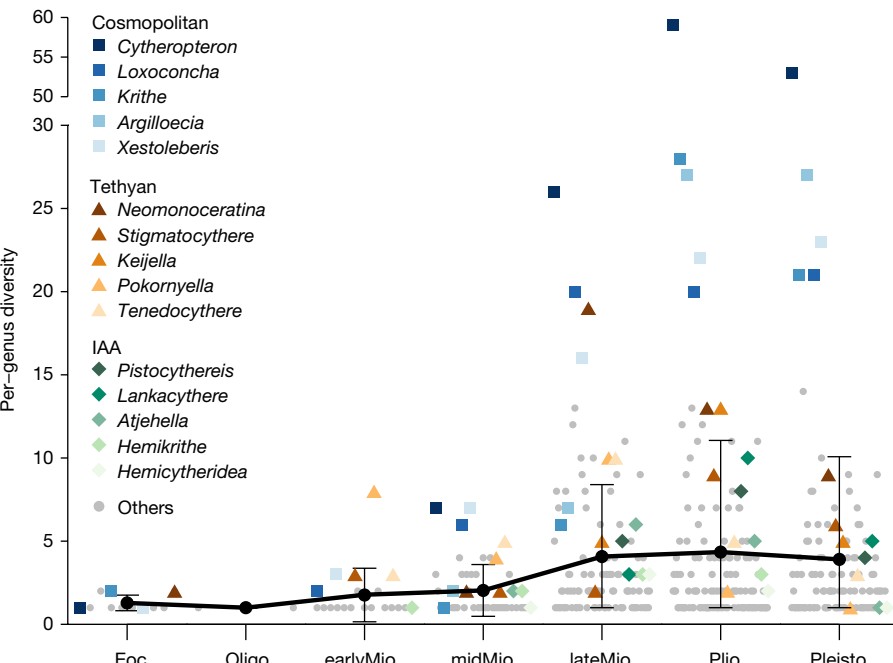

**Fig. 4 | Per-genus diversity of all ostracod genera during each interval of the Cenozoic.** The five most diverse genera of the cosmopolitan, Tethyan and IAA groups are shown in the corresponding colour used in Fig. 3. The error bar shows standard deviation (if the lower limit is below zero, the minimum value is used instead), with the black circles indicating mean per-genus diversity of all ostracod genera.

of particular interest. This process governed the palaeobiogeographic shifts from the vanishing Tethys to the emerging IAA, which are considered an important factor in shaping the latter hotspot by the hopping hotspots model[9]. Here we investigated how the migration of the Tethyan relicts may leave long-lasting implications on the diversity and biogeographic history of the IAA hotspot. We focused on the five most diverse genera of the Tethyan group (originated in the broad Tethyan region during the Palaeogene[52,53]), cosmopolitan group (exhibited global distributions during the Cenozoic[54,55]) and IAA group (originated in the central Indo-Pacific region during the Neogene[54–56]), respectively, and compared their proliferation across the Cenozoic (Fig. 3 and Extended Data Fig. 6). From the Eocene to early Miocene, the Tethyan genera exhibited a strong increase in diversity (2 to 14 species) and gained dominance over the other groups. During the middle–late Miocene, all three groups showed substantial proliferation with the cosmopolitan genera being more diverse than the Tethyan and IAA genera (75 compared with 46 and 20, respectively). The long-term trend of diversification proceeded to the Pliocene for the cosmopolitan and IAA groups with remarkable increases in diversity (156 and 28, respectively), whereas that of the Tethyan genera declined (42 species). Finally, slight Pleistocene drops in diversity probably represented sampling artefacts instead of true extinction, at least for the cosmopolitan and IAA groups, and the cessation of diversification corresponded to the maturation of the IAA hotspot. The evolution of three biogeographic groups closely mirrors that of the whole ostracod assemblage, as the average per-genus diversity reached a value of 1.29, 2.62 and 4.34 in the Eocene, Miocene and Pliocene, despite a slight decrease to 3.9 in the Pleistocene, indicating a pronounced diversification at species level throughout the Cenozoic (Fig. 4). The waxing and waning of these three groups clearly reflect a changing biogeographic affinity of the IAA with other regions. Before the final closure of the Tethys Seaway, the Tethyan ancestors were able to reach the IAA after a long-distance dispersal. They persisted and proliferated in the IAA even after the seaway closed, and thereby supported the early-stage development of the IAA hotspot. Since the middle Miocene, the IAA situated at the centre of the vast Indo-Pacific domain with high connectivity and dispersal may be more prone to accommodate markedly radiating cosmopolitan taxa from the Indian and Pacific

oceans, which then outcompeted the earlier occupants of the Tethyan genera. In addition, the endemic IAA genera were comparatively less diverse, possibly because of their youngest age; nevertheless, they also experienced a prominent diversification since the late Miocene. Their success may indicate the importance of tropical hotspots in generating and exporting biodiversity within a relatively short geological time. We conclude that the diversification of the IAA hotspot was accompanied by profound biogeographic changes, as the Tethyan taxa were gradually replaced by cosmopolitan and IAA taxa during the Neogene–Quaternary. Tethyan faunal elements from older, vanished hotspots did become evolutionary fuel for the emergence of the IAA, as proposed by the hopping hotspots model, but such a unique palaeobiogeographic shift was not the sole mechanism that contributed to tropical diversification. In fact, the stepwise collision of Australia with Southeast Asia and the switching on and off of the Indonesian Throughflow could both be important processes that facilitated the convergence, accumulation and eventually proliferation of taxa in the central IAA.

Another possible ecological explanation for the success of cosmopolitan and IAA genera is their adaptation to the glacio-eustatic sea-level fluctuations of the Plio-Pleistocene. During this interval, ostracod taxa with strong depth-preference conservatism may have been forced to repeatedly migrate downslope and upslope to follow successive cycles of large-amplitude sea-level changes[13]. Most Tethyan genera were found exclusively in shallow-marine environments and had relatively narrow depth niches[52,53]. They may be less adaptable to large-scale sea-level changes than some deeper-water cosmopolitan genera (for example, *Cytheropteron*, *Krithe* and *Argilloecia*), which thrived in a wider range of depths from the continental shelf to the continental slope[13]. To sum up, the Plio-Pleistocene decline of Tethyan genera is probably due to a combination of long-term palaeobiogeographic reorganization and short-term palaeoceanographic changes.

Our results suggest that the increase in habitat size and immunity to major extinction events during the Cenozoic allowed for strong, uninterrupted long-term diversification that reached an asymptote in the IAA, making it the current global centre of biodiversity. The relief of thermal stress as Earth's climate transformed from a warm to a cold state was also essential for this process. It took tens of millions of years

for the IAA hotspot to fully develop in an ideal environment in terms of climate, habitat and biogeographic connectivity, without, by chance, a major extinction event. As we know, the opposite is now happening, with anthropogenic warming[57], habitat destruction[58] and the sixth mass extinction[44,59]. All may effectively undermine the IAA hotspot and impair its function as a cradle and museum of tropical biodiversity. Our palaeobiological results highlight the slow evolution, in contrast to the ongoing decadal degradation, of the tropical hotspot as one of the most remarkable biodiversity and biogeographic patterns, and urge conservation efforts.

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

## Methods

We first summarize how we obtained the Cenozoic IAA fossil dataset, integrating our original data with published data. We then detail the application of birth–death models to reconstruct the IAA's diversity history, and to uncover its controlling factors. Finally, we describe the raw analyses of the fossil dataset from a biogeographic perspective.

### Sediment samples, sample processing and ostracod identification

A total of 188 Neogene and Quaternary outcrop sediment samples were collected from the islands of Luzon, Cebu, Negros, Panay, Leyte, Bohol, Mindanao and Samar, Philippines, in addition to the islands of Java and Kalimantan, Indonesia (Extended Data Fig. 1). The ages of all sediment samples were determined from planktic foraminiferal and calcareous nannofossil biochronology following the geological timescale in refs. 60,61 and Southeast Asia stratigraphy[62]. We used a standard sample processing method for microfossil ostracod research; that is, outcrop samples were wet-sieved through a 63-μm mesh sieve, oven dried and then dry-sieved through a 150-μm mesh sieve[13]. We retained the >150 μm size fraction, from which adults and late moulting juveniles of ostracods were picked. Specimens <150 μm are usually early moulting juveniles for most genera, which are too delicate and too small to be preserved and identified. Therefore, the effect of mesh size on species composition and diversity measurements is likely to be negligible[13]. Samples with too many ostracod specimens were divided using a sample splitter. All ostracods in a sample or a split aliquot were picked up and sorted on a micropalaeontological cardboard slide. Most ostracods were identified to the species level. Taxon counts considered one valve or articulated carapace as one specimen, which is the standard method for counting ostracods[13].

### Data integration

We integrated published Cenozoic IAA data into this study after rigorous taxonomic harmonization, including data from Leyte Island, Philippines, in addition to the islands of Java and Kalimantan, Indonesia (Extended Data Fig. 1)[13,63]. Most of the original and published samples are in excellent ostracod preservation, with a few middle-Miocene samples in moderate preservation. Even Eocene specimens have glassy, translucent shells, and thus it is unlikely that preservation bias substantially affects our biodiversity results, as discussed in detail previously[13]. The depositional environments of the published sites vary from the continental shelf to the upper continental slope, all under fully marine conditions[13,63]. Consistently, the fossil ostracod faunas of our original IAA samples show the same diversity of depth habitats, ranging from shallow marine to the shallowest deep sea (upper bathyal). With the compilation of published sources with original data, this study addresses biotic changes in primarily continental shelf benthic ecosystems across the Cenozoic. Spatial coverage of all aggregated samples is reasonably good for representing the IAA region as a whole, especially within the Philippines and central Indonesia where modern diversity is known to be the highest[10,22] (Extended Data Fig. 4).

### Estimating speciation and extinction dynamics

The application of the birth–death model deciphered the macroevolutionary dynamics of the IAA hotspot by estimating historical variations in speciation and extinction rates across the Cenozoic while incorporating sampling biases and uncertainties associated with the age of each fossil occurrence. We carried out analyses of the raw ostracod fossil dataset based on the Bayesian framework as implemented in the PyRate 3.0 program[64,65]. The robustness of PyRate was rigorously tested against a range of potential biases, including violations of the sampling assumptions, variable preservation rates and incomplete taxon sampling, and it has been shown to correctly recover the dynamics of speciation and extinction in the face of sudden rate changes and mass extinction[66]. We analysed the datasets under time-varying birth–death models to simultaneously estimate for each clade: the parameters of the preservation process; the times of speciation (Ts) and extinction (Te) of each ostracod species; and the speciation ($\lambda$) and extinction ($\mu$) rates and their variation through time. The PyRate approach benefits from the estimation of Ts and Te of the studied species taking into account preservation biases, and thus provides estimates of species longevity (that is, the duration or time span of a species). This reduces the issues associated with estimating speciation and extinction rates and estimating the number of species through time[64,65]. Based on all taxon occurrences, the preservation rate is expressed as expected occurrences per taxon per million years (Myr) and contributes to the inference of the individual origination and extinction times of each taxon. We first determined the best-fit preservation process for the data using the -PPmodeltest option, which compares the homogeneous Poisson process, the non-homogeneous Poisson process and the time-variable Poisson process. We carried out the recently enhanced version through the reversible-jump Markov chain Monte Carlo (MCMC) algorithm (-A 4 option), which gives more accurate estimates of speciation and extinction rates compared with traditional approaches including boundary-crossers and three-timers[66].

We ran PyRate for 50 million MCMC generations for the birth–death with constrained shifts (BDCS) model[67]. Given the high resolution of the dataset, we defined time bins of 2 Myr (-fixShift option) across the Cenozoic. All analyses were set with a non-homogeneous Poisson process of preservation (determined as the best model), and accounted for varying preservation rates across taxa using the Gamma model (-mG option); that is, with Γ-distributed rate heterogeneity. We monitored chain mixing and effective sample sizes by examining the log files in Tracer 1.7.1[68] after excluding the first 10% of the samples as a burn-in period. We then combined the posterior estimates of the speciation and extinction rates across all replicates to generate rates-through-time plots (speciation, extinction and net diversification). Rates of two adjacent intervals (2 Myr) were considered significantly different when the mean of one lay outside the 95% credibility intervals of the other, and vice versa. We replicated the analyses on ten randomized datasets and calculated estimates of times of speciation and times of extinction (Ts–Te data) as the mean of the posterior samples from each replicate. Thus, we obtained ten posterior estimates of the times of speciation and extinction for all species and used them as input data in all of the subsequent analyses (see below), which therefore focused exclusively on the estimation of the birth–death parameters (that is, without remodelling preservation and re-estimating times of speciation and extinction). This procedure markedly reduces the computational burden, while allowing us to account for the preservation process and the uncertainties associated with the fossil ages.

### Estimating palaeoenvironment-dependent diversification

We then used the estimated times of speciation and extinction (Ts–Te data) of all species to identify putative causal mechanisms for the diversification of shallow-marine taxa in the IAA hotspot. We simultaneously examined the correlation of a set of palaeoenvironmental variables with speciation and extinction rates throughout the Cenozoic. We focused on temperature, sea level, habitat size and complexity (representing the effects of tectonic movements) as four abiotic factors and diversity dependency as one biotic factor, all of which have been linked to biodiversity changes in marine invertebrates and have been suggested to play an important role in the development of hotspots[2,3,36,69,70]. Trends in global temperature change are typically based on oxygen isotope ratios ($\delta^{18}O$) in benthic foraminifera and a smoothed Cenozoic sea surface temperature record is derived from ref. 71. Similarly, fluctuations in the Cenozoic global mean sea level are derived from benthic foraminiferal $\delta^{18}O$ and Mg/Ca records that document the evolution from the Eocene ice-free conditions to Quaternary bipolar ice sheets causing eustatic variations[72]. Regional changes in

habitat size and complexity are measured as total shelf area and shoreline length, respectively, using palaeogeographic reconstructions of palaeo-coastlines and flooded continental shelf distributions in the IAA for the past 50 Myr[3,73]. Shelf area is delimited between the reconstructed maximum transgression coastline and continental margin, and shoreline length is calculated using the 'yardstick' method[73,74].

After analysing the long-term drivers of the diversification dynamics, we then estimated whether palaeoenvironmental changes may have varying effects through time in the IAA. As there was differential diversification dynamics over time, it is likely that palaeoenvironmental changes had different impacts under different climatic regimes. To study such effects, we carried out a time-stratified MBD analyses assuming different diversification dynamics and correlations to each studied factor during the warm phase (23.04–13.9 Ma), cooling phase (13.9–5.33 Ma) and cold phase (5.33–0 Ma) of the Neogene–Quaternary interval, as segmented by the Middle Miocene Climate Transition and the Pliocene–Pleistocene transition[39]. Such a division scheme reflects the global climate evolution across the Neogene–Quaternary.

We used the MBD model to assess to what extent biotic and abiotic factors can explain temporal variation in speciation and extinction rates[75]. Under the MBD model, speciation and extinction rates can change through time but equally across all lineages, as in the BDCS model, through correlations with multiple time-continuous variables, and the strengths and signs (positive or negative) of the correlations are jointly estimated for each variable. We applied exponential correlations between rates and environmental variables. The MBD model also makes the hypothesis that such correlations are time continuous[66,76]. The correlation parameters ($G$) can take negative values indicating negative correlation, or positive values for positive correlations. When their value is estimated to be approximately zero, there is no correlation. A MCMC algorithm jointly estimates the baseline speciation ($\lambda 0$) and extinction ($\mu 0$) rates and all correlation parameters ($G\lambda$ and $G\mu$) using a horseshoe prior to control for over-parameterization and for the potential effects of multiple testing[75]. The horseshoe prior provides an approach to distinguish correlation parameters that should be treated as noise (and therefore are around 0) from those that are significantly different from 0 and represent true signal.

We ran the MBD model using 40 million MCMC iterations and sampling every 40,000 to approximate the posterior distribution of all parameters ($\lambda 0$, $\mu 0$, five $G\lambda$, five $G\mu$ and the shrinkage weights of each correlation parameter, $\omega G$). We summarized the results of the MBD analyses by calculating the posterior mean and 95% credibility interval of all correlation parameters and the mean of the respective shrinkage weights ($\omega$) across ten replicates, as well as the mean and 95% credibility interval of the baseline speciation and extinction rates. A given environmental variable must have $\omega > 0.5$ and 95% credibility interval for $G \neq 0$ to be considered as having a significant effect on rate variations. Regarding the biological meaning of the parameters estimated with temperature, for instance, the estimation of a positive $G\lambda$ would indicate that higher temperatures increase speciation rates, whereas a negative $G\lambda$ would indicate that higher temperatures decrease speciation rates. The same rationale applies to the extinction rate but with the parameter $G\mu$ quantifying the correlation between changes in extinction rates and this variable. Let us imagine the positive effect of temperature on speciation given by $G\lambda = 0.05$. This result means that speciation and temperature correlate positively such that speciation increased by 5% as global temperatures increased at every time step (here, every 0.1 million years).

## Sensitivity tests

Regarding the estimation of the speciation and extinction dynamics, we noted that potential sampling bias could be caused by the lack of the Oligocene data and the binning of the fossil record. To explore the sensitivity of the speciation and extinction estimates, we repeated all of the BDCS analyses for the Neogene–Quaternary alone with time

bins of 2 Myr, for the entire Cenozoic with time bins of 5 Myr and for the Neogene–Quaternary alone with time bins of 5 Myr. All of these analyses were parameterized following the main analysis. All results corroborate a reasonably robust and unambiguous pattern of strong and continuous diversification throughout the studied intervals (Extended Data Figs. 7–9).

For disentangling the palaeoenvironmental controls on diversification dynamics, we reasoned that the use of global temperature in a regional analysis of the IAA could pose certain limitations on our modelling, together with the Oligocene sampling gap. We ran two sensitivity MBD analyses using the tropical temperature record from ref. 77 for the Cenozoic and for the Neogene–Quaternary instead of the global temperature. With all other parameters unchanged, the results remain consistent with those from the main analysis, still showing that diversity dependence and habitat size are two determinants of speciation (Supplementary Tables 6 and 7).

## Raw analyses of the fossil dataset

We also carried out rarefaction, $E$(S200), the expected number of species if 200 specimens were sampled to calculate species α-diversity, taking into account the sample size artefact; thus, only samples with ≥200 individuals were used to calculate diversity. We illustrated the stratigraphic ranges for all ostracod taxa by connecting the earliest and latest occurrence of every taxon. In addition, we counted diversity per genus as the number of species in a genus and calculated the mean diversity per genus of all genera occurring within each geological interval. Among all genera that were relatively abundant and speciose throughout the Cenozoic in the IAA, we identified typical Tethyan genera that originated in the broad Tethys biogeographic area during the Palaeogene (that is, *Neomonoceratina* in the Palaeocene and *Stigmatocythere*, *Pokornyella*, *Keijella* and *Tenedocythere* in the Eocene)[52,53,78–80]; cosmopolitan genera that had a broad geographical distribution across the Cenozoic (*Cytheropteron*, *Krithe*, *Argilloecia*, *Loxoconcha* and *Xestoleberis*)[54,55]; and IAA genera that first appeared in the central Indo-Pacific during the Neogene (that is, *Pistocythereis*, *Lankacythere*, *Atjehella*, *Hemikrithe* and *Hemicytheridea* in the Miocene[54–56,81,82]). Finally, we compared the trends of their respective diversity changes per genus throughout the Cenozoic. All of the above analyses were carried out in the R programming language (version 4.3.0).

## Reporting summary

Further information on research design is available in the Nature Portfolio Reporting Summary linked to this article.

## Data availability

The data supporting the findings of this study are available via Figshare at https://doi.org/10.6084/m9.figshare.25395871.v2 (ref. 83). Source data are provided with this paper.

## Code availability

The codes supporting the findings of this study are available via Figshare at https://doi.org/10.6084/m9.figshare.25395871.v2 (ref. 83).

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

**Acknowledgements** We thank R. P. P. Wong and M. G. Y. Lo for continuous support; H. Hayashi for help with biostratigraphy; A. Zhao for help with sampling processing; J. Zhang for help with taxonomic identification; and L. Cotton, A. Lam and A. Woodhouse for comments. This work was partly supported by grants from the Research Grants Council of the Hong Kong Special Administrative Region, China (project codes: RFS2223-7S02, HKU 17300821, HKU 17300720, HKU 17302518, C7013-19G and G-HKU709/21), the Small Equipment Grant of the University of Hong Kong, the Seed Funding Programme for Basic Research of the University of Hong Kong (project codes: 202111159167, 202011159122 and 201811159076), the Faculty of Science RAE Improvement Fund of the University of Hong Kong, the Seed Funding of the HKU-TCL Joint Research Centre for Artificial Intelligence of the University of Hong Kong, and the State Key Laboratory of Marine Pollution Seed Collaborative Research Fund (SKLMP/SCRF/0031; to M.Y.). S.Y.T. was supported by the Humboldt Research Fellowship.

**Author contributions** S.Y.T. and M.Y. designed research; S.Y.T., M.Y., T.K., A.G.S.F., Y.M.A., H.P., T.I., H.I., C.P.S. and W.R. carried out research; S.Y.T., F.L.C., H.-H.M.H. and M.Y. analysed data; and S.Y.T., M.Y. and F.L.C. wrote the paper. S.Y.T. created the silhouettes of each ostracod genus in Fig. 3.

**Competing interests** The authors declare no competing interests.

**Additional information**
**Correspondence and requests for materials** should be addressed to Skye Yunshu Tian or Moriaki Yasuhara.

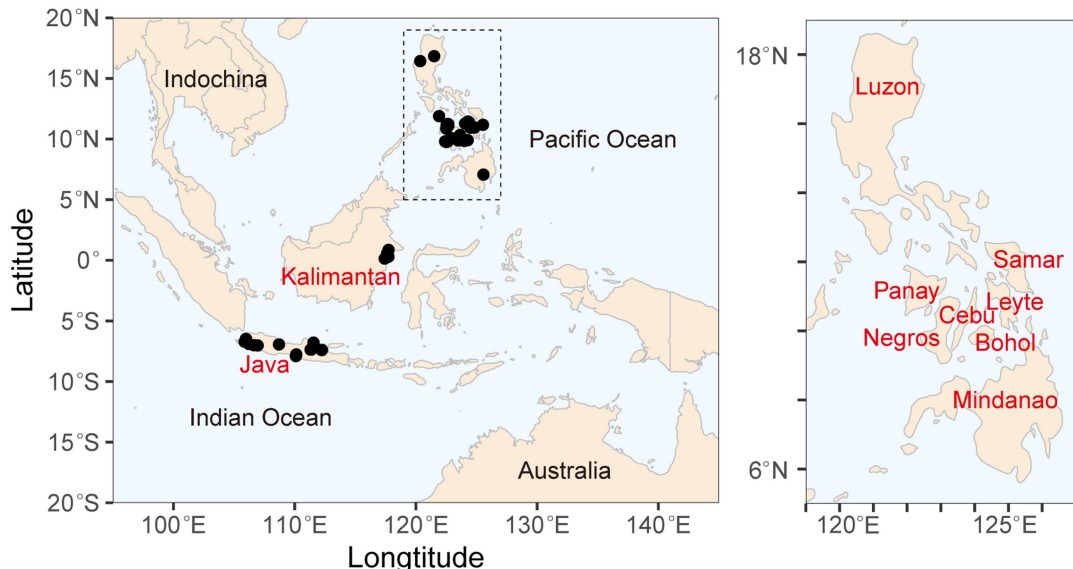

**Extended Data Fig. 1 | Map of the Indo-Australian Archipelago showing sampling location, with a magnification of the Philippine islands.** The name of each sampling island is labelled in red.

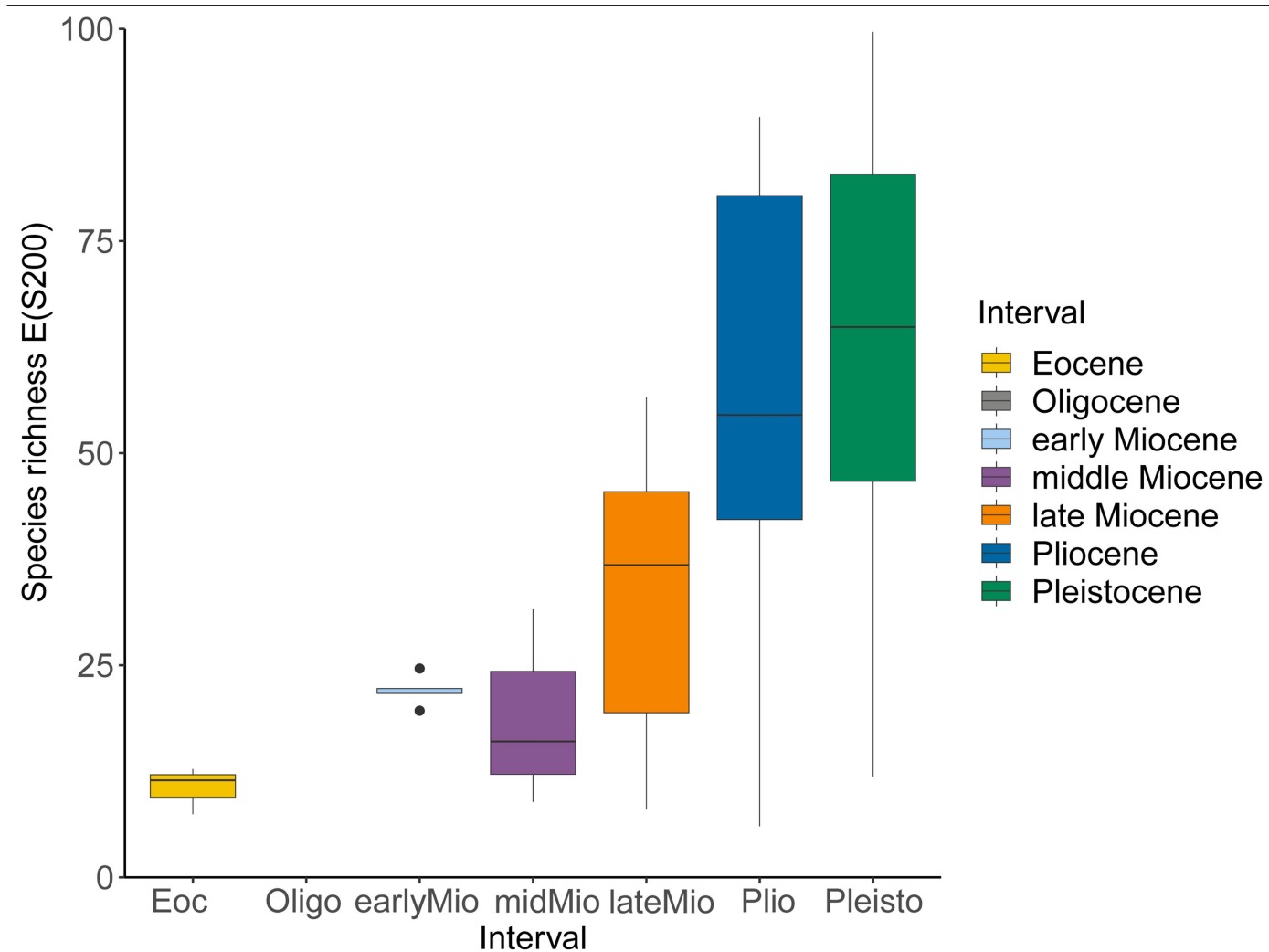

**Extended Data Fig. 2 | Ostracod diversity trajectory over the Cenozoic in the IAA, based on species richness from rarefaction E(S200) for sediment samples (n = 111).** Box hinges represent the first and third quartiles with the center as median, and whiskers extend to 1.5× the interquartile range (IQR) from the first and third quartiles. Minima and maxima beyond the whiskers are individually plotted outlying points. Eoc: Eocene; Oligo: Oligocene; earlyMio: early Miocene; midMio: middle Miocene; lateMio: late Miocene; Plio: Pliocene; Pleisto: Pleistocene.

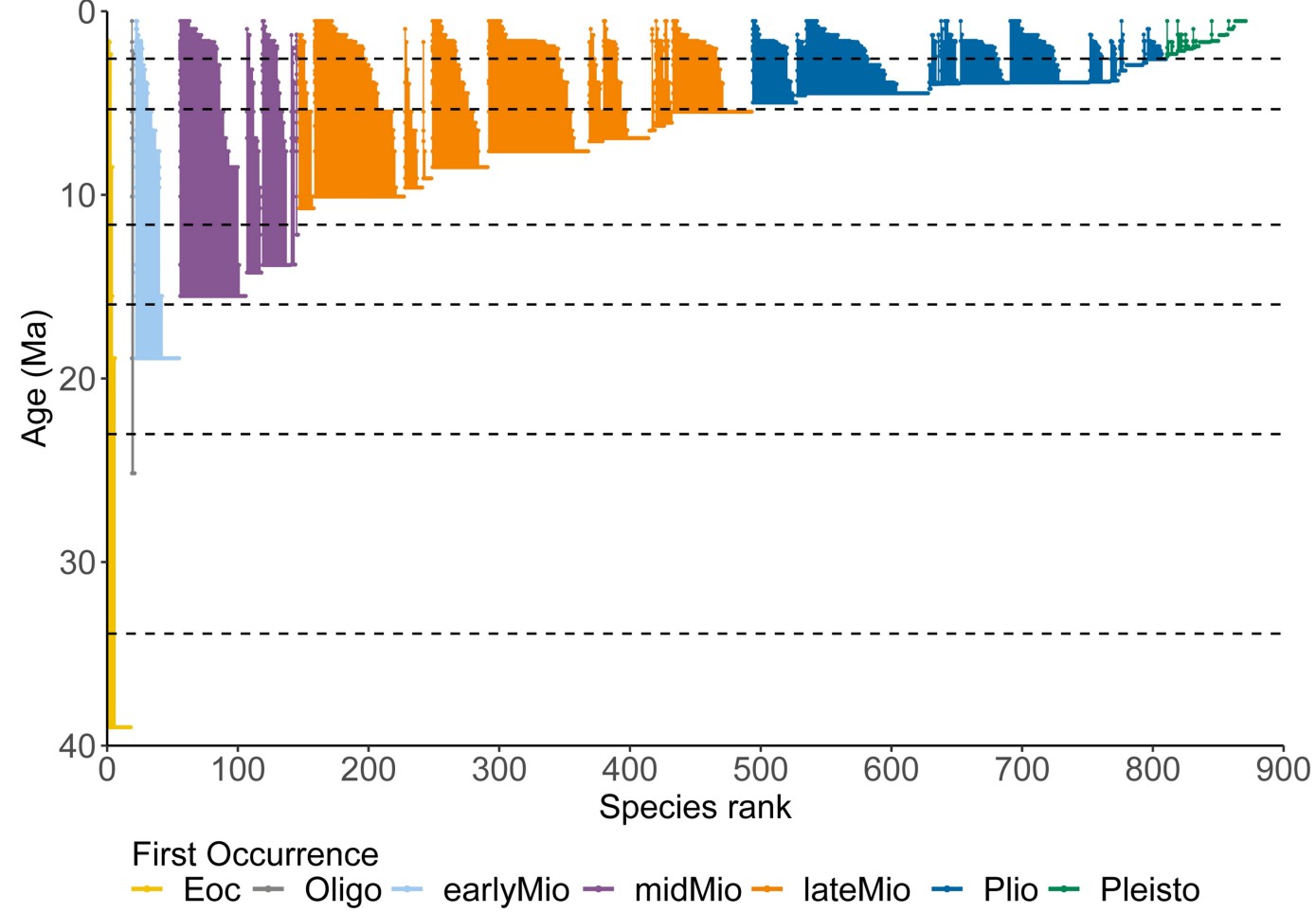

**Extended Data Fig. 3 | Stratigraphic ranges for all 874 ostracod taxa in the IAA in this study.** The ostracod taxa (780 species, and 94 genus level entries) are classified by their earliest stratigraphic occurrence, i.e., Eocene, Oligocene, early Miocene, middle Miocene, late Miocene, Pliocene, and Pleistocene taxa. The range for each taxon is computed by connecting the minimum and maximum ages of the taxon occurrence by a vertical line, and individual taxa occurrences are indicated by dots. Within each interval, the taxa are ranked by the age of first occurrence. Dashed horizontal lines indicate chronostratigraphic boundary. Abbreviations as in Extended Data Fig. 2.

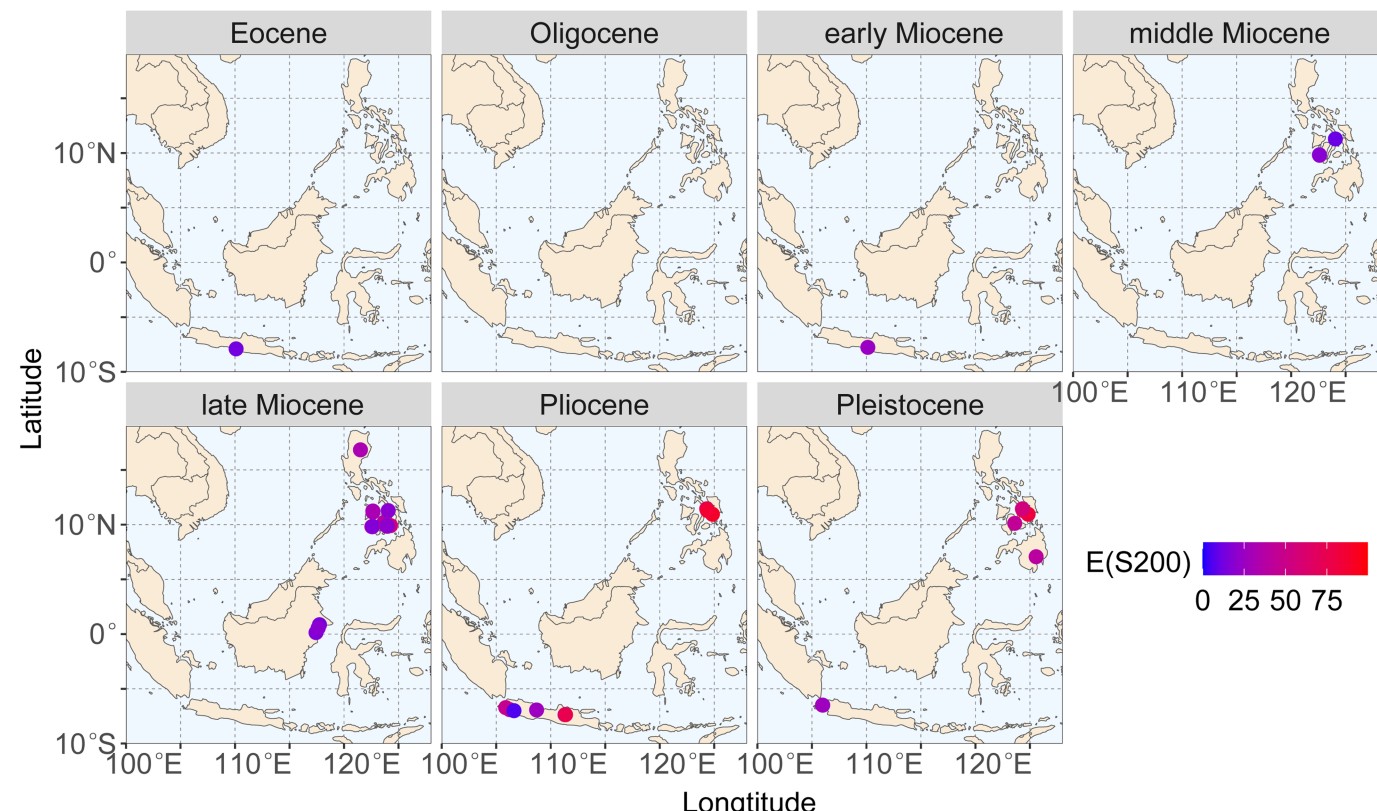

**Extended Data Fig. 4 | Diversity map showing species richness from rarefaction E(S200) in each geological interval.** Eocene: 56-33.9 Ma; Oligocene: 33.9-23.04 Ma; early Miocene: 23.04-15.99 Ma; middle Miocene: 15.99-11.65 Ma; late Miocene: 11.65-5.33 Ma; Pliocene: 5.33-2.58 Ma; Pleistocene: 2.58-0.01 Ma.

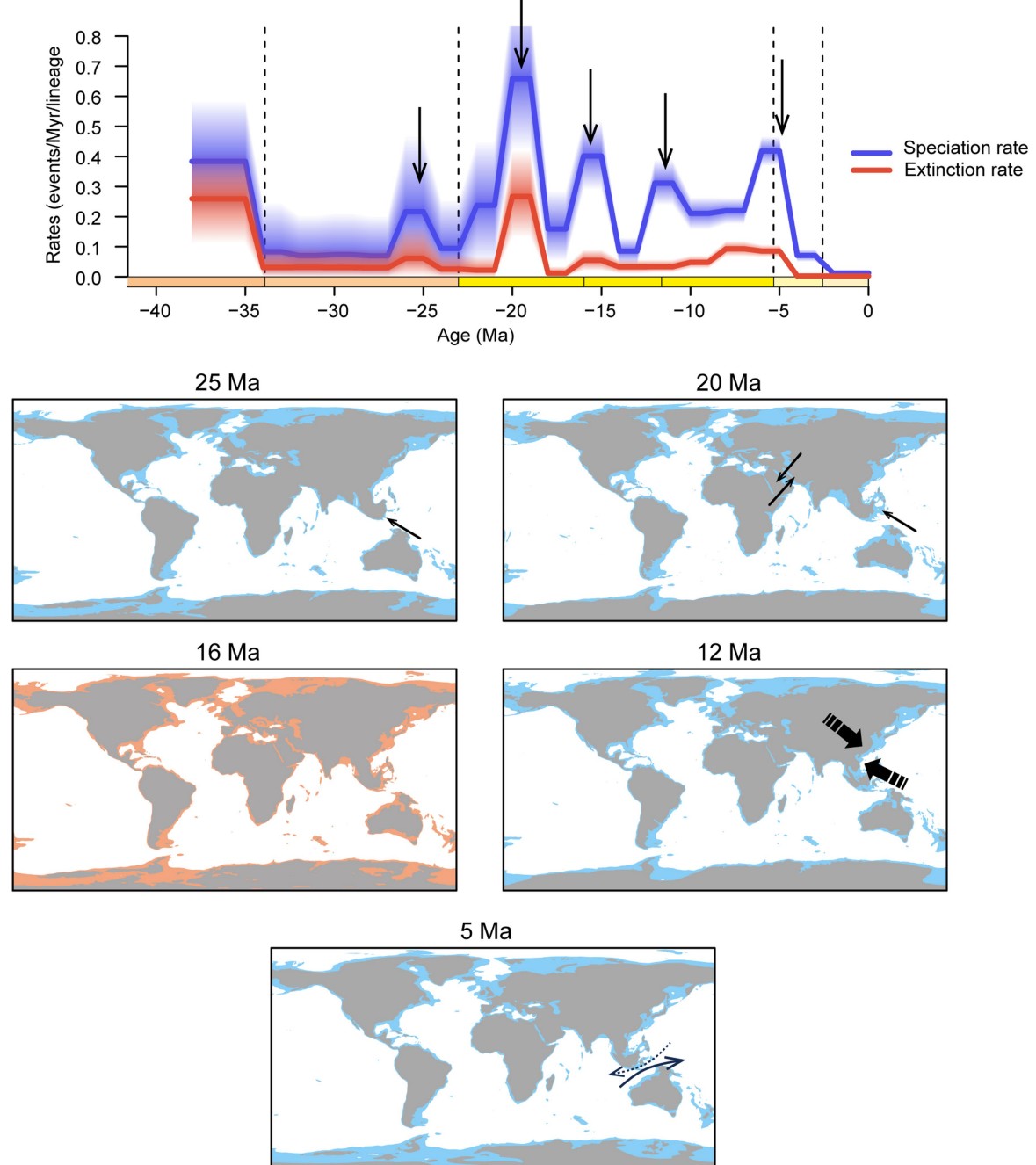

**Extended Data Fig. 5 | Schematic diagram showing the short-term biotic events in relation to key changes in tectonics, climate and oceanography.** 25 Ma: first collision of Australia with Southeast Asia; 20 Ma: closure of the Tethys Seaway and closure of the Indonesian deep-water passage; 16 Ma: Mid-Miocene Climate Optimum; 12 Ma: intensification of the East Asian Monsoon; 5 Ma: restriction of the Indonesian Throughflow. Blue and orange shaded areas in the maps indicate continental shelf. Paleogeographic maps modified from Kocsis and Scotese[73].

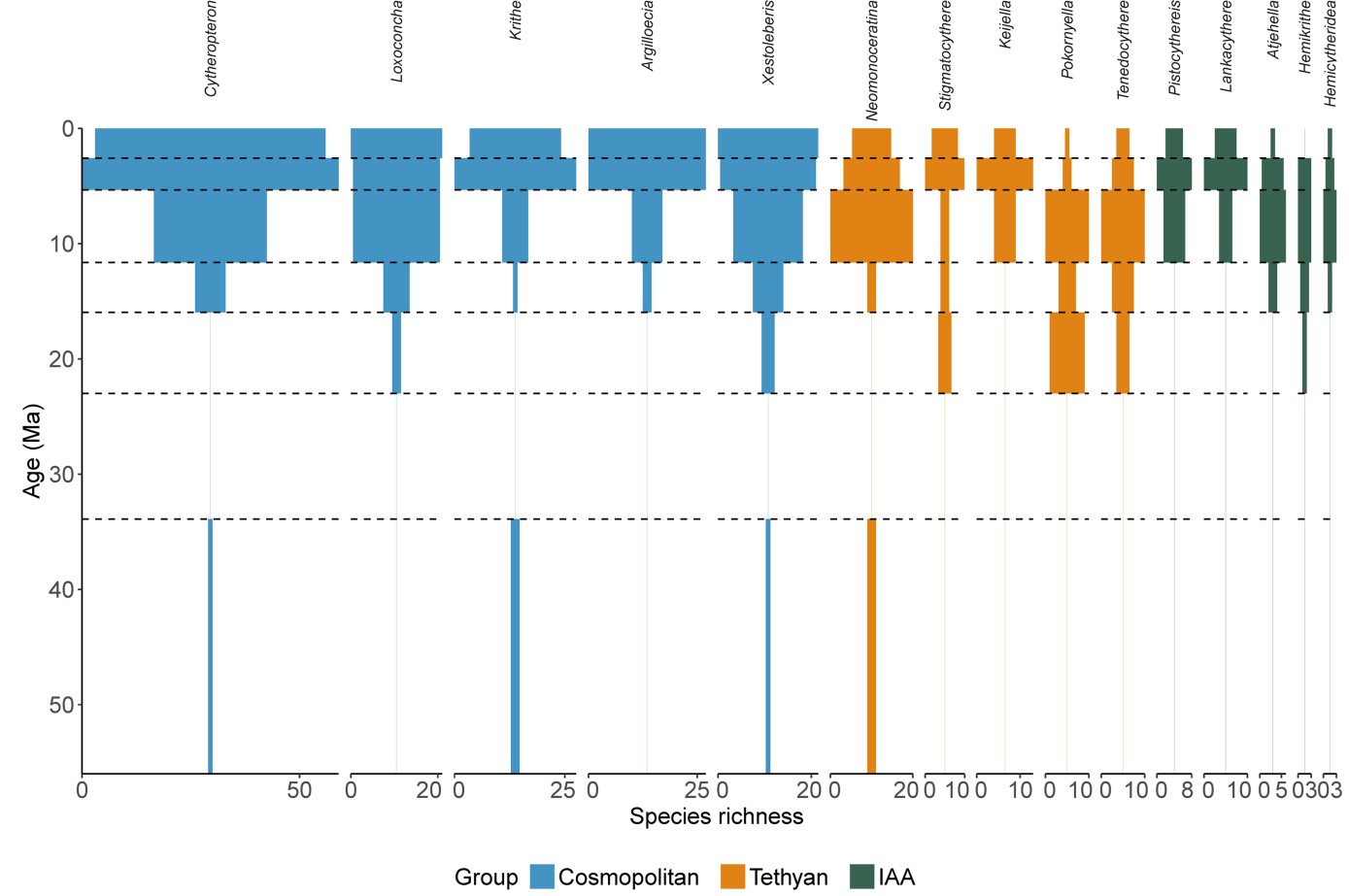

**Extended Data Fig. 6 | Species-level diversification of the five most diverse cosmopolitan, Tethyan, and IAA genera across the Cenozoic.** Dashed horizontal lines indicate chronostratigraphic boundary.

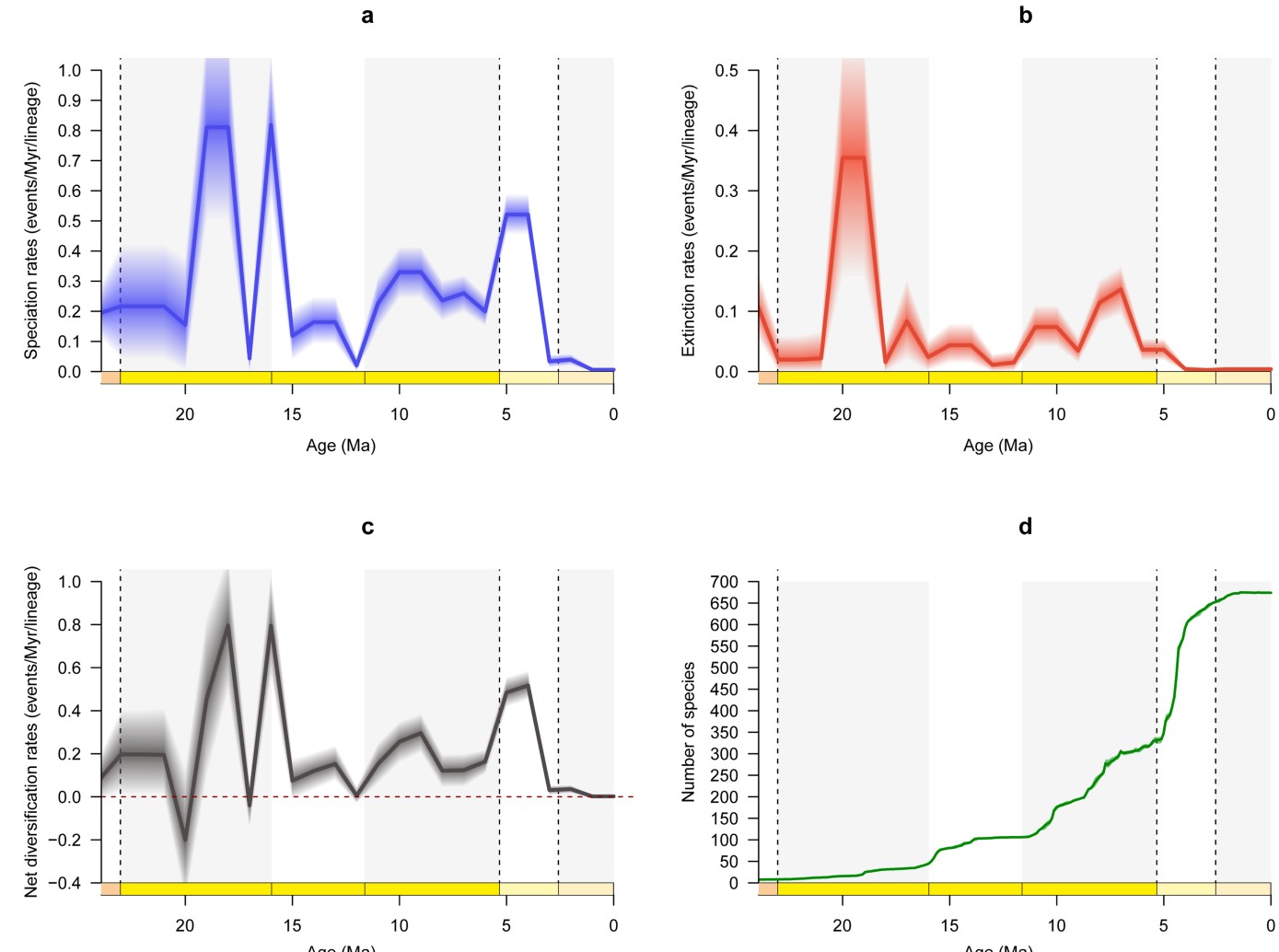

**Extended Data Fig. 7 | Estimation of regional diversity dynamics in the IAA across the Neogene-Quaternary with 2 Myr bin.** Bayesian inferences of (a) speciation rates per lineage, (b) extinction rates per lineage, (c) net diversification rates as the difference between speciation and extinction rates (rates below 0 indicate declining diversity), and (d) the changes in species richness. Solid lines indicate mean posterior rates, and the shaded areas show the 95% confidence interval. Early Miocene, late Miocene, Pleistocene are shaded. Vertical dotted lines indicate epoch boundaries.

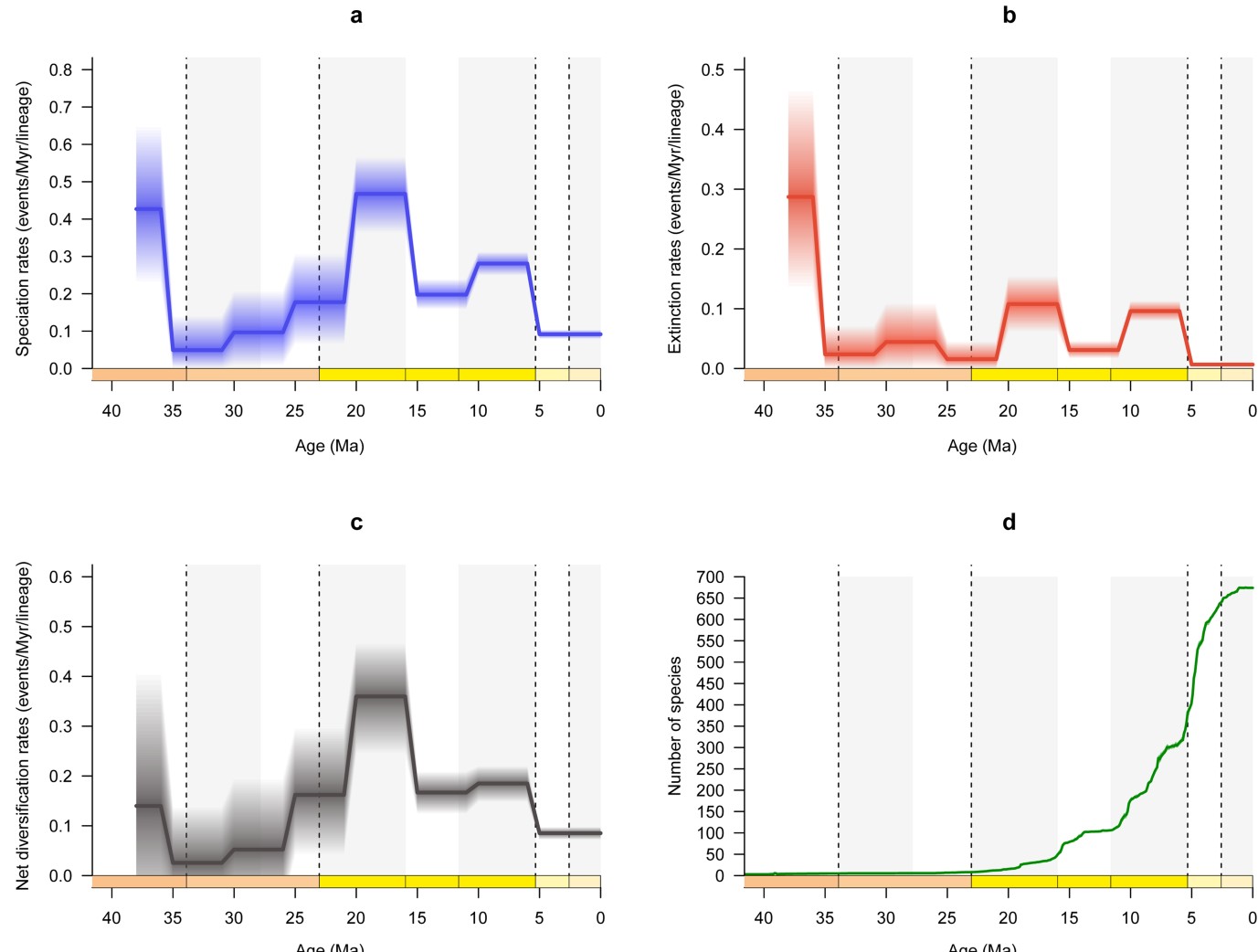

**Extended Data Fig. 8 | Estimation of regional diversity dynamics in the IAA across the Cenozoic with 5 Myr bin.** Bayesian inferences of (a) speciation rates per lineage, (b) extinction rates per lineage, (c) net diversification rates as the difference between speciation and extinction rates (rates below 0 indicate declining diversity), and (d) the changes in species richness. Solid lines indicate mean posterior rates, and the shaded areas show the 95% confidence interval. Early Oligocene, early Miocene, late Miocene, Pleistocene are shaded. Vertical dotted lines indicate epoch boundaries.

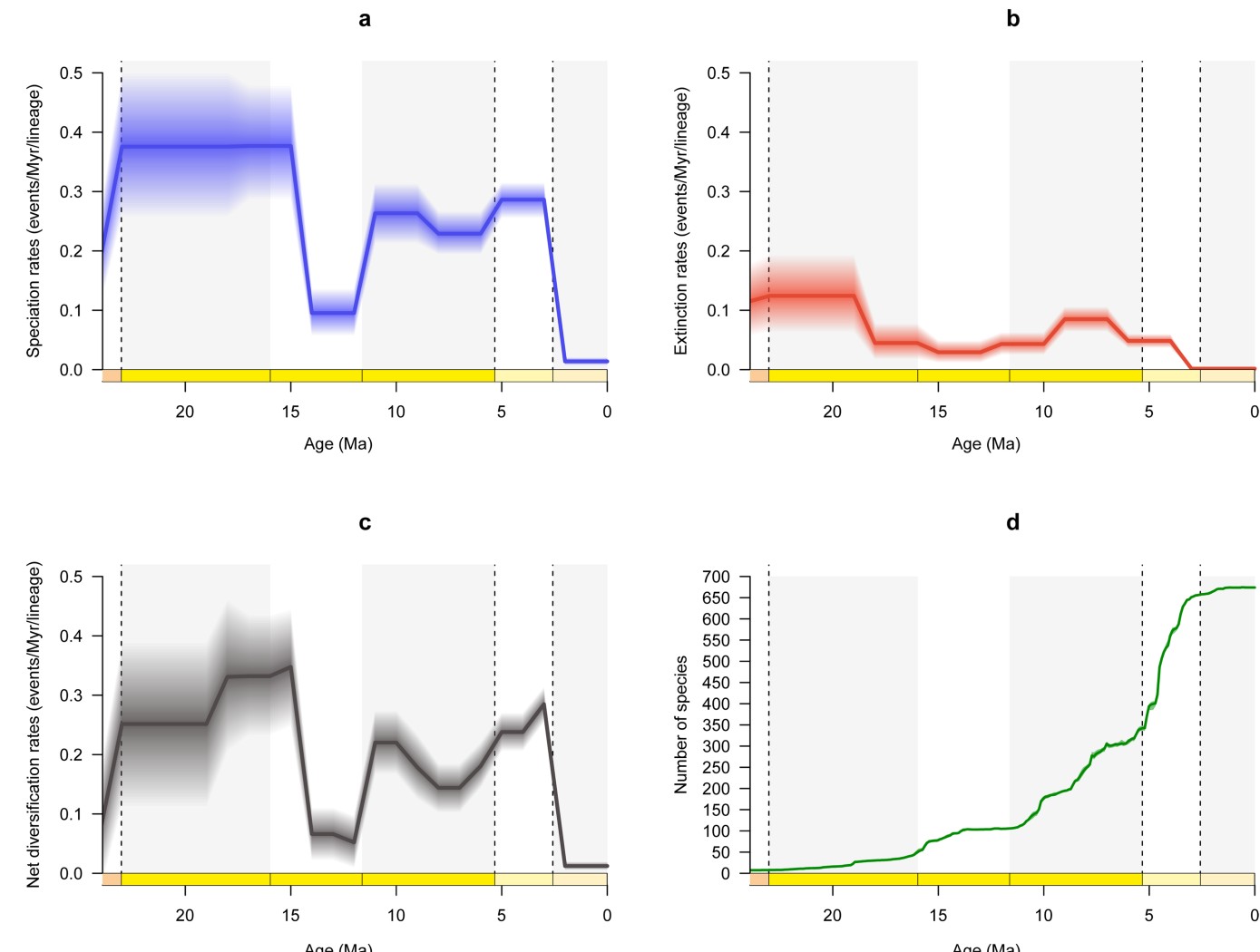

**Extended Data Fig. 9 | Estimation of regional diversity dynamics in the IAA across the Neogene-Quaternary with 5 Myr bin.** Bayesian inferences of (a) speciation rates per lineage, (b) extinction rates per lineage, (c) net diversification rates as the difference between speciation and extinction rates (rates below 0 indicate declining diversity), and (d) the changes in species richness. Solid lines indicate mean posterior rates, and the shaded areas show the 95% confidence interval. Early Miocene, late Miocene, Pleistocene are shaded. Vertical dotted lines indicate epoch boundaries.

# Reporting Summary

Please do not complete any field with "not applicable" or n/a. Refer to the help text for what text to use if an item is not relevant to your study.
For final submission: please carefully check your responses for accuracy; you will not be able to make changes later.

## Statistics

For all statistical analyses, confirm that the following items are present in the figure legend, table legend, main text, or Methods section.

| n/a | Confirmed | |
|---|---|---|
| ☐ | ☒ | The exact sample size (*n*) for each experimental group/condition, given as a discrete number and unit of measurement |
| ☐ | ☒ | A statement on whether measurements were taken from distinct samples or whether the same sample was measured repeatedly |
| ☒ | ☐ | The statistical test(s) used AND whether they are one- or two-sided *Only common tests should be described solely by name; describe more complex techniques in the Methods section.* |
| ☐ | ☒ | A description of all covariates tested |
| ☒ | ☐ | A description of any assumptions or corrections, such as tests of normality and adjustment for multiple comparisons |
| ☐ | ☒ | A full description of the statistical parameters including central tendency (e.g. means) or other basic estimates (e.g. regression coefficient) AND variation (e.g. standard deviation) or associated estimates of uncertainty (e.g. confidence intervals) |
| ☒ | ☐ | For null hypothesis testing, the test statistic (e.g. *F*, *t*, *r*) with confidence intervals, effect sizes, degrees of freedom and *P* value noted *Give P values as exact values whenever suitable.* |
| ☐ | ☒ | For Bayesian analysis, information on the choice of priors and Markov chain Monte Carlo settings |
| ☒ | ☐ | For hierarchical and complex designs, identification of the appropriate level for tests and full reporting of outcomes |
| ☒ | ☐ | Estimates of effect sizes (e.g. Cohen's *d*, Pearson's *r*), indicating how they were calculated |

*Our web collection on statistics for biologists contains articles on many of the points above.*

## Software and code

Policy information about availability of computer code

| Data collection | No software was used to collect the data. |
|---|---|
| Data analysis | We performed all diversification analyses with the fossil record using the PyRate 3.0 program. PyRate is a Bayesian modelling platform to estimate origination, extinction, and preservation rates from fossil occurrence data. The output processing was done with the R environment (R version 4.3.0.). We provided the command lines to run all the diversification models that we have performed with the fossil data sets in the Figshare digital data repository (https://doi.org/10.6084/m9.figshare.25395871.v2). |

For manuscripts utilizing custom algorithms or software that are central to the research but not yet described in published literature, software must be made available to editors and reviewers. We strongly encourage code deposition in a community repository (e.g. GitHub). See the Nature Portfolio guidelines for submitting code & software for further information.

## Data

Policy information about availability of data

All manuscripts must include a data availability statement. This statement should provide the following information, where applicable:
- Accession codes, unique identifiers, or web links for publicly available datasets
- A description of any restrictions on data availability
- For clinical datasets or third party data, please ensure that the statement adheres to our policy

All fossil data sets to repeat the analyses described here are available in the Figshare digital data repository (https://doi.org/10.6084/m9.figshare.25395871.v2).

# Research involving human participants, their data, or biological material

Policy information about studies with human participants or human data. See also policy information about sex, gender (identity/presentation), and sexual orientation and race, ethnicity and racism.

| | |
|---|---|
| Reporting on sex and gender | N/A |
| Reporting on race, ethnicity, or other socially relevant groupings | N/A |
| Population characteristics | N/A |
| Recruitment | N/A |
| Ethics oversight | N/A |

Note that full information on the approval of the study protocol must also be provided in the manuscript.

# Field-specific reporting

Please select the one below that is the best fit for your research. If you are not sure, read the appropriate sections before making your selection.

☒ Life sciences    ☐ Behavioural & social sciences    ☐ Ecological, evolutionary & environmental sciences

For a reference copy of the document with all sections, see nature.com/documents/nr-reporting-summary-flat.pdf

# Life sciences study design

All studies must disclose on these points even when the disclosure is negative.

| | |
|---|---|
| Sample size | We studied Cenozoic diversification using ~220 samples and ~48,000 specimens from 870 species. We used sedimentary rock samples for ostracod analysis. We tried to obtain >200 specimens per sample as far as possible, which is a standard procedure in micropaleontology. |
| Data exclusions | No data were excluded from our study. |
| Replication | We replicated the analyses over ten randomized data sets (randomizing occurrence age for each taxon) and estimated the times of speciation (denoted Ts) and times of extinction (denoted Te) as the mean of the posterior samples from each replicate and for each taxon. |
| Randomization | We did not perform an experiment with group allocation. We replicated the analyses on ten randomized data sets and estimated the times of speciation and times of extinction as the mean of the posterior samples from each replicate. There was no further group partitioning of data beyond the natural groupings (allocated to ostracod genera and species). |
| Blinding | Blinding was not relevant to this study, since no human subjects or experiments were involved. |

# Reporting for specific materials, systems and methods

We require information from authors about some types of materials, experimental systems and methods used in many studies. Here, indicate whether each material, system or method listed is relevant to your study. If you are not sure if a list item applies to your research, read the appropriate section before selecting a response.

## Materials & experimental systems

| n/a | Involved in the study |
|---|---|
| ☒ | ☐ Antibodies |
| ☒ | ☐ Eukaryotic cell lines |
| ☐ | ☒ Palaeontology and archaeology |
| ☒ | ☐ Animals and other organisms |
| ☒ | ☐ Clinical data |
| ☒ | ☐ Dual use research of concern |
| ☒ | ☐ Plants |

## Methods

| n/a | Involved in the study |
|---|---|
| ☒ | ☐ ChIP-seq |
| ☒ | ☐ Flow cytometry |
| ☒ | ☐ MRI-based neuroimaging |

## Palaeontology and Archaeology

| | |
|---|---|
| Specimen provenance | From Indonesia and Philippines with local collaborators as in the author list. |
| Specimen deposition | Specimens are deposited at the University of Hong Kong. |
| Dating methods | Biostratigraphy |

☒ Tick this box to confirm that the raw and calibrated dates are available in the paper or in Supplementary Information.

| | |
|---|---|
| Ethics oversight | None. Microfossil has no ethics or commercial relevance |

Note that full information on the approval of the study protocol must also be provided in the manuscript.

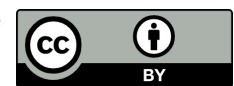

