## [Peer Review file · Nature]

Manuscript Title: Cenozoic history of the tropical marine biodiversity hotspot

Reviewer Comments & Author Rebuttals

Reviewer Reports on the Initial Version:

Referee #1 (Remarks to the Author):

The authors provide an interesting analysis of diversification dynamics in the IAA from the Eocene to the present-day. They couple these analyses to models that try to test putative drivers of diversification, including temperature, sea level, and diversity itself. The authors findings are varied and at times difficult to follow, but generally claim that diversity itself and the closing of the Tethys helped build up diversity in the IAA. The manuscript is generally well written and the data are impressive. I commend the authors. However, I had a number of questions and potential issues that arose whilst reading the manuscript, which I detail below:

As I was reading the manuscript, I wondered how well ostracod diversity correlates with overall diversity in marine hotspots and more globally. The conclusions of the authors hinge on this correlation, and I think the authors need to do more work to show this is the case, in order to make their conclusions from the ostracod record.

There seems to be a key gap in sampling data at the Oligocene, which could influence interpretations, and this needs to be addressed by the authors. How might this impact their diversification parameters? Why does this gap occur?

I wanted to know how much spatial heterogeneity in diversity there is across the study region compared to through time. This is a very important point that the authors neglect to discuss. If diversity varies significantly within a time period – e.g., more so than across time periods – this variation may invalidate some of the conclusions drawn by the authors.

What could cause both speciation and extinction (high turnover) at 20 Ma, 16 Ma, and 5 Ma? It is concerning that these bursts do not correspond with any notable changes in the potential drivers that the authors studied. This makes me wonder whether the authors are actually including / studying the key drivers, and/or whether the data fed into the models is still biased in terms of sampling (or other factors). The authors go to lengths to explain putative key drivers of diversification in the IAA based on their model, such as temperature and sea level change, but then note that the major events they have recorded (such as peak speciation and extinction at 20 Ma, 16 Ma, and 5 Ma) do not correspond with any notable changes in any of their included drivers. Thus, they are unable to say anything concrete with regard to their primary data, but instead need to propose hypotheses based on other Earth history events / previous literature. The authors in particular point to the closure of the Tethys. The fact they cannot explain these events based on the predictors included in their model, but instead point to the closure of a seaway, muddies the waters in terms of interpreting overall patterns. I wondered whether these major biogeographic events and changes could also be included directly (even if as a more categorical variable) into the model itself, alongside the continuous predictors of temperature, sea level, etc, so as to actually test which

predictors/events seem to correlate most strongly with the observed diversification patterns. (They do a test of the hopping hotspot model, to see if Tethyan faunal elements contribute to diversity dynamics, but this analysis does not get at the question at hand). The explanation by the authors is that the Tethys closure may have caused the spike in diversification due to large-scale habitat fragmentation and isolation; however, in the models they ran, fragmentation did not correspond with diversification, and they explain why this may not be important in the marine realm. At face value, this discrepancy is confusing and contradictory. Finally, the authors themselves note that the pattern of peak diversification at these intervals cannot be completely ruled out by the binning of their fossil data. However, I didn't see any mention of sensitivity tests of their binning scheme, and it would be useful to do an analysis to stress test the peaks and whether they could have been caused by the temporal binning.

Line 324: what is slightly depauperate mean? Can this be quantified? Also, I suggest the authors provide quantification of what is "much higher richness" in this context.

Line 620: I was a bit concerned to read that the author 'tentatively assume' that all aggregated samples are representative of the IAA. Why is this tentative, and how did they determine provenance?

It seems some of the environmental parameters used to explain diversification dynamics, such as temperature estimates, were not estimated from the IAA region (or roughly equivalent regions), but rather taken from global averages (using isotopic proxies). This seems problematic as the authors are not trying to describe some global signal, but rather the dynamics within a particular region. I would think the authors should measure spatially-explicit parameters (perhaps from GCMs, as they do the paleogeography) within their study region, to get at the true drivers of this regional signal. I would imagine that they would not find temperature to be significant if they did this.

The 'hopping hotspots' model should be introduced briefly.

There were some areas of the text that were unclear. For example, I would have liked to see more information on the raw analyses of the ostracod dataset – what sort of species range plots were created? I didn't find this in the methods.

Line 161: what is a substantial degree of diversity dependence? Can this be quantified? Did you fit different models to test whether the curve most closely approximates a logistic function, or was this eyeballed?

The authors state (line 166) that lineages and communities may commonly experience a time decay in their diversification rates as the growth of diversity itself becomes a limiting factor, however, it wasn't clear how this was measured, nor what the actual time delay is, on average, and how this varies across taxa. Furthermore, when did this time delay first become noticeable as diversification increased? Can this be tested statistically?

Line 177: The authors infer that diversity dependence causes the steady state in diversity seen post

2.6 Ma. While this may be true, I wondered if the authors couldn't test this in a more rigorous way, by running a model with diversity included as a predictor, alongside other key predictors such as climate. As the authors will be aware, there were significant climatic and oceanographic changes around 2.6 Ma, and I would think these abiotic changes could also explain changes in diversification dynamics at this time. Edit: this seems to have been done (with both abiotic and biotic factors in the same framework) but is not made particularly clear throughout the text and should be revised accordingly (both main text and methods).

What is known about what structures ostracod speciation dynamics? Does speciation occur primarily allopatrically, and has anyone studied drivers previously? This information would potentially provide useful context for the diversification studied here.

Line 188: it is not clear whether the authors found temperature to be most important in diversification, or whether this is a general statement derived from past knowledge. If the latter, there should be citations provided for previous work on ostracods.

The authors found a positive relationship between temperature and tropical extinction and an inverse relationship between temperature and tropical speciation. This is interesting, and the explanation is that it may have been too hot for species during greenhouse climates. I wondered if the authors considered the actual estimates of temperature (from proxy data or GCMs) for these intervals and this region, compared to the known temperature thresholds that may prove problematic for species, and, therefore, whether this explanation is actually reasonable? My concern is that the authors did not actually examine temperatures within this region. The methodology of using global estimates is problematic, and, after learning this, I was not surprised to learn that the authors found a pattern that contradicts previous work, including their own, using this methodology. I would be very curious to know whether these results hold when measuring temperature from low-latitude regions only (or the IAA only) – I would expect potentially not. Regardless of these issues, the entire discussion of these results is a bit muddled and difficult to follow, at present.

It's interesting that the authors found that habitat complexity does not correlate positively with diversification, counter to previous hypotheses. They propose a valid explanation for this surprising pattern. However, I wondered whether this relationship could have been found because of the focus on ostracods, whereas other taxa may be influenced more by environmental fragmentation. To me, this is one of the most interesting and important results from this study, and I am surprised that this lack of relationship between diversity and suitable habitat complexity is not trumpeted more as a key finding.

Overall, I think this is an interesting paper with some important analyses, but the key findings and message are difficult to disentangle from the text. I would suggest simplification and streamlining. As an example, see my previous comment above. As a further example, the authors generally found low extinction, which likely contributes to the high diversity in the region, which should likely be highlighted more clearly in the manuscript.

More minor comments:

Line 46: suggest that 'history' and 'explicitly' are not needed here.

Line 49: suggest 'a' versus 'the diversity...'

Line 54: It was unclear to me what 'per se' meant in this context. I believe you mean diversity itself drives diversity (as in diversity dependence), and 'itself' may be a better word here if so.

Line 77: remove 'and' before 'which'

Line 79: Is it the most diverse place on the planet, or is it one of the most diverse in the marine realm?

Line 94: There is an extra space between time periods.

Line 119: please insert an 'and' before the '2'

Line 144: This sentence is not complete and should not start with 'But'

Line 272: Please consider revising and better introducing the start of this sentence.

Line 602: Please fix the question marks and incomplete sentence and explanation here.

Referee #2 (Remarks to the Author):

Dear Authors,

Thank you for submitting this fantastic study detailing the evolution of one of the most important biodiversity areas of the modern world. I think the results of this paper are exciting in showing how the interactions of different biotic and abiotic variables shape the biodiversity patterns we see around us, and how biodiversity shifts longitudinally and latitudinally through time.

Overall, as you will see from my comments, I suggest primarily minor changes to the text and figures to improve the clarity for Nature's broad viewership. However, I also suggest that some methodology of subsampling/sampling standardization be carried out due the lower sample coverage deeper in time.

I recommend that this be published in Nature after moderate revisions which I have detailed in the attached PDF. I very much look forward to this being added to the literature.

Your sincerely,

Reviewer

Lines 69-70 – Why do regionally resolved Cenozoic diversity trans remain poorly understood? Taphonomic biases? Preservation potential? Poor sampling?

Line 85 – What are the other taxonomic groups and what are their limitations?

Line 99 – What is the record in the IAA like for planktonic organisms?

Line 108 - Perhaps state earlier on that this is a study that is focused on benthic organisms (though fish are mentioned a lot), explain why planktonic organisms (generally) are not being used in this study

Line 147 – It may be more appropriate to have "Pleistocene" on the figures instead of "Quaternary" as Pleistocene is used more in the text

Line 149 – Very interesting!

Line 156 - These are not strictly "stages" in the chronostratigraphic context. Interval or time bin may be better to avoid confusion

Lines 167-173 – Great stuff there!

Lines 180-181 - Or another plateau for the time being? What is our understanding of ostracodal cryptic species?

Lines 189-196 – Interesting!

Line 203 - Briefly summarise these other mechanisms if possible. Also include Woodhouse & Swain et al. (2023), these 3 articles represent a unified theory

Woodhouse, A., Swain, A., Fagan, W.F. et al. Late Cenozoic cooling restructured global marine plankton communities. *Nature* 614, 713–718 (2023). <https://doi.org/10.1038/s41586-023-05694-5>

Lines 208-212 - What is the sampling coverage of the tropics during the Cenozoic? Though I agree with your interpretations, be sure to highlight potential sampling biases

Line 222 – Put southern first so it is in order of glaciation

Line 237 - They are certainly MORE sparse compared to the terrestrial realm, but there are many of them. See:

Norris, R.D., 2000. Pelagic species diversity, biogeography, and evolution. *Paleobiology*, 26(sp4), pp.236-258.

Lines 243-247 – Awesome

Lines 261-264 - Why may they have not moved prior to this?

Line 269 - Have the authors considered how much pseudospeciation and pseudoextinction is occurring at this time? This is a general question on the whole study, and may be worth considering at some point within the manuscript considering how much turnover there seems to be at 20, 16, and 5 Ma. Clearly more morphospecies are evolving, but how many are actually becoming truly extinct?

Line 293 - More records may be needed to confirm whether it was global

Line 299 - huge spatial differences?

Lines 315-318 - This is super interesting and super important - the world around us may be the exception, not the rule when it comes to understanding life

Line 323 - How are these defined? Are Tethyan genera those that first appeared in Tethys (prior to ~ 20 Ma), and cosmopolitan first appeared anywhere other than Tethys? What about genera that first appear specifically in the IAA?

Line 340 - Is this what is shown in Figure 3 (left), it may be clearer to show Cosmopolitan, Tethyan and IAA genera individually if possible.

Lines 348-355 - Interesting!

Lines 361-365 – This is great!

Lines 369-370 - Immunity of ostracods or biodiversity as a whole?

Line 376 - And latitudinal correct?

Lines 593-595 - What biostratigraphic scheme was used? Is it calibrated to the GTS 2020?

Line 615 – See my comments on Fig S1

Lines 622-623 - It would be good to integrate some form of subsampling or sampling standardization to check whether the amount of samples per time bin has an effect.

Lines 741-743 - Was there a specific biogeographic range which defined a cosmopolitan species as opposed to a more endemic one? - So does this analyses include all taxa that also originated in the IAA? It may be useful to plot this data too?

Line 756 - Nice figure

Line 759 - Is this cumulative diversity, including extinct species?

Supplementary Data File

Table S1 - A Period is distinct chronostratigraphic class of time, perhaps use "Interval" or "Time bin"

-What is the palaeoenvironment of each of these sample sites?

Fig S1 - Is there anyway the sample sites could be identified? It may make this image too busy, but a blown out Map with sample sites, and country names may help those less familiar with IAA geography

-It would be beneficial to have the age ranges of each of these bins on this figure/in the caption

Referee #3 (Remarks to the Author):

The manuscript 'Cenozoic history of the tropical marine biodiversity hotspot' by Tian, Yasuhara, et al. is a solid manuscript that is well-written and suitable for publication in Nature. The authors use a new dataset of ostracod taxa from the Indonesian region across the Cenozoic to determine macroevolutionary processes that led to the IAA being a diversity hotspot. They found that the IAA region has exhibited an diversity trend through the Cenozoic, with four distinct peaks in speciation rate and net diversification rate, but only three peaks in extinction rates. Such peaks correspond to abiotic factors that have shaped the IAA to be a hotspot. Of distinct importance is that the absence of an mass extinctions during the Cenozoic has led to the IAA becoming a marine biodiversity hotspot. The methods used in the manuscript are appropriate for the analyses performed, and the resulting data are presented in very nice figures. The authors use statistics appropriately and report the values where necessary. The conclusions of the paper are robust, and in addition, this paper lies the groundwork and provides hypotheses for future researchers to delve into. My suggestions for improvement are below:

Line 77: Change "and which might impair understanding" to "which impairs understanding"

Line 86: Could take out "have gradually revealed" to "reveal" to save some space

Line 93: This sentence could be reworded a bit for clarity to something like "... increased richness due to immigration into the IAA in the Eocene-Oligocene..."

Lines 98–99: In this sentence, you are pointing out why the biodiversity history of the IAA is limited for these groups, but for the benthic foraminifera, it's not that their fossil record is poor but rather their species' richness is dampened. This, to me, signals that something is going on with the large benthic foraminifera diversity in the IAA region that doesn't match the results of your study. I think it's okay to leave this statement about the larger benthics in, but it's more of an interesting result rather than a limiting factor.

Line 100: Suggest changing "problem" to "limitation"

Line 115: Change "a" to "the", so "the first comprehensive Cenozoic fossil dataset..."

Line 119: This is up to the authors, but I would suggest changing "extinction rates" to "extirpation rates" as you do note earlier in the sentence that this is a regional analysis, not global. Add an "and" before "(2)".

Line 12: access or assess? I think it's supposed to be assess? I also suggest changing this sentence to be more clear to something like "...correlate the IAA's macroevolutionary dynamics with a set of biotic and abiotic parameters to assess the potential biodiversity drivers"

Line 140: Capitalize Early, Middle, and Late Miocene here and throughout the manuscript.

Lines 144–145: This sentence about not having Oligocene data should be moved after the sentence

about low Eocene diversity, so it is not confusing to readers why the Oligocene is not explicitly mentioned as also having low diversity.

Line 148: The 600 species is referring to species richness rather than diversity, so suggest changing “diversity” in this sentence to “richness”

Figure 1: Really great figures, very easy to comprehend and good info communicated. The only suggestion I have is to please plot A, B, and D with the same y-axes. At first look, it appears as though the magnitude of the origination and extinction rates are the same, which led me to think these may be turnover events. Only later did I realize that the y-axes were not the same. Plotting the data on the same y-axis scale will give readers a better sense of how larger the origination rates are compared to the extinction rates.

Table S2: Define HPD in the table caption.

Line 203: Either take out the sentence “Other mechanisms could also be possible”, or expand on this sentence a bit more to precisely state what other mechanisms could be possible.

Line 214 and elsewhere: Is there a way to quantify ‘strong’ and ‘relatively strong’ when referring to the factors that influence speciation? If not, it would be helpful to define what the authors mean by ‘very strong’, ‘strong’, and ‘relatively strong’. Maybe include some sort of key or categories on Figure 2 to be more clear.

Line 223: I’m not too certain what ‘its’ is referring to in this sentence. Please clarify.

Line 226: Here, when referencing habitat size and complexity, it would be more helpful to the readers to either include “(shelf area)” and “(habitat size)” after these phrases, respectively, or include ‘habitat size’ and ‘complexity’ on the Fig. 2 y-axis. This can be simply overlooked in the figure caption.

Lines 229-231: This is just a fun thought for the authors: Have you considered time-slicing the MBD analysis? The time slices could be different climate regimes pulled from CENOGRID by Westerhold et al. (2020), and may be a more refined way to test your hypotheses put forth in these sentences? It would be interesting to see if and how the MBD analyses change through time.

Line 275: Slight reword suggested ‘The 12-Ma event is apparent only in the speciation rate...’

Lines 287–289: Could the increase and subsequent decrease in speciation and extinction around 5 Ma also be due to closure of the Central American Seaway? Increased diversity/richness with increased extinction/origination rates are apparent for radiolarians (Lowery et al., 2020 Annual Review of Earth and Planetary Sciences) and foraminifera at global (Fraass et al., 2015 Annual Review of Earth and Planetary Sciences) and regional (Lam and Leckie, 2020, Micropaleontology) scales. The constriction of the CAS reached a critical threshold around 4.8-4.0 Ma (e.g., Haug & Tiedemann, 1998; Steph et al., 2006, 2010).

Line 299: Add 'found' between 'that' and 'huge'

Lines 298–302: This sentence is a bit chewy, and I don't completely follow and understand. Are you stating that huge differences in extinction rates among different taxa contribute to modern longitudinal diversity patterns? I think just a re-wording here for clarity is necessary.

Line 361: Delete the 'in' at the end of this sentence.

Lines 593–595: Include references for your biostratigraphic schemes used (e.g., King et al. 2020 and Wade et al., 2011 for Cenozoic planktic foraminiferal biozone ages). Also, what time scale were the age models put on? Geologic Time Scale 2020 or other?

Referee #4 (Remarks to the Author):

The manuscript “Cenozoic history of the tropical marine biodiversity hotspot” uses ostracod records and PyRate/Birth-Death models to look at origin vs extinction rate through time and biodiversity within the Coral Triangle. The paper identifies an exponential increase in biodiversity with several peaks, one of which it is postulated could be related to the closure of the Tethys seaway. Temperature and diversity are the drivers identified for biodiversity. The stability ie. lack of major extinction events in the region is also a major factor in creating today’s biodiversity in the region. In addition it also looks at the amount of Tethyan biodiversity in the assemblages and what factors may have influenced this through time.

Biodiversity, drivers of diversity and the future of the Coral Triangle are current major concerns, the paper is therefore exceptionally timely and offers a new perspective on this. Ostracode data have not been used to model Coral Triangle biodiversity before. Understanding where this biodiversity comes from is exceptionally important for our understanding of current and future biodiversity. But is also a difficult task, and therefore new perspectives are significant and of broad interest.

It is interesting to see a study for the region carried out with ostracodes as previous work, as the authors mention, has been carried out on more reefal organisms such as bryozoans, corals, and larger foraminifera. Whilst no single record is perfect, it is important for multiple perspectives and records to be used. Ostracodes therefore add to this. Perhaps the authors could comment on, or further emphasise whether ostracodes have mirrored trends of larger organisms in previous studies.

The methods used are largely standard PyRate methods, seem appropriate for the dataset and are well described for others to follow. Figures are largely appropriate and clear although I have two smaller comments:

- I think the authors do need to clarify for figure 1 that the rate of speciation given is per lineage as this is not immediately clear for general readers who do not use these methods.
- Within figure 1D it is unclear if these numbers are taken from PyRate or from counts as this could result in small differences in the timing of plateaus.
- Something to consider but not essential is whether it might be useful to plot the extinction and origination rates on the same plot to allow easy comparison on the same scale.

The largest issue with the dataset is the lack of data in the Eocene and Oligocene and given the geology of the region this is not necessarily something that can be fixed. Of the 216 samples only three are Eocene and none from the Oligocene and the graphs show very low diversity across this interval. Although PyRate can statistically account for sampling bias and preservation etc. it still is working from the data that it is given and would therefore be an underestimate of any Eocene-Oligocene biodiversity. This is very difficult to address as suitable samples likely simply do not exist or are exceptionally rare – much of the Eocene of the region is hard limestone, unsuitable for ostracod studies. The authors do mention the lack of samples, but they need to be tentative regarding any conclusions based on the Eocene and Oligocene. Another possibility is that the paper could be framed as a more Neogene rather than Cenozoic perspective, as they do have a good number of samples from the Early Miocene onwards.

I am sure the authors are well aware of this issue but I would like to know their further thoughts on

how valid the model results are for Eocene-Oligocene and whether the lack of data early on influences the interpretation of the first diversity peak.

The authors in line 263 talk about Tethyan taxa arriving in the IAA in the early Miocene creating a diversity peak, although in figure 3 it appears that there is a greater diversity of Tethyan taxa in the Late Miocene?

It is interesting that speciation increases with the middle Miocene Climatic Optimum, and I agree with the authors that it could be related to other changes related to circulation etc. Warming also does not seem to have always been detrimental to all shallow water organisms, following the PETM larger foraminifera massively diversified, possibly due to available niches but it remains fairly uncertain.

There is good discussion of events and possible causes along with comparison to the Caribbean, but I wonder if some kind of summary figure could be made indicating the tentative causes for the peaks and timing of Caribbean events.

The conclusions regarding Tethyan taxa are sensible, and the paper finishes with a strong valid point regarding the future.

References are appropriate and cover the topic well and it is overall well-written, and reads clearly.

Author Rebuttals to Initial Comments:

Resubmission of Nature manuscript 2023-08-15356

Referee #1 (Remarks to the Author):

1. The authors provide an interesting analysis of diversification dynamics in the IAA from the Eocene to the present-day. They couple these analyses to models that try to test putative drivers of diversification, including temperature, sea level, and diversity itself. The authors findings are varied and at times difficult to follow, but generally claim that diversity itself and the closing of the Tethys helped build up diversity in the IAA. The manuscript is generally well written and the data are impressive. I commend the authors. However, I had a number of questions and potential issues that arose whilst reading the manuscript, which I detail below:

Response: Thank you for your thorough review and generally positive feedback on the study. Below we have done our best to address each question you have raised.

2. As I was reading the manuscript, I wondered how well ostracod diversity correlates with overall diversity in marine hotspots and more globally. The conclusions of the authors hinge on this correlation, and I think the authors need to do more work to show this is the case, in order to make their conclusions from the ostracod record.

Response: This is a very good point. We now show that ostracods are good representative or proxy of broad marine benthic biodiversity by adding the following details at Line 120 in the Introduction:

“Overall, Ostracoda (Arthropoda: Crustacea; known as seed shrimps) is one of the few benthic microfossil organisms that has left a rich fossil record within and beyond reef ecosystems for quantitative analysis¹⁷. Their high species diversity and robust taxonomy are two other major advantages¹⁷. Benthic ostracods show normal latitudinal diversity gradient¹⁸ and depth diversity gradient¹⁹. They also display a similar biogeographic distribution with other invertebrates²⁰. Thus, instead of being a contrarian, ostracods are regarded as a normal benthic taxon that tends to follow standard ecological patterns¹⁸. In addition, small (<0.5 mm) benthic metazoan invertebrates account for the vast majority (more than two-thirds) of marine biodiversity^{18,21}, and ostracod is probably the best fossil representative for this group in terms of general biotic response^{14,17}. These features make ostracod a very useful proxy for broad marine benthic biodiversity to investigate the ancient history of hotspots prior to the time scale of modern observations^{14,17}.”

3. There seems to be a key gap in sampling data at the Oligocene, which could influence interpretations, and this needs to be addressed by the authors. How might this impact their

diversification parameters? Why does this gap occur?

Response: This is a fair point. We acknowledge that the lack of Oligocene data is a limitation of our study. Oligocene materials were rarely preserved in the tropical IAA region in shallow-marine settings, and they are often hard limestone if existed, which is unsuitable for ostracod research (Ostracods cannot be studied with thin sections and they cannot be disintegrated from the limestone without being dissolved). We discussed the sampling gap at Line 175 in the main text:

“Nevertheless, it should be noted that there is a major sampling gap across the early Oligocene, so the possibility of an earlier diversification before 25 Ma cannot be completely ruled out^{8,23}.”

To show this sampling gap does not affect the overall Cenozoic trends, we repeated the BD modelling using the Neogene data only (Extended Data Fig. 7) as a sensitivity test. We added the following sentences at Line 793 of the Materials and Methods:

“Regarding the estimation of the speciation and extinction dynamics, we noted that potential sampling bias could be caused by the lack of the Oligocene data and the binning of fossil record. To explore the sensitivity of the speciation and extinction estimates, we repeated all the BDCS analyses for the Neogene period only with time bins of 2 Myr, for the entire Cenozoic with time bins of 5 Myr, and for the Neogene period only with time bins of 5 Myr. All these analyses were parameterized following the main analysis. All results corroborate a reasonably robust and unambiguous pattern of strong and continuous diversification throughout the studied intervals (Extended Data Figs. 7-9).”

Furthermore, we repeated the MBD modelling with the Neogene data only, to show the Oligocene gap does not affect the identification of long-term diversification drivers (Extended Data Table 7). The revisions are at Line 803 and as follows:

“For disentangling the paleoenvironmental controls on diversification dynamics, we reckoned that the use of global temperature in a regional analysis of the IAA could pose certain limitations on our modelling, together with the Oligocene sampling gap. We ran two sensitivity MBD analyses using the tropical temperature record from Ref.⁷⁸ for the Cenozoic and for the Neogene instead of the global temperature. With all other parameters unchanged, the results remain consistent with those from the main analysis, still showing that diversity dependence and habitat size are two determinants of speciation (Extended Data Tables 6-7).”

4. I wanted to know how much spatial heterogeneity in diversity there is across the study region compared to through time. This is a very important point that the authors neglect to discuss. If diversity varies significantly within a time period – e.g., more so than across time periods – this variation may invalidate some of the conclusions drawn by the authors.

Response: We now show the diversity map per time interval in the Extended data Figure 4. Our sampling sites are widely distributed among the Philippines and Indonesia, covering the area of highest modern biodiversity (Förderer et al., 2018; Hoeksema, 2007; Sanciangco et al., 2013; Veron et al., 2011). For the most sampled islands, i.e., the Philippine islands and Java, we see a congruent increase in diversity across each time interval since the middle Miocene. In addition, the patterns of spatial heterogeneity within the IAA region were quite consistent through time that central Philippines represented the bullseye of species diversity, which is again consistent with modern distribution pattern of overall marine species richness (Förderer et al., 2018).

Extended data Figure 4. Diversity map showing the predicted species richness at $n=200$ individuals from rarefaction in each geological interval. Eocene: 56-33.9 Ma; Oligocene: 33.9-23.04 Ma; early Miocene: 23.04-15.99 Ma; middle Miocene: 15.99-11.65 Ma; late Miocene: 11.65-5.33 Ma; Pliocene: 5.33-2.58 Ma; Pleistocene: 2.58-0.01 Ma.

The following text was added at Line 180 to address spatial heterogeneity:

“Within the IAA, the Philippines emerged as the bullseye of ostracod diversity from the late Miocene to Pleistocene, which is congruent with modern distributions of overall marine species richness (Extended Data Fig. 4)^{11,24}.”

And the following at Line 671 of the Materials and Methods:

“Spatial coverage of all aggregated samples is reasonably good for representing the IAA region as a whole, especially within the Philippines and central Indonesia where modern diversity is known to be the highest (Extended Data Fig. 4)^{11,24}.”

5. What could cause both speciation and extinction (high turnover) at 20 Ma, 16 Ma, and 5

Ma? It is concerning that these bursts do not correspond with any notable changes in the potential drivers that the authors studied. This makes me wonder whether the authors are actually including / studying the key drivers, and/or whether the data fed into the models is still biased in terms of sampling (or other factors). The authors go to lengths to explain putative key drivers of diversification in the IAA based on their model, such as temperature and sea level change, but then note that the major events they have recorded (such as peak speciation and extinction at 20 Ma, 16 Ma, and 5 Ma) do not correspond with any notable changes in any of their included drivers. Thus, they are unable to say anything concrete with regard to their primary data, but instead need to propose hypotheses based on other Earth history events / previous literature. The authors in particular point to the closure of the Tethys. The fact they cannot explain these events based on the predictors included in their model, but instead point to the closure of a seaway, muddies the waters in terms of interpreting overall patterns. I wondered whether these major biogeographic events and changes could also be included directly (even if as a more categorical variable) into the model itself, alongside the continuous predictors of temperature, sea level, etc, so as to actually test which predictors/events seem to correlate most strongly with the observed diversification patterns. (They do a test of the hopping hotspot model, to see if Tethyan faunal elements contribute to diversity dynamics, but this analysis does not get at the question at hand). The explanation by the authors is that the Tethys closure may have caused the spike in diversification due to large-scale habitat fragmentation and isolation; however, in the models they ran, fragmentation did not correspond with diversification, and they explain why this may not be important in the marine realm. At face value, this discrepancy is confusing and contradictory. Finally, the authors themselves note that the pattern of peak diversification at these intervals cannot be completely ruled out by the binning of their fossil data. However, I didn't see any mention of sensitivity tests of their binning scheme, and it would be useful to do an analysis to stress test the peaks and whether they could have been caused by the temporal binning.

Response: Thank you very much for your elaborate and thoughtful comments. To address them, we updated the age model to the Geological Time Scale 2020, re-ran the main MBD analysis, and supplemented it with several sensitivity analyses. We also performed a time-stratified MBD analysis. The revisions are listed below.

Firstly, the entire time frame MBD analysis indicated habitat size (shelf area) as the most important abiotic driver of speciation. We revised the corresponding paragraph starting at Line 237 as follows:

“In addition to diversity dependence that imposed a strong biotic control on the IAA, we showed that habitat size (shelf area) as the most important abiotic determinant has a positive, albeit weak, effect with speciation, indicating an evolutionary species-area relationship ($G= 2.272E-7$, $\omega=0.647$; Fig. 2A; Extended Data Table 2). Larger shelf areas could promote speciation either through direct effects of area or through the effects of factors that are highly correlated with area, such as population size, species range, and environmental heterogeneity^{30,31}. The Neogene expansion of shallow-marine habitats in the IAA, which was driven by the prolonged collisions between Southeast

Asia and Australia^{22,32}, was therefore pivotal in the rise of the IAA hotspot. Unexpectedly, habitat complexity (coastline length) as another putative diversification driver has no effect here (insignificant correlation with both speciation and extinction rates) ($\omega=0.272$; Fig. 2A; Extended Data Table 2). Our results appear to contradict previous theories suggesting that complex island archipelagos of the IAA with an extensive array of shallow seas accelerate allopatric speciation via vicariance, although isolated small populations may also face higher extinction risks^{1,3}. The effects of habitat complexity on regional diversification are intriguing and worthy of further investigation. Indeed, in the marine realm, dispersal (both positively and negatively) as the dominant process may overwhelm the effects of isolation at a regional scale, where the physical barriers are often permeable and relatively sparse³³⁻³⁵. Connectivity among marine basins may be substantially restricted because of a dynamic mosaic of ever-changing geographic and oceanographic conditions within the IAA, but not strictly or permanently blocked, and thus undermines the effectiveness of conventional allopatry. On the other hand, our findings may imply the importance of sympatric speciation, given the positive effect of habitat size whereas no effect of habitat complexity^{34,36}. In large, continuous shelf habitats across wide environmental gradients, such as the IAA, diversification may occur along ecological partitions (i.e., intense competition invokes finer subdivision and niche specialization) in addition to geographic partitions^{28,34-36}. Evidence of sympatric speciation is prevalent among marine clades of all different dispersal abilities and life histories, ranging from fishes, corals, gastropods, to ostracods^{34,36,37}, which potentially highlights the importance of ecological factors in marine diversification. Collectively, our MBD analysis partly supports the *Hopping Hotspots* model that proposes tectonic activity as the principal forcer of biodiversity hotspot by creating larger and more complex shallow-marine habitats^{1,10}. We instead suggest that suitable habitat in terms of size but not complexity is important for the generation of enormous IAA diversity throughout the Cenozoic. But note that the habitat estimations available now are based on global paleogeographic reconstructions, so some uncertainty remains about the accuracy of such an inference. Other than the habitat factors, global temperature and sea level do not significantly correlate with diversity dynamics in the entire time frame analysis (G parameters overlapping with zero; Fig. 2A; Extended Data Table 2). This seemingly indicates that not all TECOG (tectonic, eustatic, climatic, oceanographic and geomorphological) processes¹ are as critical as previously thought in fostering a biodiversity hotspot, or their short-term effects may be masked across a long time frame (see the next section).”

Secondly, the time-stratified MBD analysis enables more detailed interpretation of the time-varying effects of temperature, sea level, and habitat size. The paragraph has been added at Line 281 and as follows:

“After establishing the general long-term drivers of the IAA’s diversification dynamics, we then scrutinized how their effects may vary over time across different climate regimes³⁸. A time-stratified MBD analysis revealed further details for the warm phase (23.04-13.9 Ma), cooling phase (13.9-5.33 Ma), and cold phase (5.33-0 Ma) of the Neogene period (Fig. 2B-D; Extended Data Tables 3-5). The results of the cold phase are

highly concordant with those of the entire time frame, with diversity ($G=-3.341$, $\omega=0.912$) and habitat size ($G=2.445E-7$, $\omega=0.702$) negatively and positively affecting speciation, respectively (Fig. 2D; Extended Data Table 5). Higher sea level also expedited speciation during this interval ($G=0.022$, $\omega=0.819$), likely through the species-area relationship. Large and frequent fluctuations in sea level associated with the glacial cycles may strongly impact the expansion and contraction of epicontinental seas and consequently the size of shallow-marine habitats^{39,40}. There is no correlation between diversification dynamics and any biotic or abiotic factors for the cooling phase (Fig. 2C; Extended Data Table 4), which may indicate it as a transitional stage between two opposite climate regimes. Indeed, during the warm phase, biotic and abiotic controls on the diversification of IAA faunas occurred in very different ways from those of the cold phase (Fig. 2B; Extended Data Table 3). The strong positive effect of diversity on extinction conforms to our understanding of diversity dependency ($G=336.168$, $\omega=1$), but its positive effect on speciation ($G=25.093$, $\omega=0.999$) may be explained by higher availability of empty niches when there was much less species, as the IAA hotspot just originated during the early Miocene. Higher sea level reduced extinction as expected ($G=-0.843$, $\omega=0.999$), while it also reduced speciation, albeit weaker ($G=-0.069$, $\omega=0.995$). A possible explanation is that Southeast Asia was still separated from the Australian and Pacific plates by deep oceans^{22,41}, so high sea level may not contribute to form large areas of shelf habitats in the IAA. Most interestingly, high temperature triggered both speciation ($G=0.584$, $\omega=0.993$) and extinction ($G=5.038$, $\omega=0.999$) during the warm phase, which may be due to the coexistence of warm-adapted and cold-adapted species. As Earth's climate showed a long-term cooling trend across the Cenozoic⁴², some warm-adapted species that were resilient to or even preferred very high temperatures of the Eocene and Oligocene might not yet be extinct in the relatively warm early-middle Miocene⁴³. They may have responded to temperature positively with more active proliferation. Cold-adapted species on the other hand gradually rose throughout the Neogene as the climate further cooled^{4,43,44}, but their vulnerability to high temperature may put them in much higher extinction risks during the relative warm early-middle Miocene. Indeed, recent studies suggest that tropical temperatures $>25^{\circ}\text{C}$ may be too elevated for marine organisms due to metabolic trade-offs driven by temperature effects on hypoxia sensitivity^{25,45-47}. Consistent evidence for thermal stress on tropical biodiversity comes from various marine taxa (e.g., coral, reef fish, reptile, calcareous algae) that declined in diversity in the tropics and exhibited poleward range shifts during historical warm intervals^{48,49}. Since temperature has a much stronger impact on extinction than on speciation ($G\lambda=0.58$, $G\mu=5.04$), our analysis suggests that too high a tropical temperature in warm climates slows down diversification and hinders the growth of the hotspot. The effects of temperature are not significant for other time windows, suggesting that thermal stress may be relieved in cooler climates. We need to be aware of future extinction risks and tropical biodiversity loss exacerbated by anthropogenic warming under high greenhouse gas emission scenarios, as tropical biotas are already living close to their ecophysiological thresholds⁴⁶."

Fig. 2. Bayesian inferences of correlation parameters on speciation (red) and extinction (blue) with abiotic factors including global temperature, global sea level, IAA habitat size (shelf area), and IAA habitat complexity (coastline length); and diversity dependence factor, i.e., the diversity over time of the entire ostracod assemblage. Time frame of (A) entire Cenozoic; (B) Neogene warm phase (23.04-13.9 Ma); (C) Neogene cooling phase (13.9-5.33 Ma); (D) Neogene cold phase (5.33-0 Ma). The asterisk (*) indicates a significant correlation parameter for a given variable (shrinkage weights $\omega > 0.5$).

Thirdly, we discussed key tectonic, climatic, and oceanographic events that were possibly related to the diversification peaks at ~25, 20, 16, 12, and 5 Ma. The revised text at Line 331 is as follows:

“We have shown that the long-term diversification of the IAA is influenced by a time-sensitive series of biotic and abiotic factors. Coherently, we suggested that critical shifts in tectonics, climate, and oceanography may be responsible for short-term events of speciation and extinction, but also drove the biogeographic evolution of the IAA hotspot (Extended Data Fig. 5). Diversification initiated in the IAA with concurrent speciation and extinction peaks at 25 Ma, when Australia first collided with Southeast Asia^{22,32}.

The second and the largest peak in both speciation and extinction occurred at 20 Ma. It correlated in time with two major tectonic events outside and within the IAA, namely the closure of the Tethys Seaway separating the Indo-Pacific from the Mediterranean Sea and Atlantic Ocean ^{10,50}, and the closure of the Indonesian deep-water passage between Southeast Asia and Australia ²². The first event enabled the delineation of the Indo-Pacific biogeographic province via the breakup of the formerly global tropical sea belt (i.e., the Tethys Sea), which could naturally spur both speciation and extinction through large-scale vicariance. Also, the loss of western connectivity placed the IAA in the geographic center of vast Indo-Pacific oceans, where the overlapping of species distributional ranges could translate to a center of diversity (i.e., mid domain effect) ⁵¹. The second event established a shallow-marine connection between Southeast Asia and Australia for the first time and thus assembled a large expanse of shelf seas as suitable habitat for diversification. It could have also facilitated the convergence of peripheral faunas in central IAA, leading to an increase in regional diversity and changes in biogeographic patterns. Then, the 16-Ma peak coincided with the Mid-Miocene Climatic Optimum ⁴³, which is in line with our MBD results indicating both speciation and extinction rates accelerated with high temperature in a warm climate state (Fig. 2B). Similar diversification peaks at ~18 and ~15 Ma were also found in Caribbean bryozoans ⁹, which may correspond to our 20-Ma and 16-Ma events. The 12-Ma peak was apparent only in the speciation rate when the East Asian Monsoon intensified ⁵². A stronger regional environmental gradient, such as salinity via the enhanced monsoon and resulting precipitation, could provide a possible explanation. Finally, the 5-Ma speciation and extinction peaks may be related to the restriction of the Indonesian Throughflow at the Miocene-Pliocene boundary ^{41,53}, which allowed reversed faunal dispersal from the Indian Ocean to the Pacific Ocean, as discussed in Ref. ³. In summary, abrupt transitions in Earth's physical environments likely introduced short-term instability into the biotic system. For each of the biotic events, extinctions were overcompensated with much stronger speciation to support overall diversification, indicating the role of the IAA as a reservoir for species accumulation and a source of species origination."

In summary, our new results show that the drivers of long-term diversification may largely explain the short-term events. The peaks at ~25 and 20 Ma were associated with increased shelf habitats in the IAA, and the ~16 Ma peak corresponded to a hyperthermal event in warm climates. The remaining peaks at ~12 and 5 Ma were also associated with major paleoclimatic and paleoceanographic transitions that are known to trigger important biotic changes. However, these transitions occurred in a discrete, spontaneous, and sporadic manner, which makes it infeasible to include them into time-series models, at least by currently available technique.

Regarding the discrepancy mentioned by the reviewer, we think that the closure of the Tethys Seaway was not equivalent to habitat fragmentation within the IAA, in terms of their respective effects on diversification. The Tethys closure has long been proposed as a key process driving the eastward hopping of historical biodiversity hotspots, and thus

the formation of the modern hotspot in the IAA (Renema et al., 2008). Our analysis provides strong evidence for this hypothesis, by showing the strongest peak of diversification at ~20 Ma. The breakup of the Tethys shaped the modern patterns of tropical biogeography and biodiversity in deep time (Hou & Li, 2018). It led to the delineation of the Indo-Pacific realm by forming the largest hard barrier in marine tropics, and fundamentally influenced the Cenozoic evolution of many marine lineages (Bribiesca-Contreras et al., 2019; Cowman & Bellwood, 2013a, 2013b). At the regional scale of the IAA, however, increased habitat complexity may not lead to absolute geographical isolation for allopatric speciation. There was no major hard (land) barrier within the IAA, and the complex array of small marine basins were mostly separated by permeable soft barriers, which constantly changed in response to tectonic movements, sea-level fluctuations, and prevalent ocean currents (Bellwood et al., 2012). It has been questioned before if such geographical isolation within the IAA was strong or long enough to actually promote speciation (Bellwood et al., 2012; Yasuhara et al., 2022). The results of our study indicate that it does not play as a major driver. Instead, ecological speciation may be the predominant process as discussed in our manuscript.

Finally, we performed additional sensitivity analyses as requested to see the effects of binning on the diversification patterns. The revised text is at Line 795 and as follows: “To explore the sensitivity of the speciation and extinction estimates, we repeated all the BDCS analyses for the Neogene period only with time bins of 2 Myr, for the entire Cenozoic with time bins of 5 Myr, and for the Neogene period only with time bins of 5 Myr. All these analyses were parameterized following the main analysis. All results corroborate a reasonably robust and unambiguous pattern of strong and continuous diversification throughout the studied intervals (Extended Data Figs. 7-9).”

6. Line 324: what is slightly depauperate mean? Can this be quantified? Also, I suggest the authors provide quantification of what is “much higher richness” in this context.

Response: We rephrased the relevant sentences at Line 378 to provide quantification of the evolutionary trajectories of target groups as follows:

“From the Eocene to early Miocene, the Tethyan genera displayed a strong increase in diversity (2 to 14 species) and gained dominance over the other groups. During the middle-late Miocene, all three groups showed substantial proliferation with the cosmopolitan genera being more diverse than the Tethyan and IAA genera (75 compared with 46 and 20, respectively). The long-term trend of diversification proceeded to the Pliocene for the cosmopolitan and IAA groups with remarkable increases in diversity (156 and 28, respectively), while the Tethyan genera declined (42 species).”

7. Line 620: I was a bit concerned to read that the author 'tentatively assume' that all aggregated samples are representative of the IAA. Why is this tentative, and how did they determine provenance?

Response: Although our Cenozoic IAA dataset is comprehensive, we are inclined to be cautious and conservative when evaluating the sampling completeness, since there can never be perfect sampling coverage of any region. The Eocene and Oligocene materials are rarely preserved in the IAA, so the low sampling coverage of these periods is unavoidable. Since the Neogene, our samples have reasonably good spatial coverage from the central Philippines to Indonesia, where the modern center of diversity lies. **Please see our response to Comment No. 4 for the revised text and figure.**

8. It seems some of the environmental parameters used to explain diversification dynamics, such as temperature estimates, were not estimated from the IAA region (or roughly equivalent regions), but rather taken from global averages (using isotopic proxies). This seems problematic as the authors are not trying to describe some global signal, but rather the dynamics within a particular region. I would think the authors should measure spatially-explicit parameters (perhaps from GCMs, as they do the paleogeography) within their study region, to get at the true drivers of this regional signal. I would imagine that they would not find temperature to be significant if they did this.

Response: Thank you for bringing up this point. We tend to think it is more problematic with climatic modelling, because what we need is basically a time series temperature record. Using climatic models to perform a time series for any particular region with a spatial constraint, i.e., the Coral Triangle region only, is probably too risky. Climatic models are not perfect, there are so many models vs. proxy inconsistency in different time slices in different ways. And they are better for showing spatial patterns in a time slice. In addition, ostracods are coastal benthos. If we use pelagic plankton with wide spatial distribution throughout open ocean areas that are covered well by climatic modelling spatial grids (such as planktic foraminifera as recent *Nature* paper Fenton et al. (2023) did), it could be better justified as an ambiguous but important first step toward the challenging quantification of 4D diversity dynamics in long Earth history. But even another *Nature* study using planktonic foraminifera (Woodhouse et al., 2023) has adopted a much cautious approach, i.e., comparing with global climatic change (based on oxygen isotope stack) and not with climatic modelling, probably partly because of the above-mentioned modelling issues. It is challenging to use climatic modelling even for planktonic organisms. Additionally, our model organism is coastal benthos, and their distribution is much narrower, i.e., continental shelf along coastlines. Therefore, we think climatic models will not accurately resolve such a case.

We further think our approach with a global temperature record is cautious, assuming it reflects the regional temperature trend in the relative sense as the first

approximation. We would like to move forward to a quantitative approach, and this is the reason why we took a Bayesian model selection approach incorporating as many drivers as possible. We believe this is an important first step. But including regionally constrained parameters such as temperature from climatic modelling may be too bold as of today and it comes with high risk of misleading to wrong direction.

At the same time, we totally understand your point. As a doable alternative, we performed two sensitivity tests using a published Cenozoic record of tropical sea surface temperature from Scotese et al. (2021), which is supposed to more closely resemble the actual temperature of the IAA compared to the global average. The results from these MBD analyses remained the same qualitatively. We included the new MBD results in the section of *Sensitivity tests* at Line 803:

“For disentangling the paleoenvironmental controls on diversification dynamics, we reckoned that the use of global temperature in a regional analysis of the IAA could pose certain limitations on our modelling, together with the Oligocene sampling gap. We ran two sensitivity MBD analyses using the tropical temperature record from Ref. ⁷⁸ for the Cenozoic and for the Neogene instead of the global temperature. With all other parameters unchanged, the results remain consistent with those from the main analysis, still showing that diversity dependence and habitat size are two determinants of speciation (Extended Data Tables 6-7).”

9. *The ‘hopping hotspots’ model should be introduced briefly.*

Response: We revised the Introduction at Line 78 to add more details about the *Hopping Hotspots* model as follows:

“The current knowledge of the fossil record suggests that the locations of peak diversity (=biodiversity hotspot) shifted throughout the Cenozoic from the western Tethys during the Eocene to the Arabian Peninsula during the late Eocene-Oligocene, before being established at the current location of the IAA in Southeast Asia in the early Miocene: the process known as the *Hopping Hotspots* model ^{3,10}. Plate tectonics is postulated to be the ultimate driver of this process by regulating the broad-scale availability and configuration of shallow-marine habitats with successive continent collisions ¹⁰. Each hopping of the biodiversity hotspots from the ancient location to the new one was likely underpinned by considerable speciation and extinction events, but also could be associated with the paleobiogeographic shifts of some component taxa tracking suitable habitats ³.”

10. *There were some areas of the text that were unclear. For example, I would have liked to see more information on the raw analyses of the ostracod dataset – what sort of species range plots were created? I didn’t find this in the methods.*

Response: We added the following sentence at Line 816 in the Materials and Methods for clarification:

“We illustrated the stratigraphic ranges for all ostracod taxa by connecting the earliest and latest occurrence of every taxon.”

11. Line 161: what is a substantial degree of diversity dependence? Can this be quantified? Did you fit different models to test whether the curve most closely approximates a logistic function, or was this eyeballed?

Response: We fitted the multivariate birth-death (MBD) model to the data including self-diversity, and other abiotic variables that can explain the variations in speciation and extinction through time (Lehtonen et al., 2017). In the MBD model, we assume that speciation and/or extinction can correlate with multiple variables at the same time. The significance of the correlation is estimated with the shrinkage weight (ω) being greater than 0.5 (closer to 1 means strong effect). For speciation rate, we found a significant ($\omega\lambda_{\text{Diversity}} = 0.863$) and strong negative diversity dependence as measured by $G\lambda_{\text{Diversity}} = -2.735$ (95% HPD: -4.2589, -0.4659). These results mean that adding more ostracod species in the IAA tends to slow down speciation rate (Extended Data Table 2), which brings support for the hypothesis of diversity dependence.

In addition, our time-stratified MBD analyses also showed strong and significant effects of diversity dependence for the warm phase ($\omega\mu_{\text{Diversity}} = 1$; $G\mu_{\text{Diversity}} = 336.168$) and cold phase ($\omega\lambda_{\text{Diversity}} = 0.912$; $G\lambda_{\text{Diversity}} = -3.341$) (Extended Data Tables 3 and 5). These results further reinforced our interpretation of diversity dependency in driving the IAA ostracod diversification.

12. The authors state (line 166) that lineages and communities may commonly experience a time decay in their diversification rates as the growth of diversity itself becomes a limiting factor, however, it wasn't clear how this was measured, nor what the actual time delay is, on average, and how this varies across taxa. Furthermore, when did this time delay first become noticeable as diversification increased? Can this be tested statistically?

Response: In this context, “time decay” means that rates of diversification slow down with time, because adding more species into the system leads to more intense competition and inhibits ensuing diversification. From our results, there is a clear long-term trend of decreasing rates of diversification since the early Miocene, despite short-term fluctuations. This is not speculation but rather a quantitative result obtained from our Bayesian inferences. We further attributed this slowdown of speciation rates to a diversity-dependent effect as measured by the corresponding correlation parameter in the MBD model, with $G\lambda_{\text{Diversity}} = -2.735$. This slow down coupled with damped increase in diversity is consistent with a model of diversity-dependent diversification (Cornell, 2013; Rabosky, 2013). Knowing that clade’s diversity and other abiotic factors acted in

concert as drivers of ostracod diversification (Extended Data Table 2), we were able to tease apart the role of each factor with the MBD model. Unfortunately, at this scale, we are unable to discuss how this varies across taxa as it would imply running many analyses at the level of genus or ecologically similar subclades (Condamine et al., 2021; Condamine et al., 2019). For this first assessment, we treated all taxa as a whole regional entity when estimating the diversification dynamics, which has been done previously in several case studies (Condamine et al., 2020; Lehtonen et al., 2017; Neubauer et al., 2022).

To increase the clarity of our meaning, we rephrased the corresponding sentence at Line 218 and as follows:

“Indeed, lineages and communities may commonly experience a slowdown in their diversification rates coupled with damped increase in diversity.”

13. Line 177: The authors infer that diversity dependence causes the steady state in diversity seen post 2.6 Ma. While this may be true, I wondered if the authors couldn't test this in a more rigorous way, by running a model with diversity included as a predictor, alongside other key predictors such as climate. As the authors will be aware, there were significant climatic and oceanographic changes around 2.6 Ma, and I would think these abiotic changes could also explain changes in diversification dynamics at this time. Edit: this seems to have been done (with both abiotic and biotic factors in the same framework) but is not made particularly clear throughout the text and should be revised accordingly (both main text and methods).

Response: We revised the first sentence of the MBD result paragraph at Line 206 as follows:

“We deciphered potential drivers of IAA’s long-term diversity dynamics by simultaneously assessing the effects of key biotic (diversity dependency) and abiotic (habitat size and complexity, temperature, and sea level) factors, using a multivariate birth-death (MBD) model spanning the entire time frames (i.e., Eocene-Present).”

We agree that significant climatic and paleoceanographic changes occurring at around 2.6 Ma could affect diversification dynamics for several reasons. However, we also think a static stage (diversity plateau) in this case is more likely to be the result of diversity-dependent diversification, in which both speciation and extinction rates dropped to baseline levels so that the net diversification reached zero. Environmental changes on the other hand tend to trigger fluctuations in the biotic system instead of maintaining stability.

14. What is known about what structures ostracod speciation dynamics? Does speciation occur primarily allopatrically, and has anyone studied drivers previously? This information

would potentially provide useful context for the diversification studied here.

Response: This is a very good point. Cronin intensively studied ostracod speciation dynamics in the 1980s (Cronin, 1987, 1988; Cronin & Schmidt, 1988), and found that allopatry is not the dominant mechanism compared with sympatry and parapatry, as geographic isolation did not lead to speciation in many cases. Accordingly, we added the following discussion at Line 264 to emphasize the prevalence of sympatric speciation not only in ostracods, but also commonly in other marine taxa:

“Evidence of sympatric speciation is prevalent among marine clades of all different dispersal abilities and life histories, ranging from fishes, corals, gastropods, to ostracods^{34,36,37}, which potentially highlights the importance of ecological factors in marine diversification.”

15. Line 188: it is not clear whether the authors found temperature to be most important in diversification, or whether this is a general statement derived from past knowledge. If the latter, there should be citations provided for previous work on ostracods.

Response: We re-ran the Bayesian modelling with updates as requested by reviewers and showed that temperature effect is not significant for the entire time frame while significant for the warm phase. We explained the negative effect of temperature on diversification in warm climates at Line 305 and as follows:

“Most interestingly, high temperature triggered both speciation ($G=0.584$, $\omega=0.993$) and extinction ($G=5.038$, $\omega=0.999$) during the warm phase, which may be due to the coexistence of warm-adapted and cold-adapted species. As Earth’s climate showed a long-term cooling trend across the Cenozoic⁴², some warm-adapted species that were resilient to or even preferred very high temperatures of the Eocene and Oligocene might not yet be extinct in the relatively warm early-middle Miocene⁴³. They may have responded to temperature positively with more active proliferation. Cold-adapted species on the other hand gradually rose throughout the Neogene as the climate further cooled^{4,43,44}, but their vulnerability to high temperature may put them in much higher extinction risks during the relative warm early-middle Miocene. Indeed, recent studies suggest that tropical temperatures $>25^{\circ}\text{C}$ may be too elevated for marine organisms due to metabolic trade-offs driven by temperature effects on hypoxia sensitivity^{25,45-47}.

Consistent evidence for thermal stress on tropical biodiversity comes from various marine taxa (e.g., coral, reef fish, reptile, calcareous algae) that declined in diversity in the tropics and exhibited poleward range shifts during historical warm intervals^{48,49}. Since temperature has a much stronger impact on extinction than on speciation ($G\lambda=0.58$, $G\mu=5.04$), our analysis suggests that too high a tropical temperature in warm climates slows down diversification and hinders the growth of the hotspot. The effects of temperature are not significant for other time windows, suggesting that thermal stress may be relieved in cooler climates. We need to be aware of future extinction risks and tropical biodiversity loss exacerbated by anthropogenic warming under high

greenhouse gas emission scenarios, as tropical biotas are already living close to their ecophysiological thresholds ⁴⁶.”

16. The authors found a positive relationship between temperature and tropical extinction and an inverse relationship between temperature and tropical speciation. This is interesting, and the explanation is that it may have been too hot for species during greenhouse climates. I wondered if the authors considered the actual estimates of temperature (from proxy data or GCMs) for these intervals and this region, compared to the known temperature thresholds that may prove problematic for species, and, therefore, whether this explanation is actually reasonable? My concern is that the authors did not actually examine temperatures within this region. The methodology of using global estimates is problematic, and, after learning this, I was not surprised to learn that the authors found a pattern that contradicts previous work, including their own, using this methodology. I would be very curious to know whether these results hold when measuring temperature from low-latitude regions only (or the IAA only) – I would expect potentially not. Regardless of these issues, the entire discussion of these results is a bit muddled and difficult to follow, at present.

Response: As the reviewer suggested, we agree that the use of global temperature in a regional study must be handled with caution. **Please see our response to Comment No. 8 for details.**

Furthermore, our time-stratified MBD analysis indicated the warm phase of the Neogene (23.04-13.9 Ma) was too hot for tropical marine biodiversity as temperature had a strong positive effect on extinction during this interval (Extended Data Table 3). According to Scotese et al. (2021), estimates of tropical temperature for this interval were 25.92-28.32 °C, with an average value of 27.06 °C. Such high temperatures indeed exceeded the known threshold of 25 °C (Boag et al., 2021), and were far beyond the optimal value of 20 °C for marine organisms (Costello et al., 2023). This further supported our explanation of the detrimental effects of high temperatures. **Please see our response to Comment No. 15.**

17. It's interesting that the authors found that habitat complexity does not correlate positively with diversification, counter to previous hypotheses. They propose a valid explanation for this surprising pattern. However, I wondered whether this relationship could have been found because of the focus on ostracods, whereas other taxa may be influenced more by environmental fragmentation. To me, this is one of the most interesting and important results from this study, and I am surprised that this lack of relationship between diversity and suitable habitat complexity is not trumpeted more as a key finding.

Response: Thank you for your interest in our findings. We think our results about habitat complexity may be applicable to other taxa instead of being an ostracod-specific phenomenon. Compared with planktonic organisms and benthic organisms with a

planktonic larval stage, the dispersal capacity of ostracod is limited because its life mode is completely benthic. In theory, such brooding (direct developer) organisms have better chances for fast speciation as the result of geographic isolation (Hoeksema, 2007). The fact that ostracods are not sensitive to habitat complexity seemingly implies that other highly-dispersal organisms may be even less sensitive. So, for overall marine biodiversity, the effects of habitat complexity may not be as important as previously thought.

We expanded our discussion about habitat complexity at Line 251 and as follows: “The effects of habitat complexity on regional diversification are intriguing and worthy of further investigation. Indeed, in the marine realm, dispersal (both positively and negatively) as the dominant process may overwhelm the effects of isolation at a regional scale, where the physical barriers are often permeable and relatively sparse³³⁻³⁵. Connectivity among marine basins may be substantially restricted because of a dynamic mosaic of ever-changing geographic and oceanographic conditions within the IAA, but not strictly or permanently blocked, and thus undermines the effectiveness of conventional allopatry. On the other hand, our findings may imply the importance of sympatric speciation, given the positive effect of habitat size whereas no effect of habitat complexity^{34,36}. In large, continuous shelf habitats across wide environmental gradients, such as the IAA, diversification may occur along ecological partitions (i.e., intense competition invokes finer subdivision and niche specialization) in addition to geographic partitions^{28,34-36}. Evidence of sympatric speciation is prevalent among marine clades of all different dispersal abilities and life histories, ranging from fishes, corals, gastropods, to ostracods^{34,36,37}, which potentially highlights the importance of ecological factors in marine diversification.”

18. Overall, I think this is an interesting paper with some important analyses, but the key findings and message are difficult to disentangle from the text. I would suggest simplification and streamlining. As an example, see my previous comment above. As a further example, the authors generally found low extinction, which likely contributes to the high diversity in the region, which should likely be highlighted more clearly in the manuscript.

Response: We appreciate your positive comments. We substantially revised the structure of our Results and Discussion for easier understanding. We moved the low extinction paragraph into the *Cenozoic diversity history of the IAA hotspot* section at Line 184 for clarity and emphasis. We also specifically mentioned the lack of a sudden (massive) extinction as one of our key findings in the Abstract and conclusion.

More minor comments:

Line 46: suggest that ‘history’ and ‘explicitly’ are not needed here.

Response: We deleted them accordingly.

Line 49: suggest 'a' versus 'the diversity...'

Response: We changed accordingly.

Line 54: It was unclear to me what 'per se' meant in this context. I believe you mean diversity itself drives diversity (as in diversity dependence), and 'itself' may be a better word here if so.

Response: We replaced 'diversity per se' with 'diversity dependency'.

Line 77: remove 'and' before 'which'

Response: We deleted it accordingly.

Line 79: Is it the most diverse place on the planet, or is it one of the most diverse in the marine realm?

Response: We replaced 'on our planet' with 'in marine realm', as the terrestrial realm is out of the topic of this study.

Line 94: There is an extra space between time periods.

Response: We deleted the extra space.

Line 119: please insert an 'and' before the '2'

Response: We revised accordingly.

Line 144: This sentence is not complete and should not start with 'But'

**Response: We deleted the 'But' and revised the text to be:
"Nevertheless, it should be noted that..."**

Line 272: Please consider revising and better introducing the start of this sentence.

Response: This paragraph about short-term biotic events has been substantially revised. It is now in the section of *Diversification in relation to key biogeographic events* at Line 330. We hope you find the new paragraph clearer in structure and easier to follow.

Line 602: Please fix the question marks and incomplete sentence and explanation here.

Response: We fixed this.

Referee #2 (Remarks to the Author):

19. Thank you for submitting this fantastic study detailing the evolution of one of the most important biodiversity areas of the modern world. I think the results of this paper are exciting in showing how the interactions of different biotic and abiotic variables shape the biodiversity patterns we see around us, and how biodiversity shifts longitudinally and latitudinally through time.

Overall, as you will see from my comments, I suggest primarily minor changes to the text and figures to improve the clarity for Nature's broad viewership. However, I also suggest that some methodology of subsampling/sampling standardization be carried out due the lower sample coverage deeper in time.

I recommend that this be published in Nature after moderate revisions which I have detailed in the attached PDF. I very much look forward to this being added to the literature.

Response: Thank you very much for your positive comments. We found them very helpful and enlightening. We revised all relevant places in the manuscript and below we address the comments point by point.

20. Lines 69-70 – *Why do regionally resolved Cenozoic diversity trends remain poorly understood? Taphonomic biases? Preservation potential? Poor sampling?*

Response: Indeed, all above factors significantly limit our understanding of regional diversity patterns during the Cenozoic. Taphonomic and preservation factors are certainly very important. The Paleogene materials are rarely preserved in the tropical IAA region in shallow-marine settings, and they are often hard limestone if existed, which is unsuitable for microfossil research (ostracods cannot be studied with thin sections and they cannot be disintegrated from the limestone without being dissolved). Even more important reasons may be low sampling efforts and difficulties in data compilation for many taxonomic groups, for example corals and mollusks (Johnson, Hasibuan, et al., 2015; Johnson, Renema, et al., 2015). Although global databases of fossil occurrences are available for these taxa, there may not be enough sampling coverage at a regional scale for the duration of interest. To address this comment, we revised the corresponding text at Line 75 to be:

“However, regionally resolved Cenozoic diversity trends remain poorly understood due to the scarcity of historical data and their compilation. This is particularly true for the tropics in deeper time, making the origins of high biodiversity an enigma⁷⁻⁹.”

21. Line 85 – *What are the other taxonomic groups and what are their limitations?*

Response: The other taxonomic groups used to describe the IAA hotspot range from corals, coral reef fishes, mollusks, and larger benthic foraminifera. We revised and supplemented the Introduction at Line 98 and as follows:

“Historical evidence from benthic taxonomic groups has advanced our understanding of tropical diversification yet suffers from various limitations. Recent molecular studies on corals and reef fishes revealed their biogeographic and evolutionary history to build the IAA hotspot, suggesting it was a center of species accumulation and origin at different intervals of the Cenozoic ^{12,13}. But these phylogeny-based models are subject to uncertainties in the case of incomplete sampling and undocumented extinctions. They also are not geographically defined to quantify a clear scenario of regional diversity changes in the IAA. In addition to molecular evidence, previous paleontological studies pieced together indicated a compatible trend of increased IAA species richness at a coarse spatio-temporal resolution, which was dominated by immigration into the IAA in the Eocene-Oligocene and proliferation inside the IAA in the Oligocene-Recent ^{7,10,14-16}. However, difficulties in reconstructing a more detailed IAA history for benthic fossil groups include: a lack of sufficient fossil data or data synthesis for mollusks and bryozoans ^{9,16}; a small species pool despite a good fossil record for larger benthic foraminifera ⁷; and uncertainties in species-level identification for fossil corals ⁸.”

22. Line 99 – What is the record in the IAA like for planktonic organisms?

Response: This is a very good point. We know that marine biodiversity hotspots (both the IAA and Caribbean) are primarily shaped by high species richness of coastal benthos in specific regions. On the contrary, oceanic groups tend to have large distributional ranges in all oceans without forming a distinct peak of diversity across the longitudinal gradient (Tittensor et al., 2010). Planktic foraminifera for example, as one of the most studied planktic groups, does not show particularly high diversity in the IAA (Yasuhara et al., 2020). We revised the text at Line 95 to emphasize this:

“As one of the most conspicuous biogeographic and biodiversity patterns today, the IAA hotspot is characterized by an exceptional concentration of coastal benthic species in this region, while pelagic groups show widespread distributions without a distinguished center of diversity ^{2,11}.”

23. Line 108 - Perhaps state earlier on that this is a study that is focused on benthic organisms (though fish are mentioned a lot), explain why planktonic organisms (generally) are not being used in this study

Response: **Please see our response to Comment No. 22 above.**

In addition, fossil records of planktonic organisms are very scarce in this region for the studied time period, and planktonic organisms are far less important components of the IAA hotspot compared with coastal benthos. Also, all fishes mentioned in this manuscript refer to coral reef fish instead of pelagic fish. We revised all relevant places

in the manuscript to make this clear and specific. So, we think it makes sense to discuss the patterns of reef fishes here together with other benthic organisms, since they share similar life modes.

24. Line 147 – It may be more appropriate to have "Pleistocene" on the figures instead of "Quaternary" as Pleistocene is used more in the text

Response: We used ‘Pleistocene’ in all the figures now.

25. Line 149 – Very interesting!

Response: We are so glad that you like this.

26. Line 156 - These are not strictly "stages" in the chronostratigraphic context. Interval or time bin may be better to avoid confusion

Response: We changed ‘stage’ to ‘interval’ accordingly.

27. Lines 167-173 – Great stuff there!

Response: Thank you very much for your recognition of our study.

28. Lines 180-181 - Or another plateau for the time being? What is our understanding of ostracodal cryptic species?

Response: We totally agree that the plateau between 2.6-0 Ma could be just for the time being. The potential of further diversity increases still lies given significant changes in biotic or abiotic conditions allowing the carrying capacity to be overcome. We added the following text at Line 231 to discuss such possibilities:

“This may represent the maturation of the IAA hotspot after a long Neogene history of expansion, but the possibility of future diversity growth remains if the carrying capacity increases in response to changing environmental conditions, or if a key innovation evolves to conquer new niche space^{28,29}, which would make the current phase only a diversity plateau in the hotspot’s lifecycle.”

Regarding cryptic species, recent molecular studies have revealed a considerable level of cryptic speciation in freshwater ostracods (Karanovic et al., 2020; Schön et al., 2017; Schön et al., 2012), but it is not the case in marine ostracods. Discoveries of cryptic species are rarely made, and available evidence indicated a nearly perfect correspondence between morphologically identified species and genetically defined species in shallow-marine ecosystems (Estronza et al., 2017). In general, current morphological identification of marine ostracod based on fossilized valves is considered as robust and appropriate for species recognition and discrimination (Yasuhara et al.,

2017). We think that the presence of undetected cryptic species in our study, if there were any, would not significantly change our results. At the same time, we agree that cryptic species and the definition of species in general are very important to be considered in any biodiversity study, so we specified the use of ‘morphospecies’ at Line 151.

29. Lines 189-196 – Interesting!

Response: Thank you very much for the positive comment here.

30. Line 203 - Briefly summarise these other mechanisms if possible. Also include Woodhouse & Swain et al. (2023), these 3 articles represent a unified theory

Woodhouse, A., Swain, A., Fagan, W.F. et al. Late Cenozoic cooling restructured global marine plankton communities. *Nature* 614, 713–718 (2023). <https://doi.org/10.1038/s41586-023-05694-5>

Response: Thank you very much for recommending these excellent papers. As summarized in the beginning of our reply letter, with the newest age model, we found that temperature significantly slowed down diversification during warm climates only. We think that other mechanisms of temperature control may not be necessary for our story now. **Instead, please see the revised text under Comment No. 15 explaining the negative effect of temperature on diversification in warm climates.**

31. Lines 208-212 - What is the sampling coverage of the tropics during the Cenozoic? Though I agree with your interpretations, be sure to highlight potential sampling biases

Response: We totally agreed that sampling coverage should be mentioned in detail. **Please see our response to Comment No. 4 for the revised text and figure regarding spatial coverage of our samples.**

32. Line 222 – Put southern first so it is in order of glaciation

Response: After re-running the MBD analysis with the updated age model, we showed that sea level was not a significant driver of diversification for the entire time frame, although it was significant for the warm and cold phases. We thus discussed the effects of sea level in detail for the cold phase at Line 288:

“Higher sea level also expedited speciation during this interval ($G=0.022$, $\omega=0.819$), likely through the species-area relationship. Large and frequent fluctuations in sea level associated with the glacial cycles may strongly impact the expansion and contraction of epicontinental seas and consequently the size of shallow-marine habitats^{39,40}.”

And for the warm phase at Line 301:

“Higher sea level reduced extinction as expected ($G=-0.843$, $\omega=0.999$), while it also reduced speciation, albeit weaker ($G=-0.069$, $\omega=0.995$). A possible explanation is that Southeast Asia was still separated from the Australian and Pacific plates by deep oceans^{22,41}, so high sea level may not contribute to form large areas of shelf habitats in the IAA.”

33. Line 237 - They are certainly MORE sparse compared to the terrestrial realm, but there are many of them. See:

Norris, R.D., 2000. Pelagic species diversity, biogeography, and evolution. Paleobiology, 26(sp4), pp.236-258.

Response: Thank you for recommending this paper. Indeed, it supports our interpretation of ecological speciation, and we now cite it accordingly. We realized that there are still many physical barriers of different types in the marine realm, including hard, soft, and intermittent ones (Bowen et al., 2013). However, the hard barriers (land masses) are still relatively sparse, especially in the Indo-Pacific region without a major land bridge (Hall, 2009). Soft barriers such as ocean currents and deepwater passages are more permeable, which do not completely cut off dispersal at local or regional scales (Bowen et al., 2013).

Please refer to our reply to Comment No. 17 for the revisions.

34. Lines 243-247 – Awesome

Response: We are so glad you like this.

35. Lines 261-264 - Why may they have not moved prior to this?

Response: This is a really good question. Earlier movement of the Tethyan taxa during the Oligocene and earliest Miocene indeed occurred, but most likely in a slow and occasional way, as indicated by fossil evidence from reef fishes, corals, and larger benthic foraminifera (Renema et al., 2008). Suitable shallow-marine habitats had only been in place in the IAA until 25-20 Ma, when the Southeast Asia collided with Australia (Crame & Rosen, 2002; Leprieur et al., 2016). Therefore, the migration of large numbers of Tethyan taxa to the IAA may primarily occur during ~25-20 Ma. In addition, what we tried to emphasize in the manuscript is that the closure of the Tethys Seaway geometrically constrained the Indo-Pacific domain in the west, and such mid-domain effect may contribute to the increase of diversity in the IAA. We revised the corresponding text at Line 337 as follows:

“The second and the largest peak in both speciation and extinction occurred at 20 Ma. It correlated in time with two major tectonic events outside and within the IAA, namely

the closure of the Tethys Seaway separating the Indo-Pacific from the Mediterranean Sea and Atlantic Ocean ^{10,50}, and the closure of the Indonesian deep-water passage between Southeast Asia and Australia ²². The first event enabled the delineation of the Indo-Pacific biogeographic province via the breakup of the formerly global tropical sea belt (i.e., the Tethys Sea), which could naturally spur both speciation and extinction through large-scale vicariance. Also, the loss of western connectivity placed the IAA in the geographic center of vast Indo-Pacific oceans, where the overlapping of species distributional ranges could translate to a center of diversity (i.e., mid domain effect) ⁵¹. The second event established a shallow-marine connection between Southeast Asia and Australia for the first time and thus assembled a large expanse of shelf seas as suitable habitat for diversification. It could have also facilitated the convergence of peripheral faunas in central IAA, leading to an increase in regional diversity and changes in biogeographic patterns.”

36. Line 269 - Have the authors considered how much pseudospeciation and pseudoextinction is occurring at this time? This is a general question on the whole study, and may be worth considering at some point within the manuscript considering how much turnover there seems to be at 20, 16, and 5 Ma. Clearly more morphospecies are evolving, but how many are actually becoming truly extinct?

Response: This is a very illuminating comment. Unfortunately, at this point, we are unable to quantitatively evaluate pseudo-speciation and pseudo-extinction in the IAA, because doing so requires a comprehensive global dataset of the entire Cenozoic to estimate the time and place of the first/last occurrence of every species. Such a dataset is currently out of reach and may remain so for some time. On the other hand, the Bayesian birth-death model has been proved useful in regional-scale studies with a certain caveat that the estimated first and last appearance does not necessarily mean true speciation and global extinction respectively, see Tian et al. (2021) for example. As our focus is the IAA region itself, we are inclined to think that the pseudo-speciation and pseudo-extinction do not significantly interfere with our understanding of the growth of the hotspot, yet we agree this problem must be considered. For each of the peaks of diversification, speciation exceeded extinction. Extinction here may be true global extinction, or it may be the extirpation of species from the IAA (i.e., local extinction). In either case we see the ‘extinction’ leading to a decrease in diversity in the IAA. So, we think that our diversity estimates may reasonably reflect the actual diversity trajectory of the IAA region. We added the following text at Line 138 as a caveat:

“Given the regional scope of this study, speciation here corresponds to the first appearance of every species in the IAA to construct the emerging hotspot. The same rationale applies to extinction, which is defined as the final extirpation of any species from the IAA instead of global extinction.”

37. Line 293 - *More records may be needed to confirm whether it was global*

Response: We agree that existing evidence may not be strong enough to indicate the 5-Ma event as a global one. We revised the text at Line 358 as follows:

“Finally, the 5-Ma speciation and extinction peaks may be related to the restriction of the Indonesian Throughflow at the Miocene-Pliocene boundary ^{41,53}, which allowed reversed faunal dispersal from the Indian Ocean to the Pacific Ocean, as discussed in Ref. ³.”

38. Line 299 - *huge spatial differences?*

Response: Yes, we meant spatial differences between the IAA and Caribbean. We revised the sentence at Line 186 to be:

“Our results are in agreement with a previous study which showed that huge spatial differences in extinction rates between the IAA and Caribbean may be crucial for the modern longitudinal diversity patterns across the tropical belt, with the largest hotspot situated in the IAA in contrast to much lower diversity in the Caribbean ⁹.”

39. Lines 315-318 - *This is super interesting and super important - the world around us may be the exception, not the rule when it comes to understanding life*

Response: We really appreciate your recognition of our study.

40. Line 323 - *How are these defined? Are Tethyan genera those that first appeared in Tethys (prior to ~ 20 Ma), and cosmopolitan first appeared anywhere other than Tethys? What about genera that first appear specifically in the IAA?*

Response: The comparison of the IAA genera with the other groups is a great idea. We chose the five most abundant IAA genera that originated in central Indo-Pacific during the Neogene. They showed rapid diversification since the late Miocene. We specifically defined each group as asked by the reviewer, i.e., Tethyan group originated in broad Tethyan region during the Paleogene and cosmopolitan group exhibited global distributions during the Cenozoic.

New results are at Line 368 and also here:

“Among all paleoenvironmental events discussed above, the restriction and final closure of the Tethys Seaway during ~25–20 Ma is of particular interest. This process governed the paleobiogeographic shifts from the vanishing Tethys to the emerging IAA, which are hypothesized as an important factor in shaping the latter hotspot by the *Hopping Hotspots* model ¹⁰. Here we investigated how the migration of the Tethyan relicts may leave long-lasting implications on the diversity and biogeographic history of the IAA

hotspot. We focused on the five most diverse genera of the Tethyan group (originated in the broad Tethyan region during the Paleogene^{54,55}), cosmopolitan group (exhibited global distributions during the Cenozoic^{56,57}), and IAA group (originated in central Indo-Pacific region during the Neogene⁵⁶⁻⁵⁸), respectively, and compared their proliferation across the Cenozoic (Fig. 3; Extended Data Fig. 6). From the Eocene to early Miocene, the Tethyan genera displayed a strong increase in diversity (2 to 14 species) and gained dominance over the other groups. During the middle-late Miocene, all three groups showed substantial proliferation with the cosmopolitan genera being more diverse than the Tethyan and IAA genera (75 compared with 46 and 20, respectively). The long-term trend of diversification proceeded to the Pliocene for the cosmopolitan and IAA groups with remarkable increases in diversity (156 and 28, respectively), while the Tethyan genera declined (42 species). Finally, slight Pleistocene drops in diversity likely represented sampling artifacts instead of true extinction, at least for the cosmopolitan and IAA groups, and the cease of diversification corresponded to the maturation of the IAA hotspot. The evolution of three biogeographic groups closely mirrors that of the whole ostracod assemblage, as the average per-genus diversity reached value of 1.29, 2.62, and 4.34 in the Eocene, Miocene, and Pliocene, despite a slight decrease to 3.9 in the Pleistocene, indicating a pronounced diversification at species level throughout the Cenozoic (Fig. 4). The waxing and waning of these three groups clearly reflect a changing biogeographic affinity of the IAA with other regions. Prior to the final closure of the Tethys Seaway, the Tethyan ancestors were able to reach the IAA after a long-distance dispersal. They persisted and proliferated in the IAA even after the seaway closed, and thereby supported the early-stage development of the IAA hotspot. Since the middle Miocene, the IAA situated at the center of vast Indo-Pacific domain with high connectivity/dispersal may be more prone to accommodate drastically radiating cosmopolitan taxa from the Indian and Pacific Oceans, which then outcompeted the earlier occupants of the Tethyan genera. In addition, the endemic IAA genera were comparatively less diverse, possibly because of their youngest age; nevertheless, they also experienced a prominent diversification since the late Miocene. Their success may indicate the importance of tropical hotspots in generating and exporting biodiversity within a relatively short geological time. We conclude that the diversification of the IAA hotspot was accompanied by profound biogeographic changes, as the Tethyan taxa were gradually replaced by cosmopolitan and IAA taxa during the Neogene. Tethyan faunal elements from older, vanished hotspots did become evolutionary fuel for the emergence of the IAA, as proposed by the *Hopping Hotspots* model, but such a unique paleobiogeographic shift was not the sole mechanism that contributed to tropical diversification. In fact, the stepwise collision of Australia with Southeast Asia and the switching on-and-off of the Indonesian Throughflow could be both important processes that facilitated the convergence, accumulation, and eventually proliferation of taxa in the central IAA.”

And also more detailed information in the Materials and Methods at Line 819:

“Among all genera that were relatively abundant and speciose throughout the Cenozoic in the IAA, we identified typical Tethyan genera that originated in the broad Tethys

biogeographic area during the Paleogene (i.e., *Neomonoceratina* in the Paleocene, *Stigmatocythere*, *Pokornyyella*, *Keijella*, and *Tenedocythere* in the Eocene) ^{54,55,79-81}; cosmopolitan genera that had a broad geographical distribution across the Cenozoic (*Cytheropteron*, *Krithe*, *Argilloecia*, *Loxoconcha*, and *Xestoleberis*) ^{56,57}; and IAA genera that firstly appeared in central Indo-Pacific during the Neogene (i.e., *Pistocythereis*, *Lankacythere*, *Atjehella*, *Hemikrithe*, and *Hemicytheridea* in the Miocene) ^{56-58,82,83}. Finally, we compared the trends of their respective diversity changes per genus throughout the Cenozoic.”

Fig. 3. Sum of per-genus diversity of the five most diverse cosmopolitan, Tethyan, and IAA genera, respectively, during each interval of the Cenozoic. Eoc: Eocene; Oligo: Oligocene; earlyMio: early Miocene; midMio: middle Miocene; lateMio: late Miocene; Plio: Pliocene; Pleisto: Pleistocene. Silhouette of each ostracod genus made by Skye Yunshu Tian.

Fig. 4. Per-genus diversity of all ostracod genera during each interval of the Cenozoic. Five most diverse genera of the cosmopolitan, Tethyan, and IAA group are shown in the corresponding color used in Figure 3. Black dot indicates the mean per-genus diversity of all ostracod genera. Error bar shows standard deviation (If the lower limit is below zero, the minimum value is used instead). Abbreviations as in Fig. 3.

41. Line 340 - Is this what is shown in Figure 3 (left), it may be clearer to show Cosmopolitan, Tethyan and IAA genera individually if possible.

Response: Please see our response to Comment No. 40 for the new Figures 3 and 4. We also made another faunal diagram (Extended Data Figure 6) showing the diversification of each genus separately.

Extended data Figure 6. Species-level diversification of the five most diverse cosmopolitan, Tethyan, and IAA genera respectively across the Cenozoic. Dashed horizontal lines indicate chronostratigraphic boundary.

42. Lines 348-355 - Interesting!
 Lines 361-365 – This is great!

Response: Thank you very much for the positive comments.

43. Lines 369-370 - Immunity of ostracods or biodiversity as a whole?

Response: We think our ostracod results can be extrapolated to the broad benthic community at least in qualitative sense, because it has been shown that ostracod is a reliable proxy to reflect overall biodiversity and biogeographic patterns. We suggest that the IAA biota as a whole did not experience mass extinction during the Cenozoic. **Please refer to our reply to Comment No. 2 for the revisions.**

44. Line 376 - And latitudinal correct?

Response: We revised the concluding paragraph at Line 428 to better convey the take-home message:

“Our results suggest that the increase in habitat size and the immunity to major extinction events during the Cenozoic allowed for strong, uninterrupted long-term diversification that reached an asymptote in the IAA within millions of years, making it the current global center of biodiversity. The relief of thermal stress as Earth’s climate

transformed from a warm state to a cold state was also essential for this process. As we know, the opposite is rapidly happening, namely the current anthropogenic climate warming⁵⁹ and the ongoing sixth mass extinction^{46,60}. Both may effectively undermine the IAA hotspot and impair its function as a cradle and museum of tropical biodiversity. If we take this historical lesson seriously, we may rapidly lose the tropical hotspot as one of the most fascinating biodiversity and biogeographic patterns.”

45. Lines 593-595 - What biostratigraphic scheme was used? Is it calibrated to the GTS 2020?

Response: We updated all the ages to the Geological Time Scale 2020 (Raffi et al., 2020; Speijer et al., 2020) and re-ran all the analyses.

46. Line 615 – See my comments on Fig S1

Response: We made a new locality map showing the sampling sites and islands (Extended Data Figure 1). We also added the age range of every time bin in the caption of the diversity map (Extended Data Figure 4, **please see our reply to Comment No. 4**).

Extended data Figure 1. Map of the Indo-Australian Archipelago showing sampling location, with a magnification of the Philippine islands. The name of each sampling island is labelled in red.

47. Lines 622-623 - It would be good to integrate some form of subsampling or sampling standardization to check whether the amount of samples per time bin has an effect.

Response: Across all replicate of the dataset (a replicate is a random selection of occurrence age), we found high statistical support (significance at $P < 0.01$) for the non-homogeneous Poisson process of preservation (NHPP), which assumes a model in which

preservation rates change during the lifespan of each lineage following a bell-shaped distribution (Silvestro et al., 2014). The fossil dataset is of high quality compared to what has been published so far as estimated with the Bayesian inferences in PyRate through the preservation process (denoted q). In the current dataset, we estimated $q = 45.04$ occurrences per lineage per Myr [95% HPD: 44.2252, 46.0392]. We also estimated that the parameter α , which stands for the *Gamma* model of rate heterogeneity accounting for the heterogeneity in the preservation rate across lineages, is $\alpha = 0.501$ [95% HPD: 0.4933, 0.5101]. These results indicate high preservation rates of IAA ostracods that decrease exponentially with taxon age.

We also performed additional BD analyses to assess the effects of binning and sampling gap. We added the following at Line 795 in the Materials and Methods:

“To explore the sensitivity of the speciation and extinction estimates, we repeated all the BDCS analyses for the Neogene period only with time bins of 2 Myr, for the entire Cenozoic with time bins of 5 Myr, and for the Neogene period only with time bins of 5 Myr. All these analyses were parameterized following the main analysis. All results corroborate a reasonably robust and unambiguous pattern of strong and continuous diversification throughout the studied intervals (Extended Data Figs. 7-9).”

48. Lines 741-743 - Was there a specific biogeographic range which defined a cosmopolitan species as opposed to a more endemic one? - So does this analyses include all taxa that also originated in the IAA? It may be useful to plot this data too?

Response: Please refer to our reply to Comment No. 40.

49. Line 756 - Nice figure

Response: Thank you very much for your positive comment.

50. Line 759 - Is this cumulative diversity, including extinct species?

Response: Panel D in Figure 1 is the Bayesian estimates of ostracod diversity (number of species) throughout the Cenozoic. It is not cumulative diversity, instead showing only extant species at any given time. We revised the figure caption accordingly.

51. Supplementary Data File

Table S1 - A Period is distinct chronostratigraphic class of time, perhaps use "Interval" or "Time bin"

Response: We changed it to 'interval' accordingly.

52. What is the palaeoenvironment of each of these sample sites?

Response: Ostracod fauna in our samples generally indicated relatively shallow (continental shelf-uppermost continental slope) and fully marine conditions.

53. *Fig S1 - Is there anyway the sample sites could be identified? It may make this image too busy, but a blown out Map with sample sites, and country names may help those less familiar with IAA geography*

Response: Please refer to our reply to Comment No. 46.

54. *It would be beneficial to have the age ranges of each of these bins on this figure/in the caption*

Response: The age ranges have been added to the caption of Extended Data Figure 4.

Referee #3 (Remarks to the Author):

55. *The manuscript ‘Cenozoic history of the tropical marine biodiversity hotspot’ by Tian, Yasuhara, et al. is a solid manuscript that is well-written and suitable for publication in Nature. The authors use a new dataset of ostracod taxa from the Indonesian region across the Cenozoic to determine macroevolutionary processes that led to the IAA being a diversity hotspot. They found that the IAA region has exhibited an diversity trend through the Cenozoic, with four distinct peaks in speciation rate and net diversification rate, but only three peaks in extinction rates. Such peaks correspond to abiotic factors that have shaped the IAA to be a hotspot. Of distinct importance is that the absence of an mass extinctions during the Cenozoic has led to the IAA becoming a marine biodiversity hotspot. The methods used in the manuscript are appropriate for the analyses performed, and the resulting data are presented in very nice figures. The authors use statistics appropriately and report the values where necessary. The conclusions of the paper are robust, and in addition, this paper lies the groundwork and provides hypotheses for future researchers to delve into. My suggestions for improvement are below:*

Response: Thank you very much for your supportive and constructive comments, which really helped us to improve this manuscript. Below we carefully address each comment.

56. *Line 77: Change “and which might impair understanding” to “which impairs understanding”*

Response: We revised the sentence at Line 89 as follows:

“However, to uncover the long-term history of the IAA biodiversity, Cenozoic or Neogene fossil data are scarce and patchy in this region ^{3,9}.”

57. *Line 86: Could take out “have gradually revealed” to “reveal” to save some space*

Response: We changed accordingly.

58. Line 93: *This sentence could be reworded a bit for clarity to something like “... increased richness due to immigration into the IAA in the Eocene-Oligocene...”*

Response: We reworded the sentence at Line 105 as follows:

“In addition to molecular evidence, previous paleontological studies pieced together indicated a compatible trend of increased IAA species richness at a coarse spatio-temporal resolution, which was dominated by immigration into the IAA in the Eocene-Oligocene and proliferation inside the IAA in the Oligocene-Recent ^{7,10,14-16}.”

59. Lines 98–99: *In this sentence, you are pointing out why the biodiversity history of the IAA is limited for these groups, but for the benthic foraminifera, it’s not that their fossil record is poor but rather their species’ richness is dampened. This, to me, signals that something is going on with the large benthic foraminifera diversity in the IAA region that doesn’t match the results of your study. I think it’s okay to leave this statement about the larger benthics in, but it’s more of an interesting result rather than a limiting factor.*

Response: Thank you for this helpful comment. Indeed, modern distribution of larger benthic foraminifera (LBF) shows a clear diversity peak in the IAA (Förderer et al., 2018), but this group has a rather small species pool (~100 extant species worldwide). So, we think it may not be the best model proxy for a biodiversity study. We revised the sentence at Line 111 as follows:

“a small species pool despite a good fossil record for larger benthic foraminifera ⁷;”

60. Line 100: *Suggest changing “problem” to “limitation”*

Response: We changed it accordingly.

61. Line 115: *Change “a” to “the”, so “the first comprehensive Cenozoic fossil dataset...”*

Response: We changed it accordingly.

62. Line 119: *This is up to the authors, but I would suggest changing “extinction rates” to “extirpation rates” as you do note earlier in the sentence that this is a regional analysis, not global. Add an “and” before “(2)”.*

Response: This is a really good point, and we agree.

Please see our detailed reply and revisions under Comment No. 36.

63. Line 120: *access or assess? I think it’s supposed to be assess? I also suggest changing this sentence to be more clear to something like “...correlate the IAA’s macroevolutionary*

dynamics with a set of biotic and abiotic parameters to assess the potential biodiversity drivers”

Response: Thank you for pointing this out. We revised the sentence at Line 141 as suggested:

“We then correlated the IAA’s macroevolutionary dynamics with a set of biotic and abiotic parameters to assess the potential biodiversity drivers.”

64. Line 140: Capitalize Early, Middle, and Late Miocene here and throughout the manuscript.

Response: It is suggested that epochs in the Cenozoic, i.e., the early, middle and late do not have capital letters because they are not official geological epochs for the International Chart.

65. Lines 144–145: This sentence about not having Oligocene data should be moved after the sentence about low Eocene diversity, so it is not confusing to readers why the Oligocene is not explicitly mentioned as also having low diversity.

Response: Based on our new analysis results with updated age model, we revised the text at Line 160 to be:

“Species richness was very low during most of the Paleogene, consistent with the hypothesis that the IAA was not the place of a biodiversity hotspot during this period. With an initial increase in diversity in the late Oligocene, in line with the *Hopping Hotspots* model ¹⁰, rapid diversification began at ~25 Ma, making the IAA a rising hotspot.”

To show the lack of Oligocene data does not strongly affect our understanding of the overall diversification trend and its long-term drivers, please refer to our reply to Comment No. 3.

66. Line 148: The 600 species is referring to species richness rather than diversity, so suggest changing “diversity” in this sentence to “richness”

Response: We changed it accordingly.

67. Figure 1: Really great figures, very easy to comprehend and good info communicated.

Response: Thank you very much for your appreciation.

68. The only suggestion I have is to please plot A, B, and D with the same y-axes. At first look, it appears as though the magnitude of the origination and extinction rates are the same,

which led me to think these may be turnover events. Only later did I realize that the y-axes were not the same. Plotting the data on the same y-axis scale will give readers a better sense of how larger the origination rates are compared to the extinction rates.

Response: Thank you for your nice suggestion and we totally agree. We changed the y axes of panels A and B accordingly.

69. Table S2: Define HPD in the table caption.

Response: HPD stands for highest posterior density (also known as credibility interval for Bayesian inferences). This is now added in the caption.

70. Line 203: Either take out the sentence “Other mechanisms could also be possible”, or expand on this sentence a bit more to precisely state what other mechanisms could be possible.

Response: We deleted this sentence as suggested.

71. Line 214 and elsewhere: Is there a way to quantify ‘strong’ and ‘relatively strong’ when referring to the factors that influence speciation? If not, it would be helpful to define what the authors mean by ‘very strong’, ‘strong’, and ‘relatively strong’. Maybe include some sort of key or categories on Figure 2 to be more clear.

Response: In the multivariate birth-death model, the strength of the correlations is measured with the shrinkage weight (as denoted ω), with values of ω being greater than 0.5 indicating a significant effect of the variable on the diversification (Lehtonen et al., 2017). As we assume that speciation and/or extinction can be correlated with multiple variables at the same time, we rank the variables according to their ω values. For instance, diversity throughout the Cenozoic is the major factor of speciation variation as we found a $\omega\lambda_{\text{Diversity}} = 0.863$, stronger than the effect of habitat size ($\omega\lambda_{\text{Shelf}} = 0.647$). We now report the ω value for all variables in relevant places throughout the manuscript.

Please see the new Figure 2 under Comment No. 5.

72. Line 223: I’m not too certain what ‘its’ is referring to in this sentence. Please clarify.

Response: This section has been substantially revised following new modelling results. Sea level now is not a significant driver of diversification in the entire time frame analysis. We hope you find the new paragraph clearer in structure and easier to follow.

73. Line 226: Here, when referencing habitat size and complexity, it would be more helpful to the readers to either include “(shelf area)” and “(habitat size)” after these phrases, respectively, or include ‘habitat size’ and ‘complexity’ on the Fig. 2 y-axis. This can be simply overlooked in the figure caption.

Response: We specified ‘habitat size (shelf area)’ and ‘habitat complexity (coastline length)’ in the main text.

74. Lines 229-231: This is just a fun thought for the authors: Have you considered time-slicing the MBD analysis? The time slices could be different climate regimes pulled from CENOGRID by Westerhold et al. (2020), and may be a more refined way to test your hypotheses put forth in these sentences? It would be interesting to see if and how the MBD analyses change through time.

Response: Thank you very much for this helpful suggestion. We performed a time-stratified MBD analysis by dissecting the Neogene into three time windows: the warm phase 23.04-13.9 Ma, cooling phase 13.9-5.33 Ma, and cold phase 5.33-0 Ma. Such a division scheme reflected the climate regimes of Westerhold et al. (2020) and also fitted model properties of MBD analysis (because dividing too short time slice, e.g., 3.3-0 Ma, affected model performance). The results indicated time-varying effects of diversity, temperature, habitat size, and sea level, which we discussed in a new paragraph at Line 281 and as follows:

“After establishing the general long-term drivers of the IAA’s diversification dynamics, we then scrutinized how their effects may vary over time across different climate regimes³⁸. A time-stratified MBD analysis revealed further details for the warm phase (23.04-13.9 Ma), cooling phase (13.9-5.33 Ma), and cold phase (5.33-0 Ma) of the Neogene period (Fig. 2B-D; Extended Data Tables 3-5). The results of the cold phase are highly concordant with those of the entire time frame, with diversity ($G=-3.341$, $\omega=0.912$) and habitat size ($G=2.445E-7$, $\omega=0.702$) negatively and positively affecting speciation, respectively (Fig. 2D; Extended Data Table 5). Higher sea level also expedited speciation during this interval ($G=0.022$, $\omega=0.819$), likely through the species-area relationship. Large and frequent fluctuations in sea level associated with the glacial cycles may strongly impact the expansion and contraction of epicontinental seas and consequently the size of shallow-marine habitats^{39,40}. There is no correlation between diversification dynamics and any biotic or abiotic factors for the cooling phase (Fig. 2C; Extended Data Table 4), which may indicate it as a transitional stage between two opposite climate regimes. Indeed, during the warm phase, biotic and abiotic controls on the diversification of IAA faunas occurred in very different ways from those of the cold phase (Fig. 2B; Extended Data Table 3). The strong positive effect of diversity on extinction conforms to our understanding of diversity dependency ($G=336.168$, $\omega=1$), but its positive effect on speciation ($G=25.093$, $\omega=0.999$) may be explained by higher availability of empty niches when there was much less species, as the IAA hotspot just originated during the early Miocene. Higher sea level reduced extinction as expected ($G=-0.843$, $\omega=0.999$), while it also reduced speciation, albeit weaker ($G=-0.069$, $\omega=0.995$). A possible explanation is that Southeast Asia was still separated from the Australian and Pacific plates by deep oceans^{22,41}, so high sea level may not contribute to form large areas of shelf habitats in the IAA. Most interestingly, high temperature triggered both speciation ($G=0.584$, $\omega=0.993$) and extinction ($G=5.038$, $\omega=0.999$) during

the warm phase, which may be due to the coexistence of warm-adapted and cold-adapted species. As Earth's climate showed a long-term cooling trend across the Cenozoic ⁴², some warm-adapted species that were resilient to or even preferred very high temperatures of the Eocene and Oligocene might not yet be extinct in the relatively warm early-middle Miocene ⁴³. They may have responded to temperature positively with more active proliferation. Cold-adapted species on the other hand gradually rose throughout the Neogene as the climate further cooled ^{4,43,44}, but their vulnerability to high temperature may put them in much higher extinction risks during the relative warm early-middle Miocene. Indeed, recent studies suggest that tropical temperatures >25°C may be too elevated for marine organisms due to metabolic trade-offs driven by temperature effects on hypoxia sensitivity ^{25,45-47}. Consistent evidence for thermal stress on tropical biodiversity comes from various marine taxa (e.g., coral, reef fish, reptile, calcareous algae) that declined in diversity in the tropics and exhibited poleward range shifts during historical warm intervals ^{48,49}. Since temperature has a much stronger impact on extinction than on speciation ($G\lambda=0.58$, $G\mu=5.04$), our analysis suggests that too high a tropical temperature in warm climates slows down diversification and hinders the growth of the hotspot. The effects of temperature are not significant for other time windows, suggesting that thermal stress may be relieved in cooler climates. We need to be aware of future extinction risks and tropical biodiversity loss exacerbated by anthropogenic warming under high greenhouse gas emission scenarios, as tropical biotas are already living close to their ecophysiological thresholds ⁴⁶.”

Please see the new Figure 2 under Comment No. 5.

75. Line 275: Slight reword suggested ‘The 12-Ma event is apparent only in the speciation rate...’

Response: We changed the sentence accordingly.

76. Lines 287–289: *Could the increase and subsequent decrease in speciation and extinction around 5 Ma also be due to closure of the Central American Seaway? Increased diversity/richness with increased extinction/origination rates are apparent for radiolarians (Lowery et al., 2020 Annual Review of Earth and Planetary Sciences) and foraminifera at global (Fraass et al., 2015 Annual Review of Earth and Planetary Sciences) and regional (Lam and Leckie, 2020, Micropaleontology) scales. The constriction of the CAS reached a critical threshold around 4.8-4.0 Ma (e.g., Haug & Tiedemann, 1998; Steph et al., 2006, 2010).*

Response: Thank you for this point, but we tend to think that the evidence from radiolarians and planktic foraminifera may not be of key importance to our story. As we discussed under Comment No. 22, planktons do not show a strong IAA center of their diversity distribution. Their diversity changes at 5 Ma at regional-global scales may not be controlled by the same mechanisms as those controlling benthic organisms in the IAA. Moreover, the closure of the Central American Seaway may have a global-scale effect on tropical biota, but the significant restriction of the Indonesian Seaway at

5 Ma may control more strongly and directly the IAA diversification at a regional scale. There is strong evidence for this event triggering prominent biotic changes across the Indian and Pacific oceans (Srinivasan & Sinha, 1998, 2000).

Thus, we explained the 5 Ma peak at Line 358 as follows:

“Finally, the 5-Ma speciation and extinction peaks may be related to the restriction of the Indonesian Throughflow at the Miocene-Pliocene boundary ^{41,53}, which allowed reversed faunal dispersal from the Indian Ocean to the Pacific Ocean, as discussed in Ref. ³.”

77. Line 299: Add ‘found’ between ‘that’ and ‘huge’

Response: We added it accordingly.

78. Lines 298–302: This sentence is a bit chewy, and I don’t completely follow and understand. Are you stating that huge differences in extinction rates among different taxa contribute to modern longitudinal diversity patterns? I think just a re-wording here for clarity is necessary.

Response: We revised the concluding paragraph. Please see it under Comment No. 44.

79. Line 361: Delete the ‘in’ at the end of this sentence.

Response: We deleted it accordingly.

80. Lines 593–595: Include references for your biostratigraphic schemes used (e.g., King et al. 2020 and Wade et al., 2011 for Cenozoic planktic foraminiferal biozone ages). Also, what time scale were the age models put on? Geologic Time Scale 2020 or other?

Response: We followed the Geological Time Scale 2020 and the biostratigraphic schemes used there (Raffi et al., 2020; Speijer et al., 2020).

Referee #4 (Remarks to the Author):

81. The manuscript “Cenozoic history of the tropical marine biodiversity hotspot” uses ostracod records and PyRate/Birth-Death models to look at origin vs extinction rate through time and biodiversity within the Coral Triangle. The paper identifies an exponential increase in biodiversity with several peaks, one of which it is postulated could be related to the closure of the Tethys seaway. Temperature and diversity are the drivers identified for biodiversity. The stability ie. lack of major extinction events in the region is also a major factor in creating today’s biodiversity in the region. In addition it also looks at the amount of Tethyan biodiversity in the assemblages and what factors may have influenced this through time.

Biodiversity, drivers of diversity and the future of the Coral Triangle are current major concerns, the paper is therefore exceptionally timely and offers a new perspective on this. Ostracode data have not been used to model Coral Triangle biodiversity before. Understanding where this biodiversity comes from is exceptionally important for our understanding of current and future biodiversity. But is also a difficult task, and therefore new perspectives are significant and of broad interest.

Response: Thank you very much for your positive and constructive comments.

82. It is interesting to see a study for the region carried out with ostracodes as previous work, as the authors mention, has been carried out on more reefal organisms such as bryozoans, corals, and larger foraminifera. Whilst no single record is perfect, it is important for multiple perspectives and records to be used. Ostracodes therefore add to this. Perhaps the authors could comment on, or further emphasise whether ostracodes have mirrored trends of larger organisms in previous studies.

Response: This is a very good point. Please see our reply and revisions to Comment No. 2.

83. The methods used are largely standard PyRate methods, seem appropriate for the dataset and are well described for others to follow. Figures are largely appropriate and clear although I have two smaller comments:

- I think the authors do need to clarify for figure 1 that the rate of speciation given is per lineage as this is not immediately clear for general readers who do not use these methods.*
- Within figure 1D it is unclear if these numbers are taken from PyRate or from counts as this could result in small differences in the timing of plateaus.*
- Something to consider but not essential is whether it might be useful to plot the extinction and origination rates on the same plot to allow easy comparison on the same scale.*

Response: In the caption of Figure 1, we specified Bayesian inferences of ‘speciation rates per lineage’, ‘extinction rates per lineage’, and ‘species richness of all ostracods’ for panels A, B, and D, respectively. We also adjusted the y-axes of panels A and B to the same scale to allow more direct comparison.

84. The largest issue with the dataset is the lack of data in the Eocene and Oligocene and given the geology of the region this is not necessarily something that can be fixed. Of the 216 samples only three are Eocene and none from the Oligocene and the graphs show very low diversity across this interval. Although PyRate can statistically account for sampling bias and preservation etc. it still is working from the data that it is given and would therefore be an underestimate of any Eocene-Oligocene biodiversity. This is very difficult to address as suitable samples likely simply do not exist or are exceptionally rare – much of the Eocene of the region is hard limestone, unsuitable for ostracod studies. The authors do mention the lack of samples, but they need to be tentative regarding any conclusions based on the Eocene and

Oligocene. Another possibility is that the paper could be framed as a more Neogene rather than Cenozoic perspective, as they do have a good number of samples from the Early Miocene onwards.

I am sure the authors are well aware of this issue but I would like to know their further thoughts on how valid the model results are for Eocene-Oligocene and whether the lack of data early on influences the interpretation of the first diversity peak.

The authors in line 263 talk about Tethyan taxa arriving in the IAA in the early Miocene creating a diversity peak, although in figure 3 it appears that there is a greater diversity of Tethyan taxa in the Late Miocene?

Response: Thank you very much for this thoughtful comment. We performed sensitivity analyses to show the lack of Oligocene data does not strongly affect our understanding of the overall diversification trend and its long-term drivers. Please refer to our response to Comment No. 3.

Regarding the Tethyan genera, they migrated to the IAA prior to the final closure of the Tethys Seaway at ~20 Ma. These Tethyan ancestors proliferated at species level in the IAA during the middle-late Miocene and contributed to the growing species richness of this group. We clarified this at Line 394 and as follows:

“Prior to the final closure of the Tethys Seaway, the Tethyan ancestors were able to reach the IAA after a long-distance dispersal. They persisted and proliferated in the IAA even after the seaway closed, and thereby supported the early-stage development of the IAA hotspot.”

85. It is interesting that speciation increases with the middle Miocene Climatic Optimum, and I agree with the authors that it could be related to other changes related to circulation etc. Warming also does not seem to have always been detrimental to all shallow water organisms, following the PETM larger foraminifera massively diversified, possibly due to available niches but it remains fairly uncertain.

Response: We totally agree with the reviewer that temperature regulation on biodiversity can be nuanced and complicated in different climate regimes over evolutionary time. In fact, by the time-stratified MBD analysis, we showed that high temperature in warm climates positively affect both speciation and extinction rates (Please refer to our response to Comment No. 15), which may explain the biotic event at 16 Ma.

We revised the corresponding text at Line 351 to be:

“Then, the 16-Ma peak coincided with the Mid-Miocene Climatic Optimum⁴³, which is in line with our MBD results indicating both speciation and extinction rates accelerated with high temperature in a warm climate state (Fig. 2B).”

86. There is good discussion of events and possible causes along with comparison to the

Caribbean, but I wonder if some kind of summary figure could be made indicating the tentative causes for the peaks and timing of Caribbean events.

Response: This is an excellent idea. We made the Extended data Figure 5 to show the short-term biotic events in relation to paleoenvironmental transitions.

[REDACTED]

87. The conclusions regarding Tethyan taxa are sensible, and the paper finishes with a strong valid point regarding the future.

Response: We appreciate your positive comment.

88. References are appropriate and cover the topic well and it is overall well-written, and reads clearly.

Response: We appreciate your positive comment.

Sincerely yours,

Skye Yunshu Tian and Moriaki Yasuhara, corresponding authors

--End of the Letter--

References

- Bellwood, D. R., Renema, W., & Rosen, B. R. (2012). Biodiversity hotspots, evolution and coral reef biogeography. In *Biotic Evolution and Environmental Change in Southeast Asia* (Vol. 82, pp. 216-245). Cambridge University Press.
- Boag, T. H., Gearty, W., & Stockey, R. G. (2021). Metabolic tradeoffs control biodiversity gradients through geological time. *Current Biology*, *31*(13), 2906-2913. e2903.
- Bowen, B. W., Rocha, L. A., Toonen, R. J., & Karl, S. A. (2013). The origins of tropical marine biodiversity. *Trends in Ecology & Evolution*, *28*(6), 359-366.
- Bribiesca-Contreras, G., Verbruggen, H., Hugall, A. F., & O'Hara, T. D. (2019). Global biogeographic structuring of tropical shallow-water brittle stars. *Journal of Biogeography*, *46*(7), 1287-1299.
- Condamine, F. L., Guinot, G., Benton, M. J., & Currie, P. J. (2021). Dinosaur biodiversity declined well before the asteroid impact, influenced by ecological and environmental pressures. *Nature Communications*, *12*(1), 1-16.
- Condamine, F. L., Romieu, J., & Guinot, G. (2019). Climate cooling and clade competition likely drove the decline of lamniform sharks. *Proceedings of the National Academy of Sciences*, *116*(41), 20584-20590.
- Condamine, F. L., Silvestro, D., Koppelhus, E. B., & Antonelli, A. (2020). The rise of angiosperms pushed conifers to decline during global cooling. *Proceedings of the National Academy of Sciences*, *117*(46), 28867-28875.
- Cornell, H. V. (2013). Is regional species diversity bounded or unbounded? *Biological Reviews*, *88*(1), 140-165.
- Costello, M. J., Corkrey, R., Bates, A. E., Burrows, M. T., Chaudhary, C., Edgar, G. E., Stuart-Smith, R. D., Yasuhara, M., & Wei, C.-L. (2023). The universal evolutionary and ecological significance of 20 oC. *Frontiers of Biogeography*, *15*(4).
- Cowman, P. F., & Bellwood, D. R. (2013a). The historical biogeography of coral reef fishes: global patterns of origination and dispersal. *Journal of Biogeography*, *40*(2), 209-224.
- Cowman, P. F., & Bellwood, D. R. (2013b). Vicariance across major marine biogeographic barriers: temporal concordance and the relative intensity of hard versus soft barriers. *Proceedings of the royal society B: biological sciences*, *280*(1768), 20131541.
- Crame, J., & Rosen, B. (2002). Cenozoic palaeogeography and the rise of modern biodiversity patterns. *Geological Society, London, Special Publications*, *194*(1), 153-168.
- Cronin, T. M. (1987). Evolution, biogeography, and systematics of Puriana: Evolution and speciation in Ostracoda, III. *Memoir (The Paleontological Society)*, 1-71.
- Cronin, T. M. (1988). Geographical isolation in marine species: Evolution and speciation in Ostracoda, I. *Developments in Palaeontology and Stratigraphy*, *11*, 871-889.
- Cronin, T. M., & Schmidt, N. (1988). Evolution and biogeography of Orionina in the Atlantic, Pacific and Caribbean: evolution and speciation in Ostracoda, II. *Developments in Palaeontology and Stratigraphy*, *11*, 927-938.
- Estronza, A. M. G., Alfaro, M., & Schizas, N. V. (2017). Morphological and genetic species diversity in ostracods (Crustacea: Oligostraca) from Caribbean reefs. *Marine Biodiversity*, *47*(1), 37-53.
- Fenton, I. S., Aze, T., Farnsworth, A., Valdes, P., & Saupe, E. E. (2023). Origination of the modern-style diversity gradient 15 million years ago. *Nature*, *614*(7949), 708-712.
- Förderer, M., Rödder, D., & Langer, M. R. (2018). Patterns of species richness and the center of diversity in modern Indo-Pacific larger foraminifera. *Scientific Reports*, *8*(1), 1-9.
- Hall, R. (2009). Southeast Asia's changing palaeogeography. *Blumea-Biodiversity, Evolution and Biogeography of Plants*, *54*(1-2), 148-161.
- Hoeksema, B. W. (2007). Delineation of the Indo-Malayan centre of maximum marine biodiversity: the Coral Triangle. In *Biogeography, time, and place: distributions, barriers, and islands* (pp. 117-178). Springer.
- Hou, Z., & Li, S. (2018). Tethyan changes shaped aquatic diversification. *Biological Reviews*, *93*(2), 874-896.

- Johnson, K. G., Hasibuan, F., Mueller, W., & Todd, J. A. (2015). Biotic and environmental origins of the southeast Asian marine biodiversity hotspot: The throughflow project. *Palaios*, 30(1), 1-6.
- Johnson, K. G., Renema, W., Rosen, B. R., & Santodomingo, N. (2015). Old data for old questions: what can the historical collections really tell us about the Neogene origins of reef-coral diversity in the Coral Triangle? *Palaios*, 30(1), 94-108.
- Karanovic, I., Huyen, P. T. M., Yoo, H., Nakao, Y., & Tsukagoshi, A. (2020). Shell and appendages variability in two allopatric ostracod species seen through the light of molecular data. *Contributions to Zoology*, 89(3), 247-269.
- Lehtonen, S., Silvestro, D., Karger, D. N., Scotese, C., Tuomisto, H., Kessler, M., Peña, C., Wahlberg, N., & Antonelli, A. (2017). Environmentally driven extinction and opportunistic origination explain fern diversification patterns. *Scientific Reports*, 7(1), 4831.
- Leprieur, F., Descombes, P., Gaboriau, T., Cowman, P. F., Parravicini, V., Kulbicki, M., Melián, C. J., De Santana, C. N., Heine, C., & Mouillot, D. (2016). Plate tectonics drive tropical reef biodiversity dynamics. *Nature Communications*, 7, 1-8.
- Neubauer, T. A., Hauffe, T., Silvestro, D., Scotese, C. R., Stelbrink, B., Albrecht, C., Delicado, D., Harzhauser, M., & Wilke, T. (2022). Drivers of diversification in freshwater gastropods vary over deep time. *Proceedings of the royal society B: biological sciences*, 289(1968), 20212057.
- Rabosky, D. L. (2013). Diversity-dependence, ecological speciation, and the role of competition in macroevolution. *Annual Review of Ecology, Evolution, and Systematics*, 44, 481-502.
- Raffi, I., Wade, B., Pälke, H., Beu, A., Cooper, R., Crundwell, M., Krijgsman, W., Moore, T., Raine, I., & Sardella, R. (2020). The neogene period. In *Geologic time scale 2020* (pp. 1141-1215). Elsevier.
- Renema, W., Bellwood, D., Braga, J., Bromfield, K., Hall, R., Johnson, K., Lunt, P., Meyer, C., McMonagle, L., & Morley, R. (2008). Hopping hotspots: global shifts in marine biodiversity. *Science*, 321(5889), 654-657.
- Sanciango, J. C., Carpenter, K. E., Etnoyer, P. J., & Moretzsohn, F. (2013). Habitat availability and heterogeneity and the Indo-Pacific warm pool as predictors of marine species richness in the tropical Indo-Pacific. *PLoS One*, 8(2), e56245.
- Schön, I., Pieri, V., Sherbakov, D. Y., & Martens, K. (2017). Cryptic diversity and speciation in endemic Cytherissa (Ostracoda, Crustacea) from Lake Baikal. *Hydrobiologia*, 800, 61-79.
- Schön, I., Pinto, R. L., Halse, S., Smith, A. J., Martens, K., & Birky Jr, C. W. (2012). Cryptic species in putative ancient asexual darwinulids (Crustacea, Ostracoda). *PLoS One*, 7(7), e39844.
- Scotese, C. R., Song, H., Mills, B. J., & van der Meer, D. G. (2021). Phanerozoic paleotemperatures: The earth's changing climate during the last 540 million years. *Earth-Science Reviews*, 215, 103503.
- Silvestro, D., Schnitzler, J., Liow, L. H., Antonelli, A., & Salamin, N. (2014). Bayesian estimation of speciation and extinction from incomplete fossil occurrence data. *Systematic Biology*, 63(3), 349-367.
- Speijer, R., Pälke, H., Hollis, C., Hooker, J., & Ogg, J. (2020). The paleogene period. In *Geologic time scale 2020* (pp. 1087-1140). Elsevier.
- Srinivasan, M., & Sinha, D. (1998). Early Pliocene closing of the Indonesian Seaway: evidence from north-east Indian Ocean and Tropical Pacific deep sea cores. *Journal of Asian Earth Sciences*, 16(1), 29-44.
- Srinivasan, M., & Sinha, D. (2000). Ocean circulation in the tropical Indo-Pacific during early Pliocene (5.6-4.2 Ma): Paleobiogeographic and isotopic evidence. *Journal of Earth System Science*, 109(3), 315-328.
- Tian, S. Y., Yasuhara, M., Huang, H.-H. M., Condamine, F. L., & Robinson, M. M. (2021). Shallow marine ecosystem collapse and recovery during the Paleocene-Eocene Thermal Maximum. *Global and Planetary Change*, 207, 103649.
- Tittensor, D. P., Mora, C., Jetz, W., Lotze, H. K., Ricard, D., Berghe, E. V., & Worm, B. (2010). Global patterns and predictors of marine biodiversity across taxa. *Nature*, 466(7310), 1098.
- Veron, J. E., DeVantier, L. M., Turak, E., Green, A. L., Kininmonth, S., Stafford-Smith, M., & Peterson, N. (2011). The coral triangle. *Coral reefs: an ecosystem in transition*, 47-55.
- Westerhold, T., Marwan, N., Drury, A. J., Liebrand, D., Agnini, C., Anagnostou, E., Barnet, J. S., Bohaty, S. M., De Vleeschouwer, D., & Florindo, F. (2020). An astronomically dated record of Earth's climate and its predictability over the last 66 million years. *Science*, 369(6509), 1383-1387.
- Woodhouse, A., Swain, A., Fagan, W. F., Fraass, A. J., & Lowery, C. M. (2023). Late Cenozoic cooling restructured global marine plankton communities. *Nature*, 614(7949), 713-718.
- Yasuhara, M., Huang, H.-H. M., Reuter, M., Tian, S. Y., Cybulski, J. D., O'dea, A., Mamo, B. L., Cotton, L. J., Di Martino, E., & Feng, R. (2022). Hotspots of Cenozoic tropical marine biodiversity. *Oceanography and marine biology: an annual review*, 60, 243-300.
- Yasuhara, M., Tittensor, D. P., Hillebrand, H., & Worm, B. (2017). Combining marine macroecology and palaeoecology in understanding biodiversity: microfossils as a model. *Biological Reviews*, 92(1), 199-215.
- Yasuhara, M., Wei, C.-L., Kucera, M., Costello, M. J., Tittensor, D. P., Kiessling, W., Bonebrake, T. C., Tabor, C. R., Feng, R., & Baselga, A. (2020). Past and future decline of tropical pelagic biodiversity. *Proceedings of the National Academy of Sciences*, 117(23), 12891-12896.

Reviewer Reports on the First Revision:

Referee #1 (Remarks to the Author):

I thank the authors for their efforts to revise their manuscript. The revisions are extensive and impressive, and they have largely addressed all of my previous concerns. I did have a few remaining thoughts, listed below:

- 1) The manuscript is still difficult to follow, without clear, take-home messages. This is fine, as it's a complex topic with many components, but it is not an easy 'sell' for Nature. I would (still) be hard pressed to say what are the key findings of this study.
- 2) As a reader, I would appreciate a more comprehensive paragraph on the main methods used (i.e., a road map to follow).
- 3) I agree with Reviewer 4 that the focus of this paper should be on Neogene dynamics rather than Cenozoic dynamics.
- 4) Please revise for minor typos and English usage throughout.

Referee #2 (Remarks to the Author):

Dear Authors,

Thank you for resubmitting your manuscript and taking into consideration the changes requested by the reviewers.

The study reveals the dynamics of the IAA hotspot over the past 40 million years with insights on the tectonic and biodiversity evolution of the region which are relevant to a wide range of fields. Following on from revision, the significance of the study is clearer and the data and methodology support the study and the conclusion clearly and concisely.

Before publication, please consider the very minor edits I have suggested to further improve the clarity of the study. This is mainly in the form of minor grammar and nomenclature points.

I look forward to the study being published.

All the best,

Dr Adam Woodhouse

Line 89/90 - Consider restructuring this sentence, it is unclear

Something more akin to:

"However, to uncover the long-term history of the IAA biodiversity, we must disentangle the biological patterns within the scarce and patchy Cenozoic and Neogene fossil record of the region"

Line 220 - dampened?

Line 285 (plus other occurrences) - 23.03 Ma is not just the Neogene but also the Quaternary, this should be included

Line 303 - more weakly

Line 671 - The record is not for the entire Cenozoic

Referee #3 (Remarks to the Author):

The revised manuscript by Tian et al. 'Cenozoic history of the tropical marine biodiversity hotspot', is well-structured, of sound reasoning throughout, and meets all the qualifications to now be published in Nature. I commend the authors for taking into account all four reviewer's comments and suggestions, and providing such a thorough response letter. I am quite excited to see this paper published and look forward to citing it in future studies!

Referee #4 (Remarks to the Author):

The author's have substantially revised the manuscript and addressed comments from all reviewers exceptionally thoroughly. Then have re-run analyses, and found it did not affect the major outcomes of the paper. Specifically they clearly outline why ostracodes can be used as a representative group for diversity. And also addressed the gap in the Oligocene data via testing sensitivity and also carrying out analysis of the post Oligocene (ie. Neogene dataset). The addition of events to figure 5 in the extended data is also helpful

Overall I think it is a well written, exceptionally interesting and timely study on the large scale long term drivers of diversity and recommend it for publication. The authors are very honest in the text about the limitations of their dataset and in the interpretation of their data. I agree with the authors regarding their preference of using global climate datasets over regional models. Models are constantly being revised and often have different results. And fossil studies are always in some way limited by the data and resolution of data that is available.

I have a very small comment on line 310 where the authors refer to the high temperatures of the Eocene and Oligocene. I know what the authors mean, but it reads slightly strangely to me as the Oligocene is an 'ice-house' and compared to the beginning of the Eocene is cooler.

When discussing taxa that evolved under warmer conditions and extinction risk it may be worth mentioning this paper:

Mathes, G.H., van Dijk, J., Kiessling, W. et al. Extinction risk controlled by interaction of long-term and short-term climate change. *Nat Ecol Evol* 5, 304–310 (2021). <https://doi.org/10.1038/s41559-020-01377-w>

Also in terms of temperature drivers, I think it is not possible to test because the data does not really exist across the geological record yet. But could changes in seasonality rather than average temperature be potentially causing some of the differences seen between the warm and cold phases? I would be interested to know the authors thoughts - I do not know how seasonally influenced ostracodes are.

Author Rebuttals to First Revision:

We deeply appreciate your positive decision and helpful comments to improve our manuscript. Here we have addressed all remaining issues raised by referees, and point-by-point responses are shown in this revision letter below in **bold font**.

Referee #1 (Remarks to the Author):

I thank the authors for their efforts to revise their manuscript. The revisions are extensive and impressive, and they have largely addressed all of my previous concerns. I did have a few remaining thoughts, listed below:

1) The manuscript is still difficult to follow, without clear, take-home messages. This is fine, as it's a complex topic with many components, but it is not an easy 'sell' for Nature. I would (still) be hard pressed to say what are the key findings of this study.

Response: Thank you for bringing out this point. We further emphasize our take-home message in the last paragraph that ideal environmental conditions in terms of climate, habitat, and absence of major extinctions have made the IAA the largest biodiversity hotspot, while humans are doing the opposite to destroy it. The revised text is as follows: "Our results suggest that the increase in habitat size and immunity to major extinction events during the Cenozoic allowed for strong, uninterrupted long-term diversification that reached an asymptote in the IAA, making it the current global center of biodiversity. The relief of thermal stress as Earth's climate transformed from a warm to a cold state was also essential for this process. It took tens of millions of years for the IAA hotspot to fully develop in an ideal environment in terms of climate, habitat, and biogeographic connectivity, luckily without a major extinction event. As we know, the opposite is now happening, with anthropogenic warming⁵⁷, habitat destruction⁵⁸, and the sixth mass extinction^{44,59}. All may effectively undermine the IAA hotspot and impair its function as a cradle and museum of tropical biodiversity. Our paleobiological results highlight the slow evolution, in contrast to the ongoing decadal degradation, of the tropical hotspot as one of the most fascinating biodiversity and biogeographic patterns, urging conservation efforts."

2) As a reader, I would appreciate a more comprehensive paragraph on the main methods used (i.e., a road map to follow).

Response: An introductory paragraph to explicitly explain the structure of the Materials and Methods section could be as follows:

"In this section, we first summarize how we obtained the Cenozoic IAA fossil dataset, integrating our original data with published data. We then detail the application of birth-death models to reconstruct the IAA's diversity history, and to uncover its controlling factors. Finally, we describe the raw analyses of the fossil dataset from a biogeographic perspective."

We can add a short paragraph like the one above at the beginning of the Materials and Methods section, but only if the *Nature* editorial office thinks it is necessary, because we feel it is rather optional and this kind of summary of methods is not common in *Nature* papers.

3) I agree with Reviewer 4 that the focus of this paper should be on Neogene dynamics rather than Cenozoic dynamics.

Response: Indeed, we focused on the Neogene history of diversification and biogeography in our Discussion in the first round of revisions. However, as we do have a sample coverage of the Eocene, and our results from the birth-death models indicate that the Oligocene gap does not affect the long-term diversity trends, we think it is reasonable to present this paper as the Cenozoic dynamics even though the Neogene is explored in much better detail compared with the Paleogene.

4) Please revise for minor typos and English usage throughout.

Response: We corrected all minor mistakes in language as we spotted them.

Referee #2 (Remarks to the Author):

Dear Authors,

Thank you for resubmitting your manuscript and taking into consideration the changes requested by the reviewers.

The study reveals the dynamics of the IAA hotspot over the past 40 million years with insights on the tectonic and biodiversity evolution of the region which are relevant to a wide range of fields. Following on from revision, the significance of the study is clearer and the data and methodology support the study and the conclusion clearly and concisely.

Before publication, please consider the very minor edits I have suggested to further improve the clarity of the study. This is mainly in the form of minor grammar and nomenclature points.

I look forward to the study being published.

All the best,

Dr Adam Woodhouse

Response: Thank you very much for your recognition of our study.

Line 89/90 - Consider restructuring this sentence, it is unclear

Something more akin to:

"However, to uncover the long-term history of the IAA biodiversity, we must disentangle the biological patterns within the scarce and patchy Cenozoic and Neogene fossil record of the region"

Response: Since we realized this sentence is a bit repetitive, we revised it as:

“However, the detailed Cenozoic history of the IAA hotspot remains elusive as explicated in the next paragraph.”

Line 220 - dampened?

Response: We referred to previous publications, Cornell (2013) for example, regarding the usage of “damped” instead of “dampened”.

Line 285 (plus other occurrences) - 23.03 Ma is not just the Neogene but also the Quaternary, this should be included

Response: We included “Quaternary” in all relevant places.

Line 303 - more weakly

Response: We revised accordingly.

Line 671 - The record is not for the entire Cenozoic

Response: We removed “entire” from the sentence.

Referee #3 (Remarks to the Author):

The revised manuscript by Tian et al. 'Cenozoic history of the tropical marine biodiversity hotspot', is well-structured, of sound reasoning throughout, and meets all the qualifications to now be published in Nature. I commend the authors for taking into account all four reviewer's comments and suggestions, and providing such a thorough response letter. I am quite excited to see this paper published and look forward to citing it in future studies!

Response: We really appreciate your commendation.

Referee #4 (Remarks to the Author):

The author's have substantially revised the manuscript and addressed comments from all reviewers exceptionally thoroughly. Then have re-run analyses, and found it did not affect the major outcomes of the paper. Specifically they clearly outline why ostracodes can be used as a representative group for diversity. And also addressed the gap in the Oligocene data via testing sensitivity and also carrying out analysis of the post Oligocene (ie. Neogene dataset). The addition of events to figure 5 in the extended data is also helpful

Overall I think it is a well written, exceptionally interesting and timely study on the large scale long term drivers of diversity and recommend it for publication. The authors are very honest in the text about the limitations of their dataset and in the interpretation of their data. I agree with the authors regarding their preference of using global climate datasets over regional models. Models are constantly being revised and often have different results. And fossil studies are always in some way limited by the data and resolution of data that is available.

I have a very small comment on line 310 where the authors refer to the high temperatures of the Eocene and Oligocene. I know what the authors mean, but it reads slightly strangely to me as the Oligocene is an 'ice-house' and compared to the beginning of the Eocene is cooler.

Response: Thank you very much for your recognition of our study. We have deleted "Oligocene" from the corresponding sentence to avoid any misunderstanding.

When discussing taxa that evolved under warmer conditions and extinction risk it may be worth mentioning this paper:

Mathes, G.H., van Dijk, J., Kiessling, W. et al. Extinction risk controlled by interaction of long-term and short-term climate change. *Nat Ecol Evol* 5, 304–310 (2021).

<https://doi.org/10.1038/s41559-020-01377-w>

Response: Thank you for recommending this excellent paper. We have now cited it accordingly.

Also in terms of temperature drivers, I think it is not possible to test because the data does not really exist across the geological record yet. But could changes in seasonality rather than average temperature be potentially causing some of the differences seen between the warm and cold phases? I would be interested to know the authors thoughts - I do not know how seasonally influenced ostracodes are.

Response: The effects of seasonality on biodiversity are interesting to explore. The colder phase could potentially have stronger seasonality, but we tend to think it is too speculative to discuss further how this might affect biodiversity. To address this issue, we think it might be good to start by analyzing modern biological data of different taxonomic groups (e.g., OBIS data) with available temperature and seasonality records (e.g., World Ocean Atlas). However, this is beyond the scope of this study. In addition, temperature at a particular season may be important for certain species or taxonomic groups, but averaged temperatures usually capture/include such a signal, although it may be somewhat blurred.

Sincerely yours,

Skye Yunshu Tian, Moriaki Yasuhara, Fabien L. Condamine, on behalf of the co-authors

--End of the Letter--

References

Cornell, H. V. (2013). Is regional species diversity bounded or unbounded? *Biological Reviews*, 88(1), 140-165.